# LLM Interpretability with Identifiable Temporal-Instantaneous Representation

**Xiangchen Song**[†,1]    **Jiaqi Sun**[†,1]    **Zijian Li**[2]    **Yujia Zheng**[1]    **Kun Zhang**[1,2]

[1]Carnegie Mellon University
[2]Mohamed bin Zayed University of Artificial Intelligence
{xiangchensong,jiaqisun,kunz1}@cmu.edu

## Abstract

Despite Large Language Models' remarkable capabilities, understanding their internal representations remains challenging. Mechanistic interpretability tools such as sparse autoencoders (SAEs) were developed to extract interpretable features from LLMs but lack *temporal dependency modeling*, *instantaneous relation representation*, and more importantly *theoretical guarantees*—undermining both the theoretical foundations and the practical confidence necessary for subsequent analyses. While causal representation learning (CRL) offers theoretically-grounded approaches for uncovering latent concepts, existing methods *cannot* scale to LLMs' rich conceptual space due to inefficient computation. To bridge the gap, we introduce an identifiable temporal causal representation learning framework specifically designed for LLMs' high-dimensional concept space, capturing both time-delayed and instantaneous causal relations. Our approach provides theoretical guarantees and demonstrates efficacy on synthetic datasets scaled to match real-world complexity. By extending SAE techniques with our temporal causal framework, we successfully discover meaningful concept relationships in LLM activations. Our findings show that modeling both temporal and instantaneous conceptual relationships advances the interpretability of LLMs.

## 1 Introduction

Large Language Models (LLMs) have demonstrated remarkable capabilities across a wide range of natural language tasks, from question answering to content generation. Despite these achievements, a fundamental understanding of their internal representations remains underexplored. This gap between performance and interpretability poses significant challenges for ensuring the reliability, safety, and appropriate deployment of these increasingly powerful systems [47].

Mechanistic interpretability (MI) aims to bridge this gap by reverse-engineering neural networks to understand how they process and represent information [14]. Among all MI tools, sparse autoencoders (SAEs) have emerged as a promising approach for extracting interpretable features from LLMs [14, 52]. By decomposing the high-dimensional activations of LLMs into sparse, monosemantic features, SAEs help identify the basic units of computation within these complex systems. However, SAEs present several limitations that restrict their utility for comprehensive model understanding:

First, SAEs treat each feature as an isolated representation, failing to capture how features influence one another. This omission disregards semantic connections and transitions within a sequence, which are known as temporal or time-delayed relationships between features*. Second, SAEs lack mechanisms to represent instantaneous or logical relationships between features, such as mutual

---

[†]Equal contribution.
*Alternative approaches, such as [1, 30], use attention scores from the LLM to infer time-delayed influence.

exclusivity or co-occurrence constraints [32]. These relationships complement the temporal dynamics by encoding structural dependencies within the same time step. Third, and most critically, SAEs offer no theoretical guarantees of the uniqueness of the recovered features. This absence undermines confidence that the extracted features reflect meaningful and stable latent variables, rather than arbitrary or unstable transformations [54].

Fortunately, to address these limitations, the causal representation learning (CRL) community has proposed a range of promising frameworks with theoretical guarantees [47]. For instance, [27] and [31] use sparse causal influence and interventions to uncover temporal and instantaneous relationships among latent variables. However, these methods face significant scalability challenges due to the computational inefficiency of estimating Jacobians. As a result, they typically scale to only dozens or hundreds of concepts [56], while interpretability in LLMs demands efficient modeling of thousands or even tens of thousands of concept features [52]. In summary, although CRL offers strong theoretical guarantees for recovering meaningful features and their causal relationships, its limited scalability in high-dimensional settings remains a major obstacle to practical deployment in LLM analysis.

To bridge this gap, in this paper, we introduce a computationally efficient temporal causal representation learning framework specifically designed for the high-dimensional activation space in LLMs. Our approach builds upon recent advances in both sparse autoencoders for LLMs and causal representation learning for sequential data. The key contributions of our work are:

(1) We propose a simple yet effective framework that jointly models time-delayed causal relations between concepts and instantaneous constraints, providing a more comprehensive understanding of how information flows through LLMs.

(2) Leveraging sparsity principle, we establish theoretical guarantees for our approach, making the representations learned reliable and explainable.

(3) Grounded in the theoretical result, we design scalable and efficient algorithms tailored to the high-dimensional concept space of LLMs, significantly extending prior work in CRL.

(4) We validate our approach on synthetic datasets scaled to match real-world complexity and demonstrate its effectiveness when applied to activations from real LLMs.

## 2  Problem Setting

We begin by characterizing the generation process of LLM activations to establish interpretability guarantees. These activations—signals produced during inference—are widely assumed to be linearly generated from hidden concepts, consistent with sparse autoencoder (SAE) literature [3, 18]. However, existing formulations typically treat these concepts as independent, overlooking dependencies between them. In reality, earlier-token semantics often influence later tokens, and token generation depends jointly on the activation of multiple concepts. To account for these interactions, we introduce a data generation process with both temporal and instantaneous relations, adopting CRL terminology. Given a token sequence $s = (v_1, \ldots, v_k)$, let $\mathbf{x}_t = (x_{t,1}, \ldots, x_{t,m})$ be the $n$-dimensional activation vector at token $v_t$ for a specific layer. Following the linear representation hypothesis [43] and SAE formulation [3, 18], we assume:

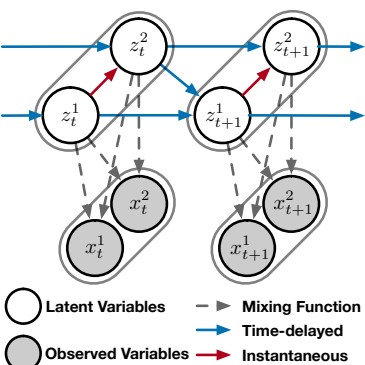

Figure 1: Graphical illustration of the data generation process.

$$\mathbf{x}_t = \mathbf{g}(\mathbf{z}_t), \tag{1}$$

where $\mathbf{g} : \mathbb{R}^n \to \mathbb{R}^m$ is the linear mixing function, and $\mathbf{x}_t$ and $\mathbf{z}_t$ are observed and latent variables, respectively. Besides, each latent variable $z_{t,i}$ is governed by a structural equation model (SEM) capturing both time-delayed and instantaneous dependencies:

$$z_{t,i} = \sum_{\tau} \sum_{j \in \mathcal{J}_{i,\tau}} \mathbf{B}_{i,j,\tau} z_{t-\tau,j} + \sum_{j \in \mathcal{K}_i} \mathbf{M}_{i,j} z_{t,j} + \epsilon_{t,i}, \tag{2}$$

where $\mathbf{B}_{i,j,\tau}$ represents the coefficient for the time-delayed effect from $z_{t-\tau,j}$ to $z_{t,i}$; $\mathcal{J}_{i,\tau}$ is the set of indices of latent variables that have a time-delayed effect on $z_{t,i}$ with lag $\tau$; $\mathbf{M}_{i,j}$ represents the

coefficient for the instantaneous effect from $z_{t,j}$ to $z_{t,i}$; $\mathcal{K}_i$ is the set of indices of latent variables that have an instantaneous effect on $z_{t,i}$; and $\epsilon_{t,i}$ denotes the temporally and spatially independent noise extracted from a distribution $p_{\epsilon_i}$. The graphical model for this process is illustrated in Figure 1.

To better understand this data generation process in the context of LLM activations, $\mathbf{x}_t$ represents activations in a specific layer $l$ for token $v_t$, and the latent variables $\mathbf{z}_t$ can be considered as the underlying causal factors that generate these activations. In this case, the instantaneous effects (coefficients $\mathbf{M}_{i,j}$) reflect semantic or syntactic relationships between different latent factors within the same token, while time-delayed effects (coefficients $\mathbf{B}_{i,j,\tau}$) represent dependencies on previous tokens. Putting them together, the underlying data generating process can be written as

$$\underbrace{\mathbf{x}_t = \mathbf{A}\mathbf{z}_t}_{\text{Linear mixing}}, \quad \underbrace{z_{t,i} = \sum_{\tau > 0} \sum_{j \in \mathcal{J}_{i,\tau}} \mathbf{B}_{i,j,\tau} z_{t-\tau,j} + \sum_{j \in \mathcal{K}_i} \mathbf{M}_{i,j} z_{t,j} + \epsilon_{t,i}, \ i = 1, \dots, m}_{\text{Linear latent temporal SEM}}. \quad (3)$$

The linear latent temporal SEM in Eq. (3) induces two types of causal relationships: a time-delayed causal graph $\mathcal{G}_d$, in which an edge $z_{t-\tau,j} \to z_{t,i}$ exists if and only if $\mathbf{B}_{i,j,\tau} \neq 0$, and an instantaneous causal graph $\mathcal{G}_e$, in which an edge $z_{t,j} \to z_{t,i}$ exists if and only if $\mathbf{M}_{i,j} \neq 0$. We assume that $\mathcal{G}_e$ is acyclic, i.e., a directed acyclic graph (DAG), which implies that $\mathbf{M}$ can be permuted to a strictly lower-triangular form. Under this assumption, the conditional distribution of $\mathbf{z}_t$ given its past satisfies the Markov property with respect to $\mathcal{G}_e$ [45], namely $p(\mathbf{z}_t \mid \mathbf{z}_{<t}) = \prod_{i=1}^{n} p(z_{t,i} \mid \mathrm{Pa}_d(z_{t,i}), \mathrm{Pa}_e(z_{t,i}))$.

**Remark on the Linearity of the Model** We acknowledge that the internal mechanisms of LLMs are inherently nonlinear due to activation functions and attention mechanisms. However, our linear approach is justified by several considerations. First, many successful mechanistic interpretability techniques [15, 42, 10, 1, 30] rely on linear representation hypotheses as approximations of localized network behavior. Second, linear models provide an interpretable bridge between the complexity of neural networks and human understanding—they serve as simplified yet informative projections of the underlying causal mechanisms. Third, empirical evidence suggests that linear approximations can capture significant portions of variance in activations within specific contexts [37, 19], particularly when examining feature-to-feature relationships within a layer.

More importantly, existing causal representation learning (CRL) methods cannot efficiently handle hundreds of latent variables, often encountering out-of-memory issues and prohibitively long computation times. A detailed discussion of these limitations is presented in Section 5.1. While nonlinear interactions certainly exist, our linear framework offers a tractable foundation for identifying causal relationships that can later be extended to incorporate more complex dependencies. This approach follows the scientific principle of starting with simpler models that capture essential phenomena before introducing additional complexity.

## 3 Theoretical Guarantees

Recent work in causal representation learning, particularly for time-series data, has made substantial progress in handling both time-delayed and instantaneous causal relations. Under general assumptions on the data-generating process, strong identifiability results can be established, including recovery of latent variables up to component-wise transformations and estimation of the Markov network up to isomorphic equivalence. However, in our linear setting, the identifiability result of [27] is not directly applicable, since one key assumption (sufficient changes) cannot be satisfied; see Appendix A.1 for a detailed discussion. Therefore, in this section we establish identifiability by exploiting the autocovariance structure induced by linearity, following ideas inspired by [58].

**Theorem 1** (Latent Indeterminacy). *Suppose the estimated model $(\hat{\mathbf{A}}, \{\hat{\mathbf{B}}_\tau\}_{\tau=1}^{L}, \hat{\mathbf{M}}, p_{\hat{\epsilon}})$ and the true model $(\mathbf{A}, \{\mathbf{B}_\tau\}_{\tau=1}^{L}, \mathbf{M}, p_{\epsilon})$ both generate $\mathbf{x}_t$ according to Eq. 3 and are observationally equivalent on the autocovariance matrices $\mathbf{R}_{\mathbf{x}}(k)$ for $k = 0, 1, \dots, L$. If conditions A1–A4 hold, then the model parameters are identifiable up to the following indeterminacies:*

$$\hat{\epsilon}_t = \mathbf{P}\epsilon_t, \quad \hat{\mathbf{A}} = \mathbf{A}\mathbf{S}, \quad (\mathbf{I} - \hat{\mathbf{M}}) = \mathbf{P}(\mathbf{I} - \mathbf{M})\mathbf{S}, \quad \hat{\mathbf{B}}_\tau = \mathbf{P}\mathbf{B}_\tau\mathbf{S},$$

*where $\mathbf{S} \in \mathbb{R}^{n \times n}$ is invertible and $\mathbf{P}$ is a signed permutation matrix.*

- *A1 (Temporally white noise). Both $\mathbf{e}_t$ and $\epsilon_t$ are temporally white, and $\epsilon_t$ has i.i.d. mutually independent components. To remove scaling indeterminacy, assume $\mathbf{\Sigma}_{\epsilon_t} = \mathbf{I}$ and zero-mean data.*

- *A2 (Rank sufficiency).* $\mathbf{A} \in \mathbb{R}^{m \times n}$ $(m \geq n)$ *has full column rank, and* $\mathbf{B}_L$ *is full rank.*

- *A3 (Process stability).* *Eq. 22 defines a stable vector autoregression, i.e., all roots of* $\det(\mathbf{I} - \sum_{\tau=1}^{L} \mathbf{B}_\tau y^{-\tau}) = 0$ *lie strictly inside the unit circle.*

- *A4 (Non-Gaussianity).* *At most one component of* $\boldsymbol{\epsilon}_t$ *is Gaussian.*

**Proof Sketch** The proof consists of four steps. First, using the VAR representation of $\mathbf{z}_t$ and the linear mixing $\mathbf{x}_t = \mathbf{A}\mathbf{z}_t$, we derive Yule–Walker-type recursions for the autocovariances $\mathbf{R}_{\mathbf{x}}(k)$, yielding a linear system whose unknowns are the transformed coefficients $\mathbf{C}_\tau = \mathbf{A}(\mathbf{I}-\mathbf{M})^{-1}\mathbf{B}_\tau \mathbf{A}^{-1}$. Second, by stacking lagged observations, we show that the system's coefficient matrix equals the covariance of a finite stacked vector of $\{\mathbf{x}_t\}$, which is positive definite—and thus nonsingular—under the assumption of no nontrivial deterministic finite linear relations. This ensures identifiability of $\mathbf{C}_\tau$ and of $\mathbf{H}\mathbf{H}^T = \mathbf{A}(\mathbf{I} - \mathbf{M})^{-1}(\mathbf{I} - \mathbf{M})^{-T}\mathbf{A}^T$. Third, exploiting column-space arguments and invertibility of $\mathbf{I} - \mathbf{M}$, we show that any two observationally equivalent models must satisfy $\hat{\mathbf{A}} = \mathbf{A}\mathbf{S}$, $(\mathbf{I} - \hat{\mathbf{M}}) = \mathbf{U}^T(\mathbf{I} - \mathbf{M})\mathbf{S}$, and $\hat{\mathbf{B}}_\tau = \mathbf{U}^T\mathbf{B}_\tau\mathbf{S}$, where $\mathbf{S}$ is invertible and $\mathbf{U}$ is orthogonal. Finally, non-Gaussianity of the innovations implies that $\mathbf{U}$ must be a signed permutation matrix, yielding a precise characterization of the remaining indeterminacies.

**Discussion** This result extends [58] to temporal processes with instantaneous relations. When $\mathbf{M} \neq \mathbf{0}$, the indeterminacy of $\mathbf{A}$ increases due to the additional mixing introduced by $\mathbf{M}$, changing the ambiguity from an orthogonal transformation to a general invertible one. Consequently, both $\mathbf{I} - \mathbf{M}$ and $\mathbf{B}_\tau$ inherit right-multiplicative indeterminacies. Without further structural assumptions on $\mathbf{M}$ or $\mathbf{B}_\tau$, component-wise identifiability is impossible. However, since the indeterminacies are precisely characterized, additional structural constraints can be imposed to further improve identifiability, as shown in the following corollaries.

**Corollary 1** (Component-wise Identifiability). *Suppose the estimated model* $(\hat{\mathbf{A}}, \{\hat{\mathbf{B}}_\tau\}_{\tau=1}^{L}, \hat{\mathbf{M}}, p_{\hat{\boldsymbol{\epsilon}}})$ *and the true model* $(\mathbf{A}, \{\mathbf{B}_\tau\}_{\tau=1}^{L}, \mathbf{M}, p_{\boldsymbol{\epsilon}})$ *both generate* $\mathbf{x}_t$ *according to Eq. 3 and are observationally equivalent on the autocovariance matrices* $\mathbf{R}_{\mathbf{x}}(k)$ *for* $k = 0, 1, \ldots, L$. *If conditions A1–A4 of Theorem 1 hold and, in addition, the following assumption on the instantaneous relations* $\mathbf{M}$ *is satisfied, then the latent variables* $\mathbf{z}_t$ *are identifiable up to permutation and scaling, i.e.,* $\hat{\mathbf{A}} = \mathbf{A}\tilde{\mathbf{P}}\mathbf{D}$, *where* $\tilde{\mathbf{P}}$ *is a permutation matrix and* $\mathbf{D}$ *is a diagonal matrix with nonzero entries.*

- *A5 (Empty or unique column supports of* $\mathbf{M}$*).* *Each column of* $\mathbf{M}$ *either has empty support or contains at least one index that does not appear in the support of any other column. The model is estimated under a sparsity constraint on* $\mathbf{M}$*.*

*Proof.* From Theorem 1, we have $\hat{\mathbf{A}} = \mathbf{A}\mathbf{S}$ and $(\mathbf{I} - \hat{\mathbf{M}}) = \mathbf{P}(\mathbf{I} - \mathbf{M})\mathbf{S}$. Since $\mathbf{P}$ only permutes rows, it does not affect column-support arguments. Under Assumption A5, any column of $\mathbf{M}$ has a unique nonzero support element, which prevents cancellation across columns. If a column $\mathbf{S}_{:,l}$ contained more than one nonzero entry, then $\mathbf{P}(\mathbf{I} - \mathbf{M})\mathbf{S}_{:,l}$ would necessarily have a strictly larger support than the corresponding column of $\mathbf{I} - \mathbf{M}$, violating the assumed sparsity. Since $\mathbf{M}$ has zero diagonal (by the SEM assumption), the same argument applies to $\hat{\mathbf{M}}$. Therefore, sparsity-enforced estimation excludes such $\mathbf{S}$, implying that $\mathbf{S}$ must be a product of a permutation and a diagonal scaling matrix. Hence, $\mathbf{z}_t$ is component-wise identifiable. $\square$

**Corollary 2** (Subspace Identifiability). *Suppose the estimated model* $(\hat{\mathbf{A}}, \{\hat{\mathbf{B}}_\tau\}_{\tau=1}^{L}, \hat{\mathbf{M}}, p_{\hat{\boldsymbol{\epsilon}}})$ *and the true model* $(\mathbf{A}, \{\mathbf{B}_\tau\}_{\tau=1}^{L}, \mathbf{M}, p_{\boldsymbol{\epsilon}})$ *both generate* $\mathbf{x}_t$ *according to Eq. 3 and are observationally equivalent on the autocovariance matrices* $\mathbf{R}_{\mathbf{x}}(k)$ *for* $k = 0, 1, \ldots, L$. *If conditions A1–A4 of Theorem 1 hold and A6 is satisfied, then the latent variables* $\mathbf{z}_t$ *are subspace identifiable, i.e.,* $\hat{\mathbf{A}} = \mathbf{A}\tilde{\mathbf{S}}$, *where each column of* $\tilde{\mathbf{S}}$ *has nonzero entries only within a single subspace.*

- *A6 (Subspace structure of* $\mathbf{M}$*).* *The instantaneous relation matrix* $\mathbf{M}$ *admits a partition of* $[n]$ *into* $K$ *disjoint subsets such that* $\mathbf{M}_{ij} \neq 0$ *only if* $i$ *and* $j$ *belong to the same subset. The model is estimated with a sparsity constraint on* $\mathbf{M}$*.*

*Proof.* From Theorem 1, we have $\hat{\mathbf{A}} = \mathbf{A}\mathbf{S}$ and $(\mathbf{I} - \hat{\mathbf{M}}) = \mathbf{P}(\mathbf{I} - \mathbf{M})\mathbf{S}$. Assumption A6 implies that columns of $\mathbf{M}$ corresponding to different subspaces have disjoint supports. Since $\mathbf{M}$ has zero

diagonal, the same property holds for $\mathbf{I} - \mathbf{M}$. If any column $\mathbf{S}_{:,l}$ mixed components from different subspaces, then the corresponding column of $(\mathbf{I} - \mathbf{M})\mathbf{S}$ would necessarily contain nonzero entries from multiple subspaces, contradicting Assumption A6. Therefore, each column of $\mathbf{S}$ can only combine latent variables within the same subspace, implying subspace identifiability of $\mathbf{z}_t$. $\quad\square$

**Discussion** Corollary 1 strengthens Theorem 1 by imposing a strong structural assumption (A5) on $\mathbf{M}$, yielding component-wise identifiability. Corollary 2 relaxes this requirement by allowing instantaneous relations within latent subspaces, leading to subspace identifiability. Both results rely on sparsity constraints imposed during estimation. Moreover, each corollary admits a natural analogue for the lagged matrices $\mathbf{B}_\tau$. In particular, the counterpart of Corollary 1 requires nonzero diagonal entries in $\mathbf{B}_\tau$, while the analogue of Corollary 2 assumes a matching subspace structure. Furthermore, if $\mathbf{M}$ is forced to be strictly triangular, the right permutation indeterminacy must match the left one, i.e., $\tilde{\mathbf{P}} = \mathbf{P}$, otherwise the strictly triangular structure will be destroyed. In practice, we therefore enforce sparsity on both instantaneous and lagged relations and strictly lower triangular structure on the instantaneous adjacency. Empirically, the recovered structures from real data approximately satisfy the assumed conditions, while synthetic experiments explicitly enforce them to validate identifiability.

## 4 Implementation

Based on the data generation process in Eq. (3) together with the identifiability result presented in the previous section, we derive the following estimation process based on the standard sparse autoencoder. Illustrated in Figure 2, the whole estimation process can be partitioned into three parts, namely (1) observation reconstruction, (2) independent noise estimation, and (3) sparsity regularization.

### 4.1 Observation Reconstruction

First, we use a linear autoencoder to enforce the invertible linear transformation between observations $\mathbf{x}_t$ and latent variables $\mathbf{z}_t$, and the reconstruction loss $\mathcal{L}_r$ is defined as

$$\mathcal{L}_r = \mathbb{E}_{\mathbf{x}_{1:T}} \left[ \sum_{t=1}^T (\mathbf{x}_t - \hat{\mathbf{x}}_t)^2 \right], \qquad (4)$$

where the reconstructed observation is calculated via a linear encoder and decoder:

$$\hat{\mathbf{x}}_t = \texttt{Decoder}(\hat{\mathbf{z}}_t) \quad \text{and} \quad \hat{\mathbf{z}}_t = \texttt{Encoder}(\mathbf{x}_t). \qquad (5)$$

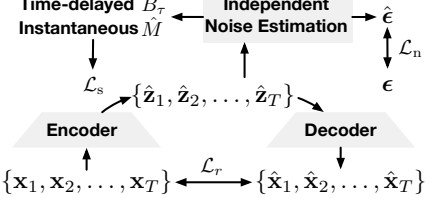

Figure 2: Illustration of estimation process. $\hat{\mathbf{B}}_\tau$ represents the learned time-delayed causal relation and $\hat{\mathbf{M}}$ is the instantaneous causal relation.

### 4.2 Independent Noise Estimation

In prior works [56, 49, 27], this terms refers to the independent prior estimation, in which they essentially utilize the independence of noise to enforce the independence of latent variables $z_{t,i}$, conditioning on parent $\text{Pa}(z_{t,i})$. In our case, since the whole process is linear, we can directly estimate and enforce the independent noise condition by learning a residual network by reversing the data generation process described in Eq. 3:

$$\hat{\boldsymbol{\epsilon}}_t = \hat{\mathbf{z}}_t - \hat{\mathbf{M}} \hat{\mathbf{z}}_t - \sum_{\tau > 0} \hat{\mathbf{B}}_\tau \hat{\mathbf{z}}_{t-\tau}, \qquad (6)$$

where the estimated latent variables are given by Eq. (5). Following the prior works, to enforce the independence of noise terms, we model the noise distribution $p(\hat{\epsilon}_{t,i})$ with isomorphic Laplacian[†] distribution, and we minimize its KL-divergence with the estimated noise term.

$$\mathcal{L}_n = \mathbb{E}_{\hat{\boldsymbol{\epsilon}}_t} \left[ ||\hat{\boldsymbol{\epsilon}}_t||_1 \right]. \qquad (7)$$

---

[†]In prior works, the distribution is Gaussian, however, we can see that in linear case as is well discussed in linear ICA literature, the density function of an isomorphic Gaussian distribution is rotation invariant, hence we utilize the Laplacian distribution in our estimation.

### 4.3 Sparsity Regularization

Without any further constraint, the noise estimation module may bring redundant causal edges from $\hat{\mathbf{z}}_{t-1}, \hat{\mathbf{z}}_{t,[m]\setminus i}$ to $\hat{z}_{t,i}$, leading to the incorrect estimation. As mentioned in Sec. 4.2, $\{\mathbf{B}_\tau\}$ and $M$ intuitively denote the time-delayed and instantaneous causal structures, since they describe how the $\hat{\mathbf{z}}_{t-1}, \hat{\mathbf{z}}_{t,[m]\setminus i}$ contribute to $\hat{z}_{t,i}$, which motivate us to remove these redundant causal edges with a sparsity regularization term $\mathcal{L}_s$ by using the L1 penalty on $\{\hat{\mathbf{B}}_\tau\}$ and $\hat{\mathbf{M}}$. Formally, we have

$$\mathcal{L}_s = \left( \sum_\tau ||\hat{\mathbf{B}}_\tau||_1 \right) + ||\hat{\mathbf{M}}||_1, \tag{8}$$

where $|| * ||_1$ denotes the L1 Norm of a matrix. And we restrict the $\mathbf{M}$ to be strictly lower triangular to match the permutation indeterminacy on both sides of $\mathbf{M}$ and $\mathbf{B}_\tau$. Finally, the total loss of the model can be formalized as:

$$\mathcal{L}_{total} = \mathcal{L}_r + \alpha \mathcal{L}_n + \beta \mathcal{L}_s, \tag{9}$$

where $\alpha, \beta$ denote the hyper-parameters.

## 5 Experiments

Our experimental evaluation addresses five key claims regarding our proposed method: (1) our estimation approach aligns with identifiability theory, accurately recovering latent structures; (2) existing CRL methods fail to handle high-dimensional data at scale; (3) our method is able to recover target relations between concepts from semi-synthetic data; (4) compared with common SAEs, our proposal achieves satisfactory results on quantitative evaluation metrics (SAEBench [24]); and (5) our method effectively learns both time-delayed and instantaneous causal relations among concepts elicited from LLM activations.

### 5.1 Synthetic Data Experiments

First, using synthetic data, we demonstrate that our method can recover both the latent variables $\mathbf{z}_t$ and the causal structure including time-delayed relations $\mathbf{B}_\tau$ and instantaneous relations $\mathbf{M}$.

**Identifiability Verification**   To establish the effectiveness of our approach, we generate simulated time series data with a latent causal process as introduced in Eq. (3). We apply our method to single time lag synthetic data generated with a randomly initialized matrix $A$ and fixed transition matrices $\mathbf{B}$ and $\mathbf{M}$ visualized in Figure 3a and 3c. Further details can be found in Appendix A.3.

We visualize the estimated parameters by plotting the recovered matrices $\hat{\mathbf{B}}$ and $\hat{\mathbf{M}}$ alongside the correlation coefficient matrix used for calculating the mean correlation coefficient (MCC) score.

As shown in Figure 3, comparing with the ground truth transition matrices $\mathbf{B}$ and $\mathbf{M}$, we observe that both time-delayed and instantaneous causal relations have been precisely recovered. Furthermore, Figure 3e demonstrates that the latent variables $\mathbf{z}_t$ are also accurately recovered, confirming the identifiability properties of our method.

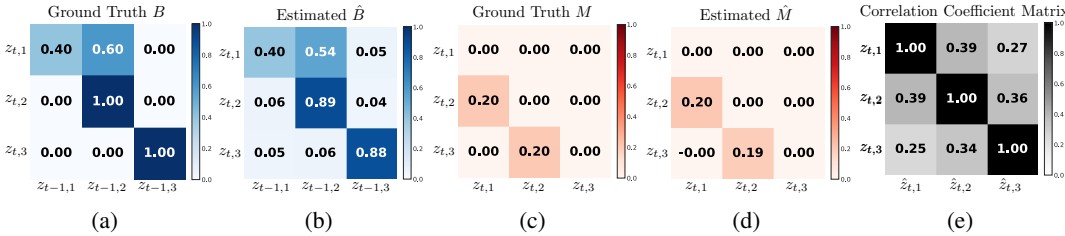

Figure 3: Visualization of recovered causal graphs of latent variables. (a) and (b) show the ground truth and estimated time-delayed matrix, respectively. (c) and (d) show the ground truth and the estimated instantaneous causal relations, respectively. (e) displays the correlation between the ground truth and recovered latent variables.

Second, we scale the synthetic experiments to dimensions matching LLM activations, illustrating why existing CRL methods fail in these high-dimensional settings.

**Challenges on Scaling to Large Language Model Activation Dimensions** Before presenting results on expanded synthetic data, we investigate the computation bottleneck: Jacobian calculation and explain why existing CRL methods do not extend efficiently to high-dimensional settings, thereby further motivating our use of a linear dynamical model.

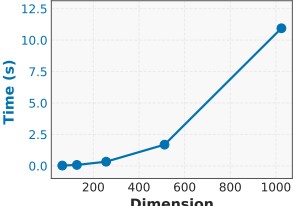 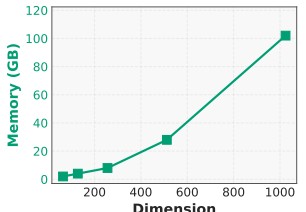 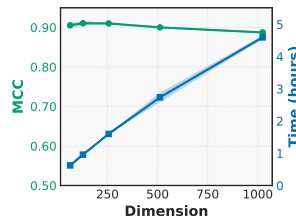

Figure 5: Computation time and memory usage for a single-step Jacobian as a function of input dimensionality. Both metrics grow superlinearly and exceed the capacity of modern GPUs when the input dimension is greater than 1000.

Figure 6: MCC and total compute time in hours required to train the linear model as a function of input dimension.

**Computation cost of Jacobian evaluation.** We take IDOL [27] as a representative method and measure both the wall-clock time and memory requirements for computing the Jacobian in prior network. Figure 5 demonstrates that both time and memory complexity grow polynomial with dimensionality. At dimensions of several thousand—which are common for Large Language Model activations—a single Jacobian evaluation will require approximately ten seconds on a modern GPU, such computation cannot fit into current generation hardware infrastructure. Since CRL training invokes this operation millions of times during the training process, the cumulative computational cost becomes prohibitive. As other CRL algorithms involve comparable Jacobian computations or more complex algorithms, this fundamental limitation applies broadly across the field.

**Advantages of linear models.** When a linear model provides an adequate approximation of the transition dynamics of the hidden concepts, the Jacobian calculation can be derived directly from model parameters such as $\mathbf{B}$ and $\mathbf{M}$, which significantly reduces the computational burden. Furthermore, such a linear model can scale efficiently with current-generation compute resources. To support this claim, we conducted a scaling experiment using the linear model on synthetic data with dimensionalities ranging from 128 to 1024. In each setting, the model was trained on 50 million samples, simulating the typical training load in real LLM SAEs with 50 million tokens. As shown in Figure 6, the proposed method scales to substantially higher dimensions while maintaining a high MCC of approximately 0.9. Additionally, it remains computationally efficient, with total computation time scaling linearly. In contrast, IDOL [27] exhausts memory when the dimensionality exceeds 200, and iCITRIS [31] fails to scale beyond 16 dimensions.

## 5.2 Semi-synthetic Experiments

Given the previous experiments on synthetic data, our proposal has been shown to recover ground-truth relationships even when the hidden dimensionality reaches one thousand, which would be challenging for existing non-parametric CRL approaches. We now proceed to evaluate real-world LLM activations, beginning with investigation (3). The experimental settings are briefly introduced below. Full details can be found in Appendix A.4.1.

Table 1: Relation recovery scores (↑) for concept–relation extraction on semi-synthetic data.

| Method | Legal | XML | Email |
|---|---|---|---|
| SAE+regression | 0.54 | 0.94 | 0.74 |
| Ours | **19.95** | **8.63** | **2.66** |

**Data preparation:** We first examine three types of text, each exhibiting an obvious syntactic pattern. For example, in legal text, sequences often begin with "APPEALS" and end with "AFFIRMED". For illustration, we focus on the legal text contrastive corpus group. We constructed two contrastive subsets from the `Pile` dataset [17]: one containing legal documents with highly structured syntax and

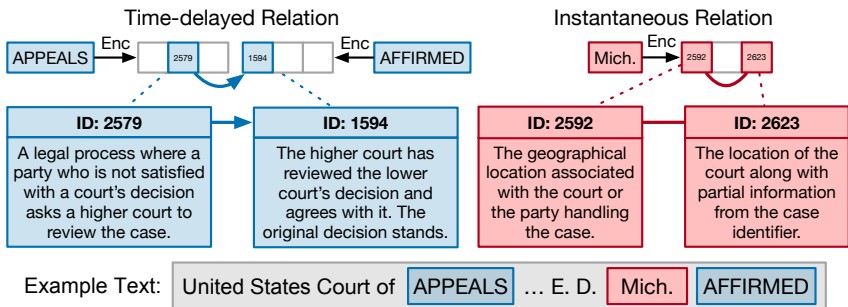

Figure 7: Case study illustrates two relation types identified in a United States legal text. The blue elements show a time-delayed relation: the term "appeals" is typically followed by "affirmed" when a higher court confirms the lower court decision. The red elements show an instantaneous relation: two geographical location concepts (2592, 2623), are activated together in the same passage.

stable temporal patterns, and the other containing unstructured non-legal text. We hypothesized that only texts containing these structured relations would yield meaningful temporal concept patterns and tested whether the model could recover them. **Baseline:** Since no directly applicable baseline exists, we used the standard SAEs trained above. As SAEs cannot capture concept-to-concept relations, we fitted a regression model to estimate temporal relation matrices $\tilde{\mathbf{B}}$ via $\mathbf{z}_t = \sum_\tau \tilde{\mathbf{B}}_\tau \mathbf{z}_{t-\tau}$. **Evaluation:** We compute the concept recovery score by first identifying the top concept pair $(i, j)$ in legal contexts (ensuring that the two corresponding concepts do not fire in the non-legal text), then taking the corresponding coefficient $\mathbf{B}_{i,j}$ and normalizing it by the standard deviation $\sigma(B)$. The ratio $\frac{\mathbf{B}_{i,j}}{\sigma(B)}$ serves as a relation recovery score, indicating how strongly the relation stands out from noise. As shown in Table 1, our method achieves a significantly higher score, demonstrating successful recovery of the relation. Finally, as concept recovery is already achievable by standard SAEs, we additionally conducted steering vector semi-synthetic experiments to verify that our proposal can also recover concepts, following the approach of [23]. Further details are provided in Appendix A.4.2.

## 5.3 Real LLM Activation Analysis

**Experiment Setup** We train our linear model on activations from the pretrained LLM `pythia-160m-deduped` [5], using `SAELens` [6] and `dictionary-learning` [35] for activation extraction. The model is trained on 50 million tokens from the `Pile` dataset [17]. To capture time-delayed influences, we set $\tau \leq 20$ in Eq. 3 and aggregate the $\mathbf{B}_\tau$ matrices using max-pooling, preserving any causal link that appears at any time step. We evaluate three feature dimensionalities: 768 (matching the LLM's hidden size and aligned with Section 3), 3072, and 6144—the latter two following common SAE training setups. Unless specified, main text examples use 3072-dimensional features with $\tau = 20$. Full training details and additional results (sensitivity and ablation studies are included) are in Appendix A.5.

Table 2: Comparison of our method against ReLU and TopK SAEs on SAEBench metrics.

| Model | Recon. Loss ↓ | Sparse Prob. ↑ | Absorp. ↓ | Autointerp ↑ |
|---|---|---|---|---|
| ReLU SAE | 0.0110 | 0.6555 | 0.0141 | 0.6791 |
| TopK SAE | **0.0097** | **0.7141** | 0.0280 | 0.6822 |
| Ours | 0.0108 | 0.6736 | **0.0139** | **0.6883** |

**Quantitative Evaluation on SAEBench** Before we dive into the details of concept relation recovery, we first present a quantitative comparison between our method and existing SAE approaches. Since our main contribution lies in recovering temporal and instantaneous concept-to-concept relations, which are not reflected in current SAE benchmarks, we expect our model to perform on par with established SAEs on SAEBench tasks. This expectation is confirmed by the results in Table 2. Additional experiment results on larger latent size and model size can be found in Appendix A.5.5.

**Case Studies** We start with an illustrative case in Figure 7, demonstrating how our model uncovers interpretable concept features with both time-delayed and instantaneous causal relationships from real-world LLM activations. This example provides an integrated view of how concepts are structured

Table 3: Representative time-delayed and instantaneous concept relations discovered.

| ID | From | ID | To | Coeff. |
|---|---|---|---|---|
| Time-delayed relations | | | | |
| 1657 | Keywords for formal and official content (e.g., *senate*, *state*, *military*) | 1664 | Verbs for official/formal usage (e.g., *deny*, *press*, *order*, *sign*) | 0.99 |
| 2641 | Adjectives of nationality (e.g. *Japanese*, *Italians*) | 2674 | Nouns that follow nationality (e.g. *brands*, *name*) | 0.81 |
| 1657 | Keywords for formal and official content (e.g., *senate*, *state*, *military*) | 1124 | Objective adjectives in formal usage (e.g., *fast*, *continuous*, *incomplete*) | 0.74 |
| Instantaneous relations | | | | |
| 2208 | Partial appellate citation with volume number | 227 | Partial appellate citation with volume number and series index | 0.23 |
| 1714 | Coding-format signals (e.g. *localization tags*, *HTML* tags) | 80 | Coding-format content (e.g. *key–value pairs*, *HTML elements*) | 0.16 |
| 1582 | Month (e.g. *March*) | 363 | Full date (e.g. *March 23, 2000*) | 0.02 |

over time and interact within a single time step. Note that feature interpretations may vary beyond this case; additional examples and discussions appear in Appendix A.5.

Figure 7 highlights two key observations: First, a time-delayed causal link between concepts related to "appeals" and "affirmed" in legal texts (features 2579 and 1594), capturing how the model reflects the procedural flow of legal judgments. Second, an instantaneous relation between two geographical location concepts (features 2592 and 2623) that are activated together in legal passages, suggesting that the model represents related spatial information simultaneously rather than sequentially. This example effectively demonstrates that both time-delayed and instantaneous relations exist among concept features, and that these are interpretable alongside the semantic meanings of the features—both of which are essential for LLM interpretability. To further demonstrate our model's capacity to uncover both types of causal relationships, we present a broader set of examples in Table 3, which showcases representative cases of both time-delayed and instantaneous interactions among concept features.

**Time-delayed causal relations.** We first observe a strong causal relation from nationality adjectives (feature 2641, "Japanese," "Italians") to the nouns they commonly modify (feature 2674, "brands," "literature"), with a coefficient of 0.81. Moreover, the coefficients across the 20-token temporal window (i.e., different $\mathbf{B}_\tau$) contribute consistently to the aggregated score. This suggests that such temporal relations can occur across a broad and uncertain time span, aligning with the semantic dynamics of real-world text generation. In formal contexts, official content words (feature 1657, "senate," "judge") influence both formal verbs (feature 1664, "deny," "order") with a coefficient of 0.99, and objective adjectives (feature 1124, "fast," "continuous") with a coefficient of 0.74. These relationships reflect how formal language constrains both action and descriptive style over time.

**Instantaneous causal relations.** Table 3 presents three distinct categories of instantaneous relations. First, we observe a relationship between two partially overlapping appellate citation features—feature 2208 (volume numbers only) and feature 227 (volume number and series index)—with a normalized coefficient of 0.23. This illustrates how the model captures structured elements that commonly co-occur in legal documents, forming a cohesive representational unit. Second, we find that coding-format signals (feature 1714, e.g., localization tags, HTML tags) have an instantaneous causal relationship with coding-format content (feature 80, e.g., key-value pairs, HTML elements), with a coefficient of 0.16. This reveals how the model processes structured syntax and its associated content as co-occurring elements. Finally, our method identifies a clear relationship between two features that both represent dates: feature 1582 (month only) and feature 363 (full date), suggesting complementary representations within the model's internal structure.

These findings demonstrate our method's ability to uncover both temporal and instantaneous causal structures in the concept space of LLM activations, offering insights into how models organize and process information. The identified relationships align with expected patterns in natural language across domains such as legal texts, temporal expressions, and structured formats, validating the effectiveness of our approach for analyzing information flow in large language models.

# 6 Related Work

**LLM Interpretability**   Understanding the internal representations of LLMs remains challenging despite significant progress [26]. Interpretability research on LLMs has explored multiple directions including: probing for linguistic knowledge [19], evaluating interpretability methods through controlled experiments [22], benchmarking SAEs' capacity to disentangle factual knowledge [12], and developing ground-truth evaluation frameworks [53, 25]. Recent work suggests that LLM representations may follow a linear organization [43], though this hypothesis has been challenged [16]. Our approach extends these efforts by focusing specifically on causal interpretability of temporal relationships in LLMs, providing a principled framework for understanding how information flows through model representations during sequential text generation. Additionally sparse autoencoders (SAEs) decompose neural activations into interpretable features [14, 7, 52]. Initial work demonstrated that SAEs can recover meaningful features from language model activations [41], leading to numerous architectural innovations including alternative activation functions [50, 46], training optimizations [8, 20], and efficient dictionary allocation mechanisms [4, 39]. Recent work has successfully scaled SAEs to larger models [18, 29, 2], enabling automated interpretation of millions of features [44]. Despite these advances, most SAE approaches treat features as isolated units without modeling temporal relationships [11, 9], lack explicit causal structure [34], and offer no identifiability guarantees [54, 33]—limitations our work directly addresses.

**Feature-based Causal Circuits**   Recent methods like Sparse Feature Circuits [36] and attribution graphs [1, 30] identify causal subnetworks explaining model behavior. These build on earlier circuit analysis methods exploring component functionalities in vision and language models [42, 10, 15]. Targeted interventional studies have revealed specific functional circuits, such as indirect object identification [55] and factual associations [37]. While these methods enable mechanistic understanding of model computations [19, 40], they primarily rely on correlational measures rather than structured causal inference [16, 43]. But they focus on stationary relationships [21] instead of modeling evolving token-to-token dependencies critical for understanding sequential reasoning.

**Causal Representation Learning**   Causal representation learning provides identifiability guarantees for latent variables [56, 27, 47]. Temporal extensions model dynamics in sequential data [57, 31, 49], with recent advances addressing non-stationarity [48, 13] and instantaneous effects [32]. Multiple distribution methods [59, 38] can recover causal structure under specific interventions or group structures. These approaches provide theoretical foundations for disentangling latent variables and identifying causal graphs [54]. However, existing CRL algorithms cannot scale to LLM dimensions due to computational bottlenecks in calculating Jacobians. Our linearized formulation maintains identifiability guarantees while enabling application to high-dimensional LLM representations—bridging theoretical CRL advances with practical LLM interpretability challenges.

# 7 Conclusion

We introduced a causal representation learning framework for LLMs that jointly models time-delayed relationships and instantaneous constraints between latent concepts. Our approach provides theoretical identifiability guarantees while solving the scalability limitations of existing CRL methods through a computationally efficient linear formulation. Synthetic experiments validated our method's ability to recover latent causal structures from toy scale to real LLM scales. When applied to real LLM activations, our approach uncovered interpretable semantic patterns, revealing information flow pathways during text generation. Future work could leverage these causal structures for targeted alignment interventions, explore cross-layer concept transformations, and integrate with mechanistic interpretability techniques.

# 8 Acknowledgment

The authors would like to thank the anonymous reviewers for helpful comments and suggestions during the reviewing process. The authors would also like to acknowledge the support from NSF Award No. 2229881, AI Institute for Societal Decision Making (AI-SDM), the National Institutes of Health (NIH) under Contract R01HL159805, and grants from Quris AI, Florin Court Capital, and MBZUAI-WIS Joint Program, and the Al Deira Causal Education project.

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

# A   Technical Appendices and Supplementary Material

## Contents

## A.1   Discussion of Non-parametric Proof in Linear Case

In this discussion, we will replay the proof in [27] with assuming linearity to show the proof does not work in linear case. The proof sketch of the original non-parametric proof adapted in linear case is shown in below. And we will show it is impossible to have full-rank coefficient matrix $\mathbf{C}_{\mathbf{z}_t}$, which leads to the failure of the this derivation.

**Non-parametric Proof Sketch in Linear Case**   If the learned model $(\hat{\mathbf{A}}, \hat{\mathbf{B}}, \hat{\mathbf{M}}, \hat{p}_{\epsilon})$ is observationally equivalent to the true model, then there exists an invertible linear map $\mathbf{H} = \mathbf{A}\hat{\mathbf{A}}^{-1}$ such that $p(\hat{\mathbf{z}}_t|\hat{\mathbf{z}}_{t-1}) = p(\mathbf{z}_t|\mathbf{z}_{t-1})|\mathbf{H}|$. For nonadjacent $(\hat{z}_{t,k}, \hat{z}_{t,l})$ in $\mathcal{M}_{\hat{\mathbf{z}}_t}$, conditional independence implies $\frac{\log p(\hat{\mathbf{z}}_t|\hat{\mathbf{z}}_{t-1})}{\partial^2 \hat{z}_{t,k}\hat{z}_{t,l}} = 0$. Expanding this derivative via the chain rule yields a linear system over terms $\frac{\partial^2 z_{t,i}}{\partial \hat{z}_{t,k}\partial \hat{z}_{t,l}}, \frac{\partial z_{t,i}}{\partial \hat{z}_{t,k}}, \frac{\partial z_{t,i}}{\partial \hat{z}_{t,l}}$, and $\frac{\partial z_{t,i}}{\partial \hat{z}_{t,k}}, \frac{\partial z_{t,j}}{\partial \hat{z}_{t,l}}$. If the corresponding coefficient matrix $\mathbf{C}_{\mathbf{z}_t}$ is full rank, then all these terms vanish. Hence each $z_{t,i}$ depends on at most one $\hat{z}_{t,k}$, and nonadjacent estimated variables cannot involve adjacent true ones.

Suppose that the estimated model $(\hat{\mathbf{A}}, \hat{\mathbf{B}}, \hat{\mathbf{M}}, p_{\hat{\epsilon}})$ is observationally equivalent to the true model $(\mathbf{A}, \mathbf{B}, \mathbf{M}, p_{\epsilon})$, then we have $p_{\hat{\mathbf{A}}\hat{\mathbf{z}}_t}(\mathbf{x}_t) = p_{\mathbf{A}\mathbf{z}_t}(\mathbf{x}_t)$. Applying the change of variable, we have

$$p_{\hat{\mathbf{A}}\hat{\mathbf{z}}_t}(\mathbf{x}_t) = p(\hat{\mathbf{z}}_t)|\hat{\mathbf{A}}| = p(\mathbf{z}_t)|\mathbf{A}| = p_{\mathbf{A}\mathbf{z}_t}(\mathbf{x}_t) \Leftrightarrow p(\hat{\mathbf{z}}_t) = p(\mathbf{z}_t)\frac{|\mathbf{A}|}{|\hat{\mathbf{A}}|} \Leftrightarrow p(\hat{\mathbf{z}}_t) = p(\mathbf{z}_t)|\mathbf{A}\hat{\mathbf{A}}^{-1}|,$$

given the invertibility of the linear mixing function $\hat{\mathbf{A}}$. For the sake of simplicity, let $\mathbf{H} = \mathbf{A}\hat{\mathbf{A}}^{-1}$ and $p(\hat{\mathbf{z}}_t) = p(\mathbf{z}_t)|\mathbf{H}|$. Similarly, we have $p(\hat{\mathbf{z}}_{t+1}) = p(\mathbf{z}_{t+1})|\mathbf{H}|$.

Let $\mathcal{M}_{\hat{\mathbf{z}}_t}$ be the Markov network of $\hat{\mathbf{z}}_t$. Then, for any pair of variables $\hat{z}_{t,k}$ and $\hat{z}_{t,l}$ that are not connected in $\mathcal{M}_{\hat{\mathbf{z}}_t}$, i.e., $(\hat{z}_{t,k}, \hat{z}_{t,l}) \notin \mathcal{E}_{\mathcal{M}_{\hat{\mathbf{z}}_t}}$, we have conditional independence on the graph due to the temporal relations between $\mathbf{z}_{t-1}$ and $\mathbf{z}_t$ and the property of Markove networks:

$$\hat{z}_{t,i} \perp \hat{z}_{t,j} | \hat{\mathbf{z}}_{t-1} \cup (\hat{\mathbf{z}}_t \setminus \{\hat{z}_{t,i}, \hat{z}_{t,j}\}).$$

We then make use of the fact that if two variables are independent conditioning on all the rest of the variables, then the cross second-order derivative of the logarithmic of the joint density w.r.t. the two variables. In our case, it is shown as:

$$\frac{\partial^2 \log p(\hat{\mathbf{z}}_t, \hat{\mathbf{z}}_{t-1})}{\partial \hat{z}_{t,i} \partial \hat{z}_{t,j}} = 0.$$

Since $\log p(\hat{\mathbf{z}}_t, \hat{\mathbf{z}}_{t-1}) = \log p(\hat{\mathbf{z}}_t | \hat{\mathbf{z}}_{t-1}) + \log p(\hat{\mathbf{z}}_{t-1})$, where $\hat{\mathbf{z}}_t$ has no functional dependence on $z_{t,k}$ or $z_{t,l}$, and therefore,

$$\frac{\partial^2 \log p(\hat{\mathbf{z}}_t | \hat{\mathbf{z}}_{t-1})}{\partial \hat{z}_{t,i} \partial \hat{z}_{t,j}} = 0. \tag{10}$$

Because $\begin{pmatrix} \hat{\mathbf{z}}_t \\ \hat{\mathbf{z}}_{t-1} \end{pmatrix} = \begin{pmatrix} \mathbf{H} & * \\ \mathbf{0} & \mathbf{H} \end{pmatrix} \begin{pmatrix} \mathbf{z}_t \\ \mathbf{z}_{t-1} \end{pmatrix}$, $p(\hat{\mathbf{z}}_t, \hat{\mathbf{z}}_{t-1}) = p(\mathbf{z}_t, \mathbf{z}_{t-1})|\mathbf{H}^2|$. Then, we divide both sides by $p(\hat{\mathbf{z}}_{t-1})$, we have:

$$p(\hat{\mathbf{z}}_t | \hat{\mathbf{z}}_{t-1}) = \frac{p(\mathbf{z}_t, \mathbf{z}_{t-1})|\mathbf{H}^2|}{p(\mathbf{z}_{t-1})|\mathbf{H}|} = p(\mathbf{z}_t | \mathbf{z}_{t-1})|\mathbf{H}|. \tag{11}$$

Its partial derivative w.r.t. $\hat{z}_{t,k}$ is

$$\frac{\partial \log p(\hat{\mathbf{z}}_t | \hat{\mathbf{z}}_{t-1})}{\partial \hat{z}_{t,k}} = \frac{\partial \log p(\mathbf{z}_t | \mathbf{z}_{t-1})}{\partial \hat{z}_{t,k}} + \frac{\partial \log |\mathbf{H}|}{\partial \hat{z}_{t,k}} \tag{12}$$

$$= \sum_{i \in [n]} \frac{\partial \log p(\hat{\mathbf{z}}_t | \hat{\mathbf{z}}_{t-1})}{\partial z_{t,i}} \frac{\partial z_{t,i}}{\partial \hat{z}_{t,k}} + \frac{\partial \log |\mathbf{H}|}{\partial \hat{z}_{t,k}} \tag{13}$$

Then, its cross second order derivative w.r.t. $\hat{z}_{t,k}$ and $\hat{z}_{t,l}$ is

$$\frac{\partial^2 \log p(\hat{\mathbf{z}}_t | \hat{\mathbf{z}}_{t-1})}{\partial \hat{z}_{t,k} \partial \hat{z}_{t,l}} = \frac{\partial^2 \log p(\mathbf{z}_t | \mathbf{z}_{t-1})}{\partial \hat{z}_{t,k} \partial \hat{z}_{t,l}} + \frac{\partial^2 \log |\mathbf{H}|}{\partial \hat{z}_{t,k} \partial \hat{z}_{t,l}} \tag{14}$$

$$= \sum_{i \in [n]} \frac{\partial \log p(\mathbf{z}_t | \mathbf{z}_{t-1})}{\partial z_{t,i}} \frac{\partial^2 z_{t,i}}{\partial \hat{z}_{t,k} \hat{z}_{t,l}} + \tag{15}$$

$$\sum_{i \in [n]} \sum_{j \in [n]} \frac{\partial^2 \log p(\mathbf{z}_t | \mathbf{z}_{t-1})}{\partial z_{t,i} \partial z_{t,j}} \frac{\partial^2 z_{t,i}}{\partial \hat{z}_{t,k}} \frac{\partial^2 z_{t,j}}{\partial \hat{z}_{t,l}} + \frac{\partial^2 \log |\mathbf{H}|}{\partial \hat{z}_{t,k} \partial \hat{z}_{t,l}} \tag{16}$$

$$= \sum_{i \in [n]} \frac{\partial \log p(\mathbf{z}_t | \mathbf{z}_{t-1})}{\partial z_{t,i}} \frac{\partial^2 z_{t,i}}{\partial \hat{z}_{t,k} \hat{z}_{t,l}} + \sum_{i \in [n]} \frac{\partial^2 \log p(\mathbf{z}_t | \mathbf{z}_{t-1})}{\partial z_{t,i}^2} \frac{\partial z_{t,i}}{\partial \hat{z}_{t,k}} \frac{\partial z_{t,i}}{\partial \hat{z}_{t,l}} +$$

$$\sum_{(z_{t,i}, z_{t,j}) \in \mathcal{M}_{\mathbf{z}_t}} \frac{\partial^2 \log p(\mathbf{z}_t | \mathbf{z}_{t-1})}{\partial z_{t,i} \partial z_{t,j}} \frac{\partial z_{t,i}}{\partial \hat{z}_{t,k}} \frac{\partial z_{t,j}}{\partial \hat{z}_{t,l}} + \frac{\partial^2 \log |\mathbf{H}|}{\partial \hat{z}_{t,k} \partial \hat{z}_{t,l}} \tag{17}$$

$$= \sum_{i \in [n]} \frac{\partial^2 \log p(\mathbf{z}_t | \mathbf{z}_{t-1})}{\partial z_{t,i}^2} \frac{\partial z_{t,i}}{\partial \hat{z}_{t,k}} \frac{\partial z_{t,i}}{\partial \hat{z}_{t,l}} + \sum_{(z_{t,i}, z_{t,j}) \in \mathcal{M}_{\mathbf{z}_t}} \frac{\partial^2 \log p(\mathbf{z}_t | \mathbf{z}_{t-1})}{\partial z_{t,i} \partial z_{t,j}} \frac{\partial z_{t,i}}{\partial \hat{z}_{t,k}} \frac{\partial z_{t,j}}{\partial \hat{z}_{t,l}},$$
$$\tag{18}$$

where $\frac{\partial^2 \log |\mathbf{H}|}{\partial \hat{z}_{t,k} \partial \hat{z}_{t,l}} = 0$ is removed from the derivation since Eq. 17, because the Jacobian matrix $\mathbf{H} = (\mathbf{I} - \mathbf{M})^{-1}$ is constant according to Eq. 10. Besides, as we assume linear mixing function in Eq. 3, the cross second order derivative $\frac{\partial^2 z_{t,i}}{\partial \hat{z}_{t,k} \hat{z}_{t,l}} = 0$, the terms including this are removed from the derivation since Eq. 17 as well.

From Eq. 18, for each value of $\mathbf{z}_t$, if we have $R = n + |\mathcal{M}_{\mathbf{z}_t}|$ different values of $\mathbf{z}_{t-1}$, i.e., $\{\mathbf{z}_{t-1}^{(r)}\}_{r\in[R]}$, to make the following matrix $\mathbf{C}_{\mathbf{z}_t}$ full rank, for each non-adjacent pair of $\hat{z}_{t,k}$ and $\hat{z}_{t,l}$ in the estimated Markov network $\mathcal{M}_{\hat{\mathbf{z}}_t}$, then we can conclude the relations between $\mathbf{z}_t$ and $\hat{\mathbf{z}}_t$. $\mathbf{C}_{\mathbf{z}_t}$ is,

$$\mathbf{C}_{\mathbf{z}_t} = \begin{pmatrix} \left(\frac{\partial^2 \log p(\mathbf{z}_t|\mathbf{z}_{t-1}^{(1)})}{\partial z_{t,i}^2}\right)_{i\in[n]}^T \oplus \left(\frac{\partial^2 \log p(\mathbf{z}_t|\mathbf{z}_{t-1}^{(1)})}{\partial z_{t,i}\partial z_{t,j}}\right)_{(z_{t,i},z_{t,j})\in\mathcal{M}_{\mathbf{z}_t}}^T \\ \vdots \\ \left(\frac{\partial^2 \log p(\mathbf{z}_t|\mathbf{z}_{t-1}^{(R)})}{\partial z_{t,i}^2}\right)_{i\in[n]}^T \oplus \left(\frac{\partial^2 \log p(\mathbf{z}_t|\mathbf{z}_{t-1}^{(R)})}{\partial z_{t,i}\partial z_{t,j}}\right)_{(z_{t,i},z_{t,j})\in\mathcal{M}_{\mathbf{z}_t}}^T \end{pmatrix}. \tag{19}$$

From the notation above, we can write down each column of $\mathbf{C}_{\mathbf{z}_t}$ as a function of $\mathbf{z}_{t-1}$ taking different values to form each row. According to Eq. 3, we have $\mathbf{z}_t = \mathbf{M}\mathbf{z}_t + \mathbf{B}\mathbf{z}_{t-1} + \epsilon_t$, which forms the joint distribution of $p(\mathbf{z}_t, \mathbf{z}_{t-1})$ by: $\begin{pmatrix} \mathbf{z}_t \\ \mathbf{z}_{t-1} \end{pmatrix} = \begin{pmatrix} (\mathbf{I}-\mathbf{M})^{-1}\mathbf{B} & (\mathbf{I}-\mathbf{M})^{-1} \\ \mathbf{I} & \mathbf{0} \end{pmatrix} \begin{pmatrix} \mathbf{z}_{t-1} \\ \epsilon_t \end{pmatrix}$, where the determinant of the Jacobian matrix of the transformation (in linear case, the Jacobian is the transformation itself) $\begin{pmatrix} (\mathbf{I}-\mathbf{M})^{-1}\mathbf{B} & (\mathbf{I}-\mathbf{M})^{-1} \\ \mathbf{I} & \mathbf{0} \end{pmatrix}$ is $-|(\mathbf{I}-\mathbf{M})^{-1}|$. By change of variable, we have $p(\mathbf{z}_t, \mathbf{z}_{t-1}) = p(\mathbf{z}_{t-1}, \epsilon_t)|(\mathbf{I}-\mathbf{M})^{-1}|$. And due to the independence between $\mathbf{z}_{t-1}$ and $\epsilon_t$, we have $p(\mathbf{z}_t, \mathbf{z}_{t-1}) = p(\mathbf{z}_{t-1})p(\epsilon_t)|(\mathbf{I}-\mathbf{M})^{-1}|$. Therefore, the conditional density is

$$p(\mathbf{z}_t|\mathbf{z}_{t-1}) = p(\epsilon_t)|(\mathbf{I}-\mathbf{M})^{-1}|$$
$$= \prod_{a\in[n]} p(\epsilon_{t,a})|(\mathbf{I}-\mathbf{M})^{-1}|$$
$$= \prod_{a\in[n]} p((\mathbf{I}_{a,:}-\mathbf{M}_{a,:})\mathbf{z}_t - \mathbf{B}_{a,:}\mathbf{z}_{t-1})|(\mathbf{I}-\mathbf{M})^{-1}|,$$

where $\mathbf{M}_{a,:}$ denotes the $a$-th row of the matrix.

Since the conditional density $p(\mathbf{z}_t|\mathbf{z}_{t-1})$ is second-smooth according to $A1$, the partial derivative of $\log p(\mathbf{z}_t|\mathbf{z}_{t-1})$ w.r.t. $z_{t,i}$ is

$$\frac{\partial \log p(\mathbf{z}_t|\mathbf{z}_{t-1})}{\partial z_{t,i}} = \sum_{a\in[n]} \frac{\partial \log p((\mathbf{I}_{a,:}-\mathbf{M}_{a,:})\mathbf{z}_t - \mathbf{B}_{a,:}\mathbf{z}_{t-1})}{\partial z_{t,i}} + \frac{\log|(\mathbf{I}-\mathbf{M})^{-1}|}{\partial z_{t,i}}$$
$$= \sum_{a\in[n]} (\mathbf{I}_{a,i}-\mathbf{M}_{a,i})\left(\frac{p'(\epsilon_{t,a})}{p(\epsilon_{t,a})}\right)$$
$$= \sum_{a\in[n]} \mathbf{K}_{a,i}\phi_a(\mathbf{z}_{t-1}, \mathbf{z}_t),$$

in which we use the abbreviation $p(\epsilon_{t,a}) = p((\mathbf{I}_{a,:}-\mathbf{M}_{a,:})\mathbf{z}_t - \mathbf{B}_{a,:}\mathbf{z}_{t-1})$, and therefore we let $\phi_a(\mathbf{z}_t, \mathbf{z}_{t-1}) = \frac{p'(\epsilon_{t,a})}{p(\epsilon_{t,a})}$. Besides, we also let $\mathbf{K} = \mathbf{I} - \mathbf{M}$ for the sake of simplicity.

Its partial derivative w.r.t. $z_{t,j}$ is:

$$\frac{\partial^2 \log p(\mathbf{z}_t|\mathbf{z}_{t-1})}{\partial z_{t,i}\partial z_{t,k}} = \sum_{a\in[n]} \mathbf{K}_{a,i}(\mathbf{I}_{a,j}-\mathbf{M}_{a,j})\left(\frac{p'(\epsilon_{t,a})}{p(\epsilon_{t,a})} - \frac{p''(\epsilon_{t,a})}{p(\epsilon_{t,a})^2}\right)$$
$$= \sum_{a\in[n]} \mathbf{K}_{a,i}\mathbf{K}_{a,j}\psi_a(\mathbf{z}_{t-1}, \mathbf{z}_t)$$

where $\psi_a(\mathbf{z}_{t-1}, \mathbf{z}_t) = -\left(\frac{p'(\epsilon_{t,a})}{p(\epsilon_{t,a})} - \frac{p''(\epsilon_{t,a})}{p(\epsilon_{t,a})^2}\right)$, given $p(\epsilon_{t,a}) = p((\mathbf{I}_{a,:}-\mathbf{M}_{a,:})\mathbf{z}_t - \mathbf{B}_{a,:}\mathbf{z}_{t-1})$.

When $j = i$,

$$\frac{\partial^2 \log p(\mathbf{z}_t|\mathbf{z}_{t-1})}{\partial z_{t,i}\partial z_{t,k}} = \sum_{a\in[n]} \mathbf{K}_{a,i}\mathbf{K}_{a,i}\psi_a(\mathbf{z}_{t-1}, \mathbf{z}_t).$$

Therefore, each row of $\mathbf{C}_{\mathbf{z}_t}$ can be represented as a vector of functions taking different values of $\mathbf{z}_{t-1}$:

$$\mathbf{C}_{\mathbf{z}_t}(\mathbf{z}_{t-1}^{(r)}) = (\sum_{a \in [n]} \mathbf{M}_{a,i} \mathbf{M}_{a,i} \psi_a(\mathbf{z}_{t-1}^{(r)}, \mathbf{z}_t))_{i \in [n]}^T \tag{20}$$

$$\oplus (\sum_{a \in [n]} \mathbf{M}_{a,i} \mathbf{M}_{a,j} \psi_a(\mathbf{z}_{t-1}^{(r)}, \mathbf{z}_t))_{(z_{t,i}, z_{t,j}) \in \mathcal{M}_{\mathbf{z}_t}}^T. \tag{21}$$

And if we have at least $R = n + |\mathcal{M}_{\mathbf{z}_t}|$ values of $\{\mathbf{z}_{t-1}^{(r)}\}_{r \in [R]}$ to make the matrix

$$\mathbf{C}_{\mathbf{z}_t} = (\mathbf{C}_{\mathbf{z}_t}(\mathbf{z}_{t-1}^{(1)}), \cdots, \mathbf{C}_{\mathbf{z}_t}(\mathbf{z}_{t-1}^{(R)}))^T$$

of shape $R \times R$ full-rank, then the relationships between the estimated $\hat{\mathbf{z}}_t$ and the true $\mathbf{z}_t$ can be solved by having nontrivial zero solutions in Eq. 18. However, it is impossible for $\mathbf{C}_{\mathbf{z}_t}$ to be $R \times R$ full-rank, because the concatenated two parts, $(\sum_{a \in [n]} \mathbf{M}_{a,i} \mathbf{M}_{a,i} \psi_a(\mathbf{z}_{t-1}^{(r)}, \mathbf{z}_t))_{i \in [n]}^T$ and $(\sum_{a \in [n]} \mathbf{M}_{a,i} \mathbf{M}_{a,j} \psi_a(\mathbf{z}_{t-1}^{(r)}, \mathbf{z}_t))_{(z_{t,i}, z_{t,j}) \in \mathcal{M}_{\mathbf{z}_t}}^T$ are both constant linear combinations of $\{\psi_a(\mathbf{z}_{t-1}^{(r)}, \mathbf{z}_t))_{(z_{t,i}, z_{t,j})}\}_{a \in [n]}$, for $\mathbf{M}$ does not change across timestamps. It constrains the rank of $\mathbf{C}_{\mathbf{z}_t}$ to be at most $n$, which implies any non empty instantaneous Markov network leads to the unidentifiability of $\mathbf{z}_t$.

## A.2 Proof for Theorem 1

**Theorem 1** (Latent Indeterminacy). *Suppose the estimated model $(\hat{\mathbf{A}}, \{\hat{\mathbf{B}}_\tau\}_{\tau=1}^L, \hat{\mathbf{M}}, p_{\hat{\epsilon}})$ and the true model $(\mathbf{A}, \{\mathbf{B}_\tau\}_{\tau=1}^L, \mathbf{M}, p_\epsilon)$ both generate $\mathbf{x}_t$ according to Eq. 3 and are observationally equivalent on the autocovariance matrices $\mathbf{R}_{\mathbf{x}}(k)$ for $k = 0, 1, \ldots, L$. If conditions A1–A4 hold, then the model parameters are identifiable up to the following indeterminacies:*

$$\hat{\boldsymbol{\epsilon}}_t = \mathbf{P}\boldsymbol{\epsilon}_t, \quad \hat{\mathbf{A}} = \mathbf{A}\mathbf{S}, \quad (\mathbf{I} - \hat{\mathbf{M}}) = \mathbf{P}(\mathbf{I} - \mathbf{M})\mathbf{S}, \quad \hat{\mathbf{B}}_\tau = \mathbf{P}\mathbf{B}_\tau\mathbf{S},$$

*where $\mathbf{S} \in \mathbb{R}^{n \times n}$ is invertible and $\mathbf{P}$ is a signed permutation matrix.*

- *A1 (Temporally white noise). Both $\mathbf{e}_t$ and $\boldsymbol{\epsilon}_t$ are temporally white, and $\boldsymbol{\epsilon}_t$ has i.i.d. mutually independent components. To remove scaling indeterminacy, assume $\boldsymbol{\Sigma}_{\boldsymbol{\epsilon}_t} = \mathbf{I}$ and zero-mean data.*

- *A2 (Rank sufficiency). $\mathbf{A} \in \mathbb{R}^{m \times n}$ $(m \geq n)$ has full column rank, and $\mathbf{B}_L$ is full rank.*

- *A3 (Process stability). Eq. 22 defines a stable vector autoregression, i.e., all roots of $\det(\mathbf{I} - \sum_{\tau=1}^L \mathbf{B}_\tau y^{-\tau}) = 0$ lie strictly inside the unit circle.*

- *A4 (Non-Gaussianity). At most one component of $\boldsymbol{\epsilon}_t$ is Gaussian.*

*Proof.* A more condensed formulation is used, which is the same as Eq. 3, for the sake of matrix transformation presentation in the proof.

$$\mathbf{z}_t = \mathbf{M}\mathbf{z}_t + \sum_{\tau=1}^L \mathbf{B}_\tau \mathbf{z}_{t-\tau} + \boldsymbol{\epsilon}_t, \tag{22}$$

$$\mathbf{x}_t = \mathbf{A}\mathbf{z}_t, \tag{23}$$

**Step 1: Construct Autocovariance Matrices** According to the formulation for $\mathbf{z}_t$, we can derive its autocovariance at lag $k$ as:

$$\mathbf{R}_{\mathbf{z}}(k) = \mathbb{E}\left[\mathbf{z}_t \left(\sum_{\tau=1}^L (\mathbf{I} - \mathbf{M})^{-1} \mathbf{B}_\tau \mathbf{z}_{t+k-\tau} + (\mathbf{I} - \mathbf{M})^{-1} \varepsilon_{t+k}\right)^T\right]$$

$$= \sum_{\tau=1}^L \mathbb{E}[\mathbf{z}_t \mathbf{z}_{t+k-\tau}^T] \mathbf{B}_\tau^T (\mathbf{I} - \mathbf{M})^{-T} + \mathbb{E}[\mathbf{z}_t \varepsilon_{t+k}^T] (\mathbf{I} - \mathbf{M})^{-T}.$$

Hence,

$$
\mathbf{R_z}(k) = \begin{cases} \displaystyle\sum_{\tau=1}^{L} \mathbf{R_z}(k-\tau)\,\mathbf{B}_\tau^T (\mathbf{I} - \mathbf{M})^{-T}, & k \neq 0, \\[2em] \displaystyle\sum_{\tau=1}^{L} \mathbf{R_z}(k-\tau)\,\mathbf{B}_\tau^T (\mathbf{I} - \mathbf{M})^{-T} + (\mathbf{I} - \mathbf{M})^{-T}, & k = 0. \end{cases}
$$

Because $\mathbf{x}_t = \mathbf{A}\mathbf{z}_t$,

$$
\mathbf{R_x}(k) = \mathbf{A}\mathbf{R_z}(k)\mathbf{A}^T.
$$

Thus,

$$
\mathbf{R_x}(k) = \begin{cases} \displaystyle\sum_{\tau=1}^{L} \mathbf{A}\mathbf{R_z}(k-\tau)\mathbf{A}^T\mathbf{A}^{-T}\mathbf{B}_\tau^T(\mathbf{I}-\mathbf{M})^{-T}\mathbf{A}^T, & k \neq 0, \\[2em] \displaystyle\sum_{\tau=1}^{L} \mathbf{A}\mathbf{R_z}(k-\tau)\mathbf{A}^T\mathbf{A}^{-T}\mathbf{B}_\tau^T(\mathbf{I}-\mathbf{M})^{-T}\mathbf{A}^T + \mathbf{A}(\mathbf{I}-\mathbf{M})^{-T}\mathbf{A}^T, & k = 0. \end{cases}
$$

Simplifying,

$$
\mathbf{R_x}(k) = \begin{cases} \displaystyle\sum_{\tau=1}^{L} \mathbf{R_x}(k-\tau)\mathbf{A}^{-T}\mathbf{B}_\tau^T(\mathbf{I}-\mathbf{M})^{-T}\mathbf{A}^T, & k \neq 0, \\[2em] \displaystyle\sum_{\tau=1}^{L} \mathbf{R_x}(k-\tau)\mathbf{A}^{-T}\mathbf{B}_\tau^T(\mathbf{I}-\mathbf{M})^{-T}\mathbf{A}^T + \mathbf{A}(\mathbf{I}-\mathbf{M})^{-T}\mathbf{A}^T, & k = 0. \end{cases}
$$

Let

$$
\mathbf{C}_\tau = \mathbf{A}(\mathbf{I} - \mathbf{M})^{-1}\mathbf{B}_\tau \mathbf{A}^{-1}.
$$

Then,

$$
\mathbf{R_x}(k) = \begin{cases} \displaystyle\sum_{\tau=1}^{L} \mathbf{R_x}(k-\tau)\mathbf{C}_\tau^T, & k \neq 0, \\[2em] \displaystyle\sum_{\tau=1}^{L} \mathbf{R_x}(k-\tau)\mathbf{C}_\tau^T + \mathbf{A}(\mathbf{I}-\mathbf{M})^{-T}\mathbf{A}^T, & k = 0. \end{cases}
$$

**Step 2: Solving the Linear System Formed by Autocovariance**  Use the derived $\mathbf{R_x}(k)$, we can construct the following linear system.

$$
\begin{pmatrix} \mathbf{R_x}(0) - \mathbf{H}\mathbf{H^T} \\ \mathbf{R_x}(1) \\ \mathbf{R_x}(2) \\ \vdots \\ \mathbf{R_x}(L) \end{pmatrix} = \begin{pmatrix} \mathbf{R_x}(1)^T & \mathbf{R_x}(2)^T & \cdots & \mathbf{R_x}(L)^T \\ \mathbf{R_x}(0) & \mathbf{R_x}(1)^T & \cdots & \mathbf{R_x}(L-1)^T \\ \mathbf{R_x}(1) & \mathbf{R_x}(0) & \cdots & \mathbf{R_x}(L-2)^T \\ \vdots & \vdots & \ddots & \vdots \\ \mathbf{R_x}(L-1) & \mathbf{R_x}(L-2) & \cdots & \mathbf{R_x}(0) \end{pmatrix} \begin{pmatrix} \mathbf{C}_1^T \\ \mathbf{C}_2^T \\ \vdots \\ \mathbf{C}_L^T \end{pmatrix}, \qquad (24)
$$

where $\mathbf{H} = \mathbf{A}(\mathbf{I} - \mathbf{M})^{-1}$.

Let's look at the lower $mL$ rows of the coefficient matrix of the linear system:

$$
\mathbf{Q} = \begin{pmatrix} \mathbf{R_x}(0) & \mathbf{R_x}(1)^T & \cdots & \mathbf{R_x}(L-1)^T \\ \mathbf{R_x}(1) & \mathbf{R_x}(0) & \cdots & \mathbf{R_x}(L-2)^T \\ \vdots & \vdots & \ddots & \vdots \\ \mathbf{R_x}(L-1) & \mathbf{R_x}(L-2) & \cdots & \mathbf{R_x}(0) \end{pmatrix}.
$$

It is clear that $\mathbf{Q} = \mathbb{E}[\vec{\mathbf{x}_t}\vec{\mathbf{x}_t}^T]$, $\vec{\mathbf{x}_t} = (\mathbf{x}_t^T, \mathbf{x}_{t-1}^T, \cdots, \mathbf{x}_{t+L-1}^T)^T$. Because the process $\{\mathbf{x}_t\}$ admits no nontrivial deterministic linear relation among $\mathbf{x}_t, \mathbf{x}_{t-1}, \ldots, \mathbf{x}_{t-L+1}$ due to $\epsilon$, $\mathbf{Q} = \mathbb{E}[\vec{\mathbf{x}_t}\vec{\mathbf{x}_t}^T]$ is always positive definite and therefore nonsingular.

Thus, we can solve the lower $mL$ linear system to obtain $\mathbf{C}_k, k = 1, \cdots, L$ and then use the solved $\mathbf{C}_k$ to obtain $\mathbf{HH}^T$. Therefore, suppose we have two models, $(\mathbf{A}, \mathbf{M}, \{\mathbf{B}_\tau\}_{\tau=1}^L, \mathbf{\Sigma}_\epsilon)$ and $(\hat{\mathbf{A}}, \hat{\mathbf{M}}, \{\hat{\mathbf{B}}_\tau\}_{\tau=1}^L, \mathbf{\Sigma}_{\hat{\epsilon}})$, and they are observationally equivalent on $\mathbf{R}_\mathbf{x}(k), k = 0, 1, \cdots, L$, then we have the relations between there parameters as follows:

$$\hat{\mathbf{A}}(\mathbf{I} - \hat{\mathbf{M}})^{-1}(\mathbf{I} - \hat{\mathbf{M}})^{-T}\hat{\mathbf{A}}^T = \mathbf{A}(\mathbf{I} - \mathbf{M})^{-1}(\mathbf{I} - \mathbf{M})^{-T}\mathbf{A}^T \tag{25}$$

$$\hat{\mathbf{A}}(\mathbf{I} - \hat{\mathbf{M}})^{-1}\hat{\mathbf{B}}_\tau\hat{\mathbf{A}}^{-1} = \mathbf{A}(\mathbf{I} - \mathbf{M})^{-1}\mathbf{B}_\tau\mathbf{A}^{-1}, \forall \tau = 1, \cdots, L. \tag{26}$$

**Step 3: Further Investigate $\mathbf{B}_\tau$ and $\mathbf{M}$** From the RHS of Eq. 25, we have:

$$\mathbf{HH}^T = \mathbf{A}(\mathbf{I} - \mathbf{M})^{-1}(\mathbf{I} - \mathbf{M})^{-T}\mathbf{A}^T, \tag{27}$$

which gives $\mathrm{Col}(\mathbf{HH}^T) \subseteq \mathrm{Col}(\mathbf{A})$. Thank to the fact that $\mathbf{A}$ has full column rank, entailing its hasing left inverse $\mathbf{A}^{-1}\mathbf{A} = \mathbf{I}$, and $\mathbf{I} - \mathbf{M}$ being invertible, we have

$$\mathbf{A} = \mathbf{HH}^T\mathbf{A}^{-T}(\mathbf{I} - \mathbf{M})^T(\mathbf{I} - \mathbf{M}), \tag{28}$$

which further gives $\mathrm{Col}(\mathbf{A}) \subseteq \mathrm{Col}(\mathbf{HH}^T)$.

Therefore, $\mathrm{Col}(\mathbf{A}) = \mathrm{Col}(\mathbf{HH}^T)$. As we can derive $\mathrm{Col}(\hat{\mathbf{A}}) = \mathrm{Col}(\mathbf{HH}^T)$ from the LHS from Eq. 25, we have: $\mathrm{Col}(\mathbf{A}) = \mathrm{Col}(\hat{\mathbf{A}})$. Since both $\mathbf{A}$ and $\hat{\mathbf{A}}$ are column wise full rank, we have

$$\underline{\hat{\mathbf{A}} = \mathbf{AS}} \tag{29}$$

for some invertible $\mathbf{S} \in \mathbb{R}^{d \times d}$.

From $\mathbf{HH}^T = \hat{\mathbf{H}}\hat{\mathbf{H}}^T$, with $\mathbf{H} = \mathbf{A}(\mathbf{I} - \mathbf{M})^{-1}$ showing that $\mathbf{H}$ has left inverse, we have

$$\mathbf{H}^{-1}\hat{\mathbf{H}}(\mathbf{H}^{-1}\hat{\mathbf{H}})^T = \mathbf{I}_d \tag{30}$$

Therefore, we can derive that $\mathbf{H}^{-1}\hat{\mathbf{H}} = \mathbf{U}$, where $\mathbf{U}$ is an orthogonal matrix. However, since $\hat{\mathbf{H}}$ only has left inverse (the same as $\mathbf{H}$), we cannot directly conclude how $\mathbf{U}$ plays the role in the relationship between $\mathbf{H}$ and $\hat{\mathbf{H}}$ but need the following derivation.

Given the invertibility of $\mathbf{I} - \mathbf{M}$ and $\mathrm{Col}(\mathbf{A}) = \mathrm{Col}(\hat{\mathbf{A}})$, we know $\mathrm{Col}(\mathbf{H}) = \mathrm{Col}(\hat{\mathbf{H}}) \Rightarrow \hat{\mathbf{H}} \in \mathrm{Col}(\mathbf{H})$. Since $\mathbf{H}$ has right inverse only in the column space $\mathrm{Col}(\mathbf{H})$, we have

$$\mathbf{H}(\mathbf{H}^{-1}\hat{\mathbf{H}}) = \mathbf{HU} \Rightarrow (\mathbf{HH}^{-1})\hat{\mathbf{H}} = \mathbf{HU} \xrightarrow{\hat{\mathbf{H}} \in \mathrm{Col}(\mathbf{H})} \mathbf{I}_d\hat{\mathbf{H}} = \mathbf{HU} \Rightarrow \hat{\mathbf{H}} = \mathbf{HU}. \tag{31}$$

Therefore,

$$\hat{\mathbf{A}}(\mathbf{I} - \hat{\mathbf{M}})^{-1} = \mathbf{A}(\mathbf{I} - \mathbf{M})^{-1}\mathbf{U} \tag{32}$$

$$\Rightarrow \mathbf{AS}(\mathbf{I} - \hat{\mathbf{M}})^{-1} = \mathbf{A}(\mathbf{I} - \mathbf{M})^{-1}\mathbf{U} \tag{33}$$

$$\Rightarrow \underline{(\mathbf{I} - \hat{\mathbf{M}}) = \mathbf{U}^T(\mathbf{I} - \mathbf{M})\mathbf{S}}. \tag{34}$$

Finally, substituting Eq. 26 with Eq. 34 and Eq. 29, for all $\tau = 1, \cdots, L$ we have,

$$\hat{\mathbf{A}}(\mathbf{I} - \hat{\mathbf{M}})^{-1}\hat{\mathbf{B}}_\tau\hat{\mathbf{A}}^{-1} = \mathbf{A}(\mathbf{I} - \mathbf{M})^{-1}\mathbf{U}\hat{\mathbf{B}}_\tau\mathbf{S}^{-1}\mathbf{A}^{-1} = \mathbf{A}(\mathbf{I} - \mathbf{M})^{-1}\mathbf{B}_\tau\mathbf{A}^{-1} \tag{35}$$

$$\Rightarrow \mathbf{U}\hat{\mathbf{B}}_\tau\mathbf{S}^{-1} = \mathbf{B}_\tau \tag{36}$$

$$\Rightarrow \underline{\hat{\mathbf{B}}_\tau = \mathbf{U}^T\mathbf{B}_\tau\mathbf{S}}. \tag{37}$$

So far, all the relations derived between the true model $(\mathbf{A}, \{\mathbf{B}_\tau\}_{\tau=1}^L, \mathbf{M}, \Sigma_\epsilon)$ and the estimated model $(\hat{\mathbf{A}}, \{\hat{\mathbf{B}}_\tau\}_{\tau=1}^L, \hat{\mathbf{M}}, \Sigma_{\hat{\epsilon}})$ are underlined.

**Step 4: Utilizing Non-Gaussianity** Suppose the true model $(\mathbf{A}, \{\mathbf{B}_\tau\}_{\tau=1}^L, \mathbf{M}, \Sigma_{\mathbf{e}})$ and the estimated model $(\hat{\mathbf{A}}, \{\hat{\mathbf{B}}_\tau\}_{\tau=1}^L, \hat{\mathbf{M}}, \Sigma_{\hat{\mathbf{e}}})$ are observationally equivalent:

$$\mathbf{x}_t = \mathbf{A} \left( (\mathbf{I} - \mathbf{M}) - \sum_{\tau=1}^L \mathbf{B}_\tau z^{-\tau} \right)^{-1} \epsilon_t \tag{38}$$

$$= \hat{\mathbf{A}} \left( (\mathbf{I} - \hat{\mathbf{M}}) - \sum_{\tau=1}^L \hat{\mathbf{B}}_\tau z^{-\tau} \right)^{-1} \hat{\epsilon}_t \tag{39}$$

$$= \mathbf{A}\mathbf{S} \left( \mathbf{U}^T(\mathbf{I} - \mathbf{M})\mathbf{S} - \sum_{\tau=1}^L \mathbf{U}^T\mathbf{B}_\tau\mathbf{S}z^{-\tau} \right)^{-1} \hat{\epsilon}_t \tag{40}$$

$$= \mathbf{A} \left( (\mathbf{I} - \mathbf{M}) - \sum_{\tau=1}^L \mathbf{B}_\tau z^{-\tau} \right)^{-1} \mathbf{U}\hat{\epsilon}_t \quad \text{(commute } \mathbf{B}_\tau \text{ with } z^{-\tau}) \tag{41}$$

$$\Rightarrow \mathbf{A} \left( (\mathbf{I} - \mathbf{M}) - \sum_{\tau=1}^L \mathbf{B}_\tau z^{-\tau} \right) (\epsilon_t - \mathbf{U}\hat{\epsilon}_t) = \mathbf{0} \tag{42}$$

$$\Rightarrow (\epsilon_t - \mathbf{U}\hat{\epsilon}_t) = \mathbf{0}. \tag{43}$$

Therefore, we can derive that $\hat{\epsilon}_t = \mathbf{U}^T\epsilon_t$. Because there is at most one Gaussian component in $\epsilon_t$ and $\Sigma_{\epsilon_t} = \mathbf{I}$, we know that $\mathbf{U}^T$ must be a signed permutation, i.e., $\underline{\mathbf{U}^T = \mathbf{P}}$, where $\mathbf{P}$ is a permutation matrix and taking values from $\{+1, -1\}$ for its nonzero entries. To conclude, we derive $\hat{\mathbf{A}} = \mathbf{A}\mathbf{S}$, $(\mathbf{I} - \hat{\mathbf{M}}) = \mathbf{U}^T(\mathbf{I} - \mathbf{M})\mathbf{S}$, and $\hat{\mathbf{B}}_\tau = \mathbf{U}^T\mathbf{B}_\tau\mathbf{S}$, where $\mathbf{B}$ is an invertible matrix and $\mathbf{P}$ is a signed permutation matrix. $\qquad\square$

## A.3 Synthetic Experiments

We conduct two synthetic verification experiments to validate our linear temporal instantaneous ICA method. Instruction is provided in the `synthetic/README.md` file in our code repository.

### A.3.1 Fixed Structure Experiment

For the first synthetic verification experiment, we generate data using fixed time-delayed influence functions and instantaneous relations with the following ground truth matrices:

$$\mathbf{B} = \begin{bmatrix} 0.4 & 0.6 & 0 \\ 0 & 1 & 0 \\ 0 & 0 & 1 \end{bmatrix}, \quad \mathbf{M} = \begin{bmatrix} 0 & 0 & 0 \\ 0.2 & 0 & 0 \\ 0 & 0.2 & 0 \end{bmatrix}. \tag{44}$$

The data generation process follows a structured temporal model. We initialize the first hidden state $z_0$ by sampling from a uniform distribution $\mathcal{U}(0, 1)$. For subsequent time steps, we compute the historical influence as $\mathbf{z}_{\text{hist}} = \mathbf{B}\,\mathbf{z}_{t-1}$ and then construct $z_t$ iteratively: the first dimension receives only historical influence plus noise, while remaining dimensions $i \geq 2$ incorporate both historical and instantaneous dependencies:

$$z_t^{(1)} = z_{\text{hist}}^{(1)} + \epsilon_t^{(1)} \tag{45}$$

$$z_t^{(i)} = z_{\text{hist}}^{(i)} + w_{\text{inst}} \cdot z_t^{(i-1)} + \epsilon_t^{(i)}, \quad i \geq 2 \tag{46}$$

where $\epsilon_t$ is Laplace noise with scale 1.0, and $w_{\text{inst}} = 0.2$. The observations are generated as $\mathbf{x}_t = Az_t$ where $A$ is a $3 \times 3$ randomly initialized mixing matrix .

We train the model for 50,000 steps with batch size 1024 (approximately 51 million total samples) using the Adam optimizer with learning rate $8 \times 10^{-3}$ and weight decay $6 \times 10^{-4}$. The loss function includes reconstruction error, KL divergence term, and L1 regularization penalties: $1 \times 10^{-3}$ for matrix $\mathbf{M}$ and $1 \times 10^{-8}$ for matrix $\mathbf{B}$. We enforce the lower-triangular constraint on $\mathbf{M}$ to ensure identifiability.

### A.3.2 Scalability Experiment

For the second synthetic experiment, we evaluate scalability across different dimensions ranging from 64 to 1024. We randomly sample a sparse time-delayed transition matrix $\mathbf{B}$ where only 10% of the entries are non-zero, generated using a randomly initialized matrix with 10% masking.

For the instantaneous mixing matrix $\mathbf{M}$, we use a chain structure where $\mathbf{M}_{i,i-1} = 0.5$ for $i \geq 2$ and all other entries are zero:

$$M = \begin{bmatrix} 0 & 0 & 0 & \cdots & 0 \\ 0.5 & 0 & 0 & \cdots & 0 \\ 0 & 0.5 & 0 & \cdots & 0 \\ \vdots & \ddots & \ddots & \ddots & \vdots \\ 0 & \cdots & 0 & 0.5 & 0 \end{bmatrix} \tag{47}$$

The training hyperparameters are modified from the first experiment: learning rate increased to $1 \times 10^{-3}$ and the sparsity coefficient for $\mathbf{B}$ increased to $1 \times 10^{-5}$ to account for the higher dimensional setting, while maintaining $1 \times 10^{-3}$.

Both experiments use identical noise characteristics (Laplace distribution with unit scale), sequence length of 1 (two time steps total), and Mean Correlation Coefficient (MCC) as the primary evaluation metric to measure the quality of source recovery while accounting for permutation ambiguity inherent in ICA methods.

## A.4 Semi-synthetic Experiments

### A.4.1 Target Concept Relation Recovery

Before attempting to recover concept relationships from real-world LLM activations, and based on the proven and verified identifiability of our model, we first present a semi-synthetic setting to verify that our proposal can reveal obvious concept relations from contrastive corpus pairs.

**Data Preparation**    We constructed two contrastive collections of texts drawn from the `Pile` dataset [17]. We considered three types of text: legal documents, emails, and XML files. For each type, we constructed two contrastive corpora: one containing the relation of interest, and the other lacking it. Specifically, for legal text, the relation is defined by sequences beginning with "APPEALS" and ending with "AFFIRMED"; for emails, sequences start with forwarding or reply markers (e.g., dashes) and end with common words like "Subject" or "Thanks"; and for XML, sequences start with a UTF encoding label followed by tags such as "UTF-8" or "!DOCTYPE". We hypothesized that only texts containing these structured relations would yield meaningful temporal concept patterns, and we directly tested whether the model can successfully recover such patterns.

**Baseline Construction**    Although there is no directly applicable baseline, we leveraged standard SAEs we had trained above to serve as our baseline method. Since SAEs themselves cannot capture the concept-to-concept relations, we train a regression model to find temporal relation coefficient matrices $\tilde{\mathbf{B}}$s by solving the following regression task: $\mathbf{z}_t = \sum_\tau \tilde{\mathbf{B}}_\tau \mathbf{z}_{t-\tau}$.

**Evaluation Metric**    We calculate the concept recovery score by first obtaining the top fired feature index pair $(i, j)$ related to the legal context (restricted to positions where the concepts of interest ought to fire but do not fire in the contrastive non-legal text), and then taking the corresponding entry $\mathbf{B}_{i,j}$ in the aggregated temporal relation coefficient matrices. We then calculate the relation recovery score, similar to a signal-to-noise ratio, by: `relation recovery score` $= \frac{\mathbf{B}_{i,j}}{\sigma(B)}$, where $\sigma(\mathbf{B})$ denotes the standard deviation of matrix $\mathbf{B}$. Such ratio indicates the extent that the target concept relation entry in the matrix is more significant than a random noise; the larger the score is the more significant the relation recovery.

**Results**    All results are shown in Table 1, which verifies that our proposal can identify the concept-relation of interest from contrastive corpus pairs. For the demonstrated results, we used the same trained model as in the experiments on recovering relations from real-world LLM activations.

### A.4.2 Steering Vector Recovery

Except for the relationships between concepts, our model is also able to recover the concepts as current SAEs. To verify this, semi-synthetic benchmarks like SSAE [23] can offer valuable insights into concept identifiability. Following this setting, we tested whether our model can recover steering vectors from paired text. Specifically, we constructed five categories of word pairs where only a single interpretable concept changes, including gender, plurality, comparative, tense, and negation. While these changes are intuitive, ensuring the word pairs capture a clear ground-truth concept is non-trivial. Despite this challenge, our model demonstrated strong performance in identifying the underlying concept differences. Specifically, (1) the average correlation of concept differences within each category exceeded 0.86; (2) assuming one ground-truth pair, the correlation rose above 0.93; and (3) the maximum correlation within a category reached over 0.94. These results support our claim that our model can indeed recover meaningful steering vectors. The word lists for the five categories is summarized in Table 4.

## A.5 LLM Activation Experiments

In addition to the experimental results presented in Section 5.3 of the main text, we provide here: (1) detailed settings for training and inference; (2) visualizations of training losses and sparsity values; (3) comparisons across different hyper-parameter settings, and (4) extended experiment on SAEBench with larger latent size and base language models.

### A.5.1 Details on the Real-world Experiments Settings

**Training** We train our linear model on activations from the pretrained LLM `pythia-160m-deduped` [5], using SAELens [6] and dictionary-learning [35] for activation extraction. Importantly, in the original implementation of dictionary-learning [35], activations are loaded using an object named `ActivationBuffer`, which is refreshed with new activations once a predefined consumption threshold is reached. During each refresh, a random shuffling is applied. However, this randomization disrupts the temporal structure of the LLM activations. To preserve temporal information, we modify the corresponding refresh function to disable the random shuffling. Details of this modification can be found in the `examples/README.md` file in our code repository.

The model is trained on a total of 50 million tokens from the `Pile` dataset [17]. To capture time-delayed influences, we consider two values of $\tau$, namely $\{5, 20\}$, as described in Eq. 3. While our main results focus on the setting with $\tau = 20$, which offers better guarantees for capturing rich temporal semantics, this choice will be further justified in a later section of the supplementary materials. To address the distributed and uncertain nature of time-delayed dependencies—where some relations manifest over longer time spans and others over shorter ones—we aggregate the $\mathbf{B}_\tau$ matrices using max-pooling. This operation preserves any causal link that appears at any time step. We refer to the resulting aggregated matrix as $aggB$. Unless otherwise specified, the weight of the independence constraint on the noise term is set to $\alpha = 0.1$ in Eq. 9.

To better enforce sparsity in the hidden feature activations, we apply TopK filtering [8] in addition to the $\ell_1$ sparsity term included in the final loss function. Given the importance of feature dimensionality in Sparse Autoencoders (SAEs), we evaluate three configurations: 768 (which directly matches the LLM's hidden size and aligns with the identifiability discussion in Section 3), 3072, and 6144—the latter follow the considerations of SAE literature. We optimize the loss function defined in Eq. 9 using the Adam optimizer with a learning rate of 0.01 and a weight decay of 0.0001. Unless otherwise specified, we use a random seed of 123; additional experiments were conducted with seeds 456 and 789 for robustness.

**Inference** During inference, our primary goal is to interpret the hidden features—particularly those activated by significant entries in the time-delayed ($aggB$) or instantaneous ($\mathbf{M}$) relation matrices. This selection process differs from conventional SAE interpretation, which typically examines feature importance across the entire feature space by measuring activation strength for a given prompt. In contrast, our method emphasizes the relational structure of features—how they connect to form semantic transitions. We aim to understand the meaning of each feature by analyzing how both types of relations (instantaneous and time-delayed) link features together.

Table 4: Summary of word pairs in the five categories

| Categories | Pairs |
|---|---|
| Gendered Pairs | (male, female), (actor, actress), (prince, princess), (king, queen), (god, goddess), (wizard, witch), (boy, girl), (man, woman), (father, mother), (son, daughter), (brother, sister), (husband, wife), (nephew, niece), (uncle, aunt), (gentleman, lady), (monk, nun), (grandfather, grandmother), (lord, lady), (spokesman, spokeswoman) |
| Plurality Pairs | (cat, cats), (dog, dogs), (apple, apples), (box, boxes), (child, children), (book, books), (car, cars), (tree, trees), (house, houses), (bird, birds), (chair, chairs), (table, tables), (shoe, shoes), (shirt, shirts), (sock, socks), (cup, cups), (plate, plates), (pen, pens), (bag, bags), (door, doors), (window, windows), (lamp, lamps), (phone, phones), (laptop, laptops), (flower, flowers), (cloud, clouds), (mountain, mountains), (river, rivers), (lake, lakes), (egg, eggs), (grape, grapes), (potato, potatoes), (tomato, tomatoes), (bus, buses), (kiss, kisses), (wish, wishes), (match, matches), (dish, dishes), (baby, babies), (lady, ladies), (city, cities), (party, parties), (family, families), (knife, knives), (leaf, leaves), (wolf, wolves) |
| Comparative Pairs | (fast, faster), (tall, taller), (small, smaller), (old, older), (young, younger), (short, shorter), (long, longer), (high, higher), (low, lower), (strong, stronger), (weak, weaker), (rich, richer), (poor, poorer), (hard, harder), (soft, softer), (loud, louder), (bright, brighter), (dark, darker), (clean, cleaner), (easy, easier), (happy, happier), (cool, cooler), (deep, deeper), (wide, wider), (narrow, narrower), (thick, thicker), (thin, thinner), (heavy, heavier), (light, lighter), (safe, safer), (cheap, cheaper) |
| Tense Change Pairs | (walk, walked), (run, ran), (eat, ate), (go, went), (write, wrote), (speak, spoke), (drink, drank), (drive, drove), (read, read), (sleep, slept), (sit, sat), (stand, stood), (fly, flew), (begin, began), (buy, bought), (bring, brought), (build, built), (catch, caught), (choose, chose), (come, came), (cut, cut), (dig, dug), (do, did), (draw, drew), (fall, fell), (feel, felt), (find, found), (get, got), (give, gave), (have, had), (hear, heard), (hide, hid), (hold, held), (keep, kept), (know, knew), (leave, left), (lose, lost), (make, made), (meet, met), (pay, paid), (ride, rode), (say, said), (see, saw), (sell, sold), (send, sent), (sing, sang), (sit, sat), (teach, taught), (think, thought) |
| Negative Prefix Pairs | (possible, impossible), (legal, illegal), (visible, invisible), (complete, incomplete), (fair, unfair), (known, unknown), (fortunate, unfortunate), (able, unable), (happy, unhappy), (certain, uncertain), (clear, unclear), (real, unreal), (necessary, unnecessary), (likely, unlikely), (available, unavailable), (comfortable, uncomfortable), (pleasant, unpleasant), (reliable, unreliable), (acceptable, unacceptable), (usual, unusual), (wanted, unwanted), (expected, unexpected), (connected, disconnected), (understood, misunderstood), (placed, misplaced) |

Our feature selection process involves the following steps: First, we select the top 100 coordinates (we also tried 200, though 100 proved sufficient) from either $aggB$ or $\mathbf{M}$, and extract the corresponding feature dimensions. Next, we generate 10,000 prompts from the `EleutherAI/pile` dataset, convert them into token streams, and feed them into the trained model to observe how each token responds to each selected concept feature. Finally, for each selected feature, we collect the tokens whose activations exceed a threshold (set to 3.0), along with their corresponding prompts. These tokens are viewed as consequences of the activation of the given feature, while the associated prompts serve as contexts that reveal the token and therefore, feature's meaning.

### A.5.2 Visualizations of Training Loss and Sparsity Metrics

Here, we compare the training dynamics across different settings by examining the reconstruction loss (Eq. 4), the independence of the estimated noise term (Eq. 7), and the sparsity of both time-delayed and instantaneous relations (Eq. 8). The comparisons are made with respect to variations in hidden feature dimensionality, the sparsity weight on learned relations (i.e., $\beta$ in Eq.9), the temporal coverage of delayed relations, as determined by $\tau \in \{5, 20\}$, and the parameter of the TopK filtering of the hidden features.

We begin by examining the training dynamics with $\tau = 5$, comparing different settings of the sparsity constraint ($\beta \in \{0.1, 0.01\}$), TopK values ($\{0, 25, 100\}$, where 0 indicates that TopK is disabled), and hidden dimensions (z_dim $\in \{768, 3072, 6144\}$). The corresponding results are presented in Figure 8. It is worth noting that certain unstable training batches occasionally impact the overall stability during training. However, since most of the configurations eventually converge and our primary interest lies in the behavior at convergence, we cap the y-axis at 5.0 to improve the clarity of the visualizations. Our key findings are summarized below.

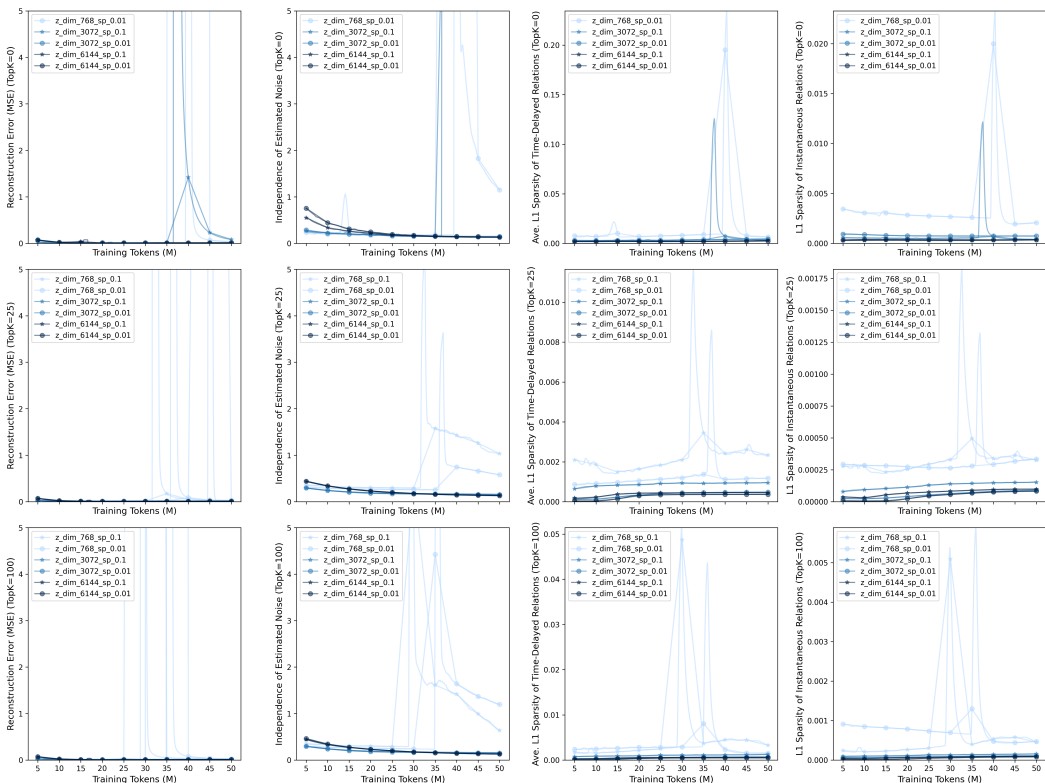

Figure 8: Dynamics of reconstruction loss, noise independence, and time-delayed and instantaneous relations sparsity with setting $\tau$ to 5. The x-axis starts at 5M tokens, and the y-axis values are capped at 5 to enhance visualization clarity.

**Insights on the Number of Training Tokens and the Impact of Hidden Feature Dimensionality**
From Figure 8, we observe that 50M training tokens are sufficient for convergence across all settings when the hidden feature dimensionality is greater than 768—specifically, at 3072 and 6144. From the subplots in the first column, it is evident that higher-dimensional hidden features provide greater stability during training. This increased robustness likely helps mitigate the effects of noisy or unstable batches within the token stream, leading to more consistent optimization of the objective. Consequently, in the subsequent case studies, including Section 5.3 of the main content, we focus on the settings with hidden dimensionalities of 3072 and 6144.

**Impact of TopK Filtering** The training process is in general more stable after applying TopK filtering. More specifically, comparing the sub-diagrams from the first row in Figure 8 to the second

and the third rows, we can see that the decrease of the reconstruction error is significantly less effected by some of the token batches, especially, for the setting when feature dimension is set to 3072 or 6144.

**Impact of Sparsity Strength**  In general, when $\beta$ sets to 0.01 (pay attention to the round marker in Figure 8 as oppose to star marker), both the time-delayed and the instantaneous relations show lower sparsity compared with a stronger sparsity weight. This might be due to a weaker constrain that results a better optimization results, while the stronger one might increase the sharpness of the potential solution space. This also indicates that 0.01 is sufficient for achieving our goal of sparse causal relations in our model.

### A.5.3 Sensitivity and Ablation Studies

**Sensitivity Study on $\alpha$ and $\beta$**  We conducted additional comparisons with $\beta = 0, 0.001, 0.005,$ 0.05, 1.0 and $\alpha = 0, 0.001, 0.01$ to cover a broader hyperparameter range, using $\tau = 5$ and feature dimension 3072. The results are shown in the two tables below, with our selected setting in bold text. The tables highlight that (1) concept relationships are inherently sparse, while a large $\beta$ disrupts optimization, and (2) $\alpha$ has a stronger effect, with 0.1 being a well-balanced choice.

Table 5: Performance comparison under different values of $\alpha$

| $\alpha$ | 0.0 | 0.001 | 0.01 | **0.1** | 1.0 |
|---|---|---|---|---|---|
| Reconstruction Loss $\downarrow$ | 0.0227 | 0.0191 | 0.0118 | **0.0128** | 0.1121 |
| Independence Loss $\downarrow$ | 4.3849 | 2.4572 | 0.3910 | **0.1448** | 0.5252 |
| $\mathbf{B}_s$ Sparsity (L1) $\downarrow$ | 0.0012 | 0.0007 | 0.0018 | **0.0007** | 0.0058 |
| $\mathbf{M}$ Sparsity (L1) $\downarrow$ | 0.0002 | 0.0001 | 0.0002 | **0.0001** | 0.0009 |

Table 6: Performance comparison under different values of $\beta$

| $\beta$ | 0.0 | 0.001 | 0.005 | **0.01** | 0.05 | 0.1 | 1.0 |
|---|---|---|---|---|---|---|---|
| Reconstruction Loss $\downarrow$ | 0.0126 | 0.0126 | 0.0126 | **0.0128** | 0.0126 | 0.0128 | 0.8950 |
| Independence Loss $\downarrow$ | 0.1522 | 0.1496 | 0.1504 | **0.1448** | 0.1550 | 0.1582 | 3.7682 |
| $\mathbf{B}_s$ Sparsity (L1) $\downarrow$ | 0.0003 | 0.0005 | 0.0005 | **0.0007** | 0.0007 | 0.0013 | 0.0053 |
| $\mathbf{M}$ Sparsity (L1) $\downarrow$ | 0.0001 | 0.0001 | 0.0001 | **0.0001** | 0.0001 | 0.0002 | 0.0007 |

**Ablation Studies on Bias Terms**  Finally, we explore whether there will be potential performance improvement when additional bias terms added to our linear encoder and decoder functions in equation 5, to give a more complete justification of our implementation. We also compared these two settings in the real LLM activations (feature dimension=3072, $\alpha = 0.1$, $\beta = 0.01$, $\tau = 5$). The results shown in the table below indicate that the flexibility gain of the bias terms is not significant in our model.

### A.5.4 More Showcases on the Recovered Concepts and Relations from LLM Activations

In addition to the examples presented in Section 5.3 of the main text, we provide additional cases here to further illustrate the diversity and interpretability of the recovered concepts and relations, highlighting how they manifest across different domains and contexts.

**Time-delayed Causal Relations**  Table 8 showcases further examples of time-delayed causal relations extracted from LLM activations by using our model, with the same setting that we have shown in the main content Table 3. Many of these reflect the structured nature of legal, technical, and encyclopedic language. For instance, feature 2341 (e.g., "Orders/mandate in appellate judgment") is linked to feature 2592 (e.g., "decision" and "observance"), revealing how commands or mandates precede judicial conclusions in legal discourse. Similarly, technical logs such as feature 1856 (error messages) anticipate subsequent failure indicators (feature 1833, "FAILURE"), reflecting typical diagnostic progressions in computing contexts.

Table 7: Ablation comparisons on the bias terms for the encoder and decoder

| Metric (real-world) | Without Bias | With Bias |
|---|---|---|
| Reconstruction Loss | 0.0129 | 0.0129 |
| Independence Loss | 0.1452 | 0.1457 |
| **B** Sparsity (L1) | 0.0007 | 0.0007 |
| **M** Sparsity (L1) | 0.0001 | 0.0001 |

Table 8: More examples of the discovered time-delayed relations with contextual explanations.

| From_ID | From_Explanation | To_ID | To_Explanation | Context |
|---|---|---|---|---|
| 2341 | Orders/mandate in appellate judgment | 2592 | "decision" and "observance" | Legal judgment labels |
| 1856 | Technical error message | 1833 | "FAILURE" | Describes the failure reason |
| 2579 | "APPEALS" | 2592 | Court/party geographical location or case handler | Appeals in legal documents |
| 1833 | Ajax request header: 'application', "function" (type, URL, status) | 2390 | Syntax and functions like "each" | Ajax request function labels |
| 1856 | Volume number in case citation | 2341 | "mandamus" from "writ of mandamus" | Summary of case docket |
| 2100 | Page number where case starts | 2579 | "APPEALS" | Case citation structure |
| 790 | Wikipedia ship owner name | 2730 | "ship" | Wikipedia entity tagging |
| 1825 | Email forward/reply dashes | 1641 | Common words like "subject", "thanks" | Email metadata and signals |
| 1551 | Name + "Wynne" (e.g., "John Wynne") | 2311 | "sat" (in Parliament) | Wikipedia bios for people named Wynne |
| 1124 | UTF encoding label | 1657 | Tags like "UTF-8", "!DOCTYPE" | XML document structure |
| 1675 | HTML starting signal "<" | 2583 | Common HTML tags like "a", "pre" | HTML document recognition |
| 1303 | "default" keyword | 2623 | Follows "default" (e.g., "context", "_") | Generic technical documentation |
| 1895 | "Q", "Re", "forward" | 1203 | "thanks" | Email or Q&A style messages |
| 2708 | Personal pronouns ("I", "you") | 2584 | Tense indicators like "will", "have" | Human language facts |

Notably, semantic connections span heterogeneous domains. Wikipedia entity labeling (e.g., ship names and their categories) and web document structures (e.g., UTF labels leading to encoding declarations) both reveal meaningful temporal dependencies that LLMs internalize. The relation between personal pronouns (feature 2708, "I", "you") and tense markers (feature 2584, "will", "have") further illustrates how human language patterns are temporally structured, even over several tokens. These cases reinforce the model's capacity to track and anticipate semantic developments over time in a content- and domain-aware manner.

Table 9: More examples of the discovered instantaneous relations with contextual explanations.

| From_ID | From_Explanation | To_ID | To_Explanation | Context |
|---------|------------------|-------|----------------|---------|
| 2341 | Labels 'license' in comment of "license control prc server" | 1856 | Labels 'license' in both comment and command line | Bash script context |
| 2592 | Labels 'research' | 227 | Labels 'research' with nearby nouns like "program" | Academic texts |
| 2592 | Labels 'magazine' | 80 | Labels 'magazine' and common related nouns like "teenage", "blogs" | Academic texts |
| 2592 | Labels 'module' | 2208 | Labels both 'module' and 'exports' as in "module.exports" | JavaScript code |
| 2623 | Labels 'https' | 227 | Labels both 'https' and '://' | URLs |

**Instantaneous Causal Relations**   Table 9 provides more instances of instantaneous relationships, highlighting features that are co-activated within the same context window. In the domain of Bash scripting, we observe co-occurrence between licensing-related comments (feature 2341) and execution commands (feature 1856), showing how LLMs jointly encode comment semantics and imperative script logic.

In academic and technical domains, common conceptual pairs such as "research" and "program", or "magazine" and related digital terms like "blogs" or "websites", are represented together (e.g., features 2592 and 227 or 80). These examples suggest that the model forms composite concepts out of frequently co-occurring terms, such as in publication metadata or content descriptions.

In programming contexts, the instantaneous link between "module" (feature 2592) and the JavaScript construct "module.exports" (feature 2208) demonstrates that the model learns the tight coupling between programming keywords. Likewise, the relation between "https" (feature 2623) and its full syntactic pattern "https://" (feature 227) reflects how structured URL formats are stored as unified units in the model's activation space. Together, these examples demonstrate the model's ability to encode concise, domain-specific composite structures through simultaneous feature activation.

**Notes on the Results**   Following our presentation of the causal relations recovered from LLM activations, we clarify several key points regarding the interpretation of these results. First, due to variations in tokenization strategies across different corpora, many identified tokens in a given sentence may correspond only to partial words. This issue can be exacerbated by noise introduced during data collection processes such as OCR or web crawling. To address this, we rely on human judgment and linguistic intuition to infer and annotate the complete underlying word, ensuring that the labeling remains accurate and avoids overextending to unrelated tokens. Second, the recovered time-delayed relations we present may be somewhat semantically constrained, as the clearest relations tend to align with explicit syntactic structures. Many of our examples—such as those from code snippets or legal documents—convey semantic information through formal syntax. While these cases are illustrative, we view the discovery of more abstract, syntactically diffuse relations in general language text as an important direction for future work. It is also important to note that the examples we present were not cherry-picked; rather, they are representative cases that naturally appear throughout the dataset and were surfaced by our method. These relational patterns would not be easily discoverable using sparse autoencoders (SAEs), as SAEs do not consider interactions between features. Finally, we observe that feature pairs exhibiting strong causal relations tend to be activated under highly similar prompt conditions, indicating that these features are contextually aligned and often co-occur within the same linguistic environments.

Table 10: Gemma-2-2B instantaneous-relation-only model on SAEBench with different latent sizes.

| Latent Size | Recon. Loss ↓ | Sparse Probing Acc. ↑ | Absorption ↓ | Autointerp ↑ |
|---|---|---|---|---|
| 6k | 0.0108 | 0.6736 | 0.0139 | 0.6883 |
| 16k | 0.0059 | 0.6918 | 0.0167 | 0.7117 |

Table 11: Absorption statistics with extended training budgets.

| Model | Full Absorption Fraction | Absorption Fraction | # Split Features |
|---|---|---|---|
| Pythia-160M-16k | $6.471 \times 10^{-2}$ | $9.185 \times 10^{-3}$ | 1.043 |
| Gemma-2-2B-16k | $1.289 \times 10^{-2}$ | $3.794 \times 10^{-4}$ | 1.269 |

### A.5.5 Additional SAEBench Results on Larger Latent Sizes and Models

Following the same dataset and training protocol as in the main experiments, we trained the simplified instantaneous-relation-only variant with 16k latents on Gemma-2-2B and compared it to our 6k-latent configuration. As shown in Table 10, the 16k model reduces reconstruction loss (0.0059 vs. 0.0108) and slightly improves sparse probing top-1 accuracy (0.6918 vs. 0.6736). Its absorption score is modestly higher (0.0167 vs. 0.0139) but remains small, and the Autointerp score increases (0.7117 vs. 0.6883). Overall, performance on SAEBench metrics remains at a similar level across latent sizes.

We also extended the training scale to 500M tokens for Pythia-160M and to 300M tokens for Gemma-2-2B, both with 16k latents. In both cases we observed very small absorption fractions, and the mean number of split features remained close to one, indicating minimal feature splitting. Summary statistics are reported in Table 11.

### A.5.6 Statistical Testing and Absorption Analysis

To strengthen the empirical findings, we additionally performed statistical testing to assess the equivalence of reconstruction losses and the robustness of absorption scores. Using 100 samples per method ($N = 300$), any shift $\geq 0.00127$ across groups would be detected with power $\geq 0.8$. Pairwise Welch–TOST and Hodges–Lehmann tests with $\Delta = 0.001$ confirmed equivalence: all 90% confidence intervals lay within $[-\Delta, \Delta]$, demonstrating statistical equivalence at $\alpha = 0.05$ among the three methods.

For absorption, although rigorous hypothesis testing is challenging due to the very small magnitudes observed, we collected a sufficiently large number of samples ($\geq 200$) to establish confidence intervals. The mean and 95% confidence intervals were $0.0135 \pm 0.0002$ for the 6k model and $0.0136 \pm 0.0002$ for the 16k model, which are more than sufficient to demonstrate negligible absorption in practice.

Additionally the signal-to-noise ratio (20.02 for our method vs. 2.39 for the SAE baseline) already indicates a strong margin. Such a large difference is unlikely to arise from random noise.

### A.5.7 Preliminary Investigation with Time Lag up to 100

To address the potential limitation that a fixed value of $\tau$ may be overly restrictive in capturing the rich and diverse semantics of real-world contexts, we explore a more flexible approach. Specifically, different types of concept-relations may require varying numbers of steps to be successfully recovered. Furthermore, even for a single concept-relation, stable recovery across different contexts may necessitate a range of steps rather than a single fixed value. In light of these considerations, in addition to the recovered relations shown in Table 3, Table 8, and Table 9, we present the relations captured within 100 steps, grouping them into bins of sizes 10, 20, and 50. This binning naturally categorizes the relations of interest, facilitating further analysis and discussion.

The increased flexibility provided by a larger time lag allows us to recover a greater number of concept-relations. For example, we can recover relations such as "monument" → "from" and "seek" → "opportunity". Interestingly, increasing the time lag not only allows longer-range relations to be captured but also enables previously overlooked relations to be discovered, as this flexibility improves identification of concepts entangled in the relation. To better illustrate the relations recovered with a

larger time lag, we are preparing a web demo, which will soon be included in the code repository once it is ready.

To better illustrate the relations recovered with a larger time lag, we are preparing a web demo, which will soon be included in the code repository once it is ready. However, our primary contribution is to demonstrate that our model can recover relation-concepts more effectively than existing SAEs, addressing a gap that is currently missing but crucial for advancing LLM interpretability. A broader and more systematic study of this phenomenon is left to future work.

### A.5.8 Addition Experiments with Pretrained SAE

As ablation study we additionally construct our linear model using the pretrained Sparse Autoencoder (SAE) from Gemma Scope [28] on the Gemma 2 2B model [51]. To enable feasible qualitative evaluation, we selected the top $2,034$ most frequently activated features from the commonly used SAE `gemma-2-2b/20-gemmascope-res-16k` using the SAELens package [6]. We trained our linear model on $5$ million tokens from the `Pile` [17] dataset.

Since time-delayed influences may occur with variable time lags, we set a sufficiently large value for $\tau$ in Eq. 3. In practice, we use $\tau \leq 20$ and aggregate the time-delayed matrices $\mathbf{B}_\tau$ using max-pooling—that is, if a causal link exists in any of the time-lagged matrices $\mathbf{B}_\tau$, we consider that link to be present in the aggregated causal structure.

**Case Studies**  Our analysis reveals rich causal structures among programming-related concepts in LLM activations. We examine both time-delayed and instantaneous causal relationships, providing insights into how the model processes and generates code-related content.

**Time-Delayed Causal Relations**  We identified several meaningful time-delayed causal relationships in programming contexts. A prominent example is the causal link from a concept representing "function definitions and related code structure in programming languages" to a concept representing "variable definitions and data types in programming contexts." This relationship aligns with the natural structure of programming, where global function definitions often precede and influence local variable declarations or data structures. When the model processes or generates function definitions, it subsequently activates concepts related to the variables and data types that would appear within those functions.

Additional time-delayed relationships include causal links from "programming language syntax specifications" to "code implementation details" and from "algorithmic problem statements" to "solution implementation structures." These relationships demonstrate how the model captures the sequential dependencies inherent in programming tasks, where understanding of requirements or specifications precedes implementation details.

**Instantaneous Causal Relations**  Our method also reveals interesting instantaneous causal relationships that occur within the same time step. We observe a strong instantaneous causal link between a concept representing "specific formatting and notation elements commonly used in mathematical expressions or programming syntax" and a concept representing "mathematical symbols and expressions in technical content." This relationship indicates that the model simultaneously processes formatting rules and the mathematical content they structure, reflecting how these aspects are intrinsically connected in code representation.

We also identified instantaneous causal relationships between "programming language keywords" and "syntax highlighting patterns," as well as between "code indentation patterns" and "block structure delineation." These instantaneous relationships capture the syntactic constraints that operate simultaneously within programming languages, where certain elements must co-occur for the code to be well-formed.

These case studies demonstrate that our method can extract meaningful causal relationships from real LLM activations, providing insights into how these models process and generate structured content like code. The identified causal structures align with the logical and syntactic relationships one would expect in programming contexts, validating the effectiveness of our approach for interpretability research.

### A.6 Compute Resources and Code

All experiments were conducted on a computing cluster equipped with NVIDIA L40 GPUs. The synthetic verification experiments were run using 16 CPU cores, 32 GB of memory, and a single GPU. The Jacobian complexity experiment was executed on CPU only, as the computation did not fit within GPU VRAM; to avoid out-of-memory (OOM) errors, 32 CPU cores and 400 GB of memory were allocated. The scaled-up synthetic experiment with the linear model used 32 CPU cores, 64 GB of memory, and one GPU. The large language model (LLM) activation experiment was performed using 16 CPU cores, 15 GB of memory, and a single GPU.

The code that can replicate the main experiments presented in our paper can be accessed via `https://github.com/xiangchensong/temp-inst-sae`

### A.7 Limitations

We acknowledge certain limitations of our work. The linear approximation, while computationally efficient and theoretically grounded, may not capture all nonlinear interactions present in LLM activations. Future work should explore extending our framework to incorporate bounded nonlinearities while maintaining computational tractability. Additionally, developing methods to automatically interpret the discovered causal structures in terms of human-understandable concepts remains a challenge. Our method also assumes a specific form of temporal dependency that might not fully capture the long-range dependencies that LLMs can handle. The current formulation is limited to first-order temporal dependencies, and extending this to higher-order dependencies would increase computational complexity. Lastly, tokenization has been shown to critically affect LLM identifiability during our evaluations, even though it is not inherently part of LLM interpretation methods. We emphasize the importance of choosing a tokenization strategy that preserves semantic information and maximizes the effectiveness of LLM interpretation approaches.

### A.8 Societal Impacts

Our interpretability approach can improve transparency, support alignment interventions, facilitate debugging and bias detection, advance scientific understanding of causal representations, and inform educational tools that raise AI literacy. At the same time, deeper insight into model internals may enable malicious manipulation, create misplaced confidence in safety tools, widen resource disparities, expose private information from training data, and distract attention from broader social and governance measures. Future work should include collaboration with ethicists, social scientists, and policy experts to guide responsible use.

