# OpenReview forum: "LLM Interpretability with Identifiable Temporal-Instantaneous Representation"
_NeurIPS.cc/2025/Conference — NeurIPS 2025 poster_

### Official Review · Reviewer_KYay · 2025-06-21

**Clarity:** 1
**Significance:** 2
**Originality:** 3
**Rating:** 3
**Confidence:** 3

**Summary:**

This paper introduces a framework that integrates sparse autoencoders with causal representation learning to interpret LLMs. It captures temporal and instantaneous causal relationships in model activations, provides theoretical guarantees, and scales to LLMs' high-dimensional concept space, enabling the discovery of meaningful and interpretable features.

**Questions:**

1. Will the performance drop after the method is applied?

**Ethical Concerns:**

["NO or VERY MINOR ethics concerns only"]

**Final Justification:**

Thanks for the rebuttal. The concerns are from the time series for the synthetic experiments and human-readable interpretability. First, the authors use the time series to demonstrate a language problem. It is not reasonable. Second, the authors do not provide the conditions under which the assumption (Equation 2) holds. Third, the goal of interpretability is to understand the LLM. Since the authors say, "From the LLM perspective, concept-relations embedded in LLM activations are intrinsically tied to real natural language semantics, which are highly complex. Exactly recovering the complete set of concepts and relations remains an open goal for the entire mechanistic interpretability community. Hence, it is challenging to construct synthetic data that covers all concept-relations in natural languages. We interpret your suggestion as asking whether we can use controlled natural language data with some known concept-relations to demonstrate that such phenomena in LLMs can be recovered by our model." It can be evidence that the LLM as a judge is a better explainer to understand how the LLM works.

**Limitations:**

This paper does not mention the limitation. As far as I know, the Interpretability will harm the performance.

**Paper Formatting Concerns:**

No concerns

**Quality:**

2

**Strengths And Weaknesses:**

Strengths:
1. This paper introduces the use of the linear representation hypothesis to provide theoretical guarantees for sparse autoencoders, bridging a gap between empirical interpretability and formal causal representation learning.
2. It effectively combines sparse autoencoders with temporal causal modeling, enabling the discovery of both time-delayed and instantaneous conceptual relationships in LLM activations—something prior methods could not scale to or capture.

Weakness:
1. The paper's objective lacks clarity—while it aims to enhance LLM interpretability, the connection between the linear representation hypothesis and LLMs is not well established. In particular, it remains unclear how variables like $x_t$ and $z_t$ are concretely defined within the LLM context.
2. The use of synthetic time series data undermines the relevance of the evaluation, as it does not accurately reflect the structural or statistical properties of LLM activations, which the method ultimately targets.
3. The paper lacks comparison with strong baseline methods and omits quantitative evaluation metrics, making it difficult to assess the practical effectiveness or improvements of the proposed approach.

---

> ### Author Rebuttal · Authors · 2025-07-30
>
> Thank you for acknowledging our contribution between empirical interpretability and formal causal representation learning and, more importantly, the scalability issue we address. We now address your concerns one by one.
>
> >**W1. The paper's objective lacks clarity—while it aims to enhance LLM interpretability, the connection between the linear representation hypothesis and LLMs is not well established. In particular, it remains unclear how variables like $x_t$ and $z_t$ are concretely defined within the LLM context.**
>
> **AW1:** The Linear Representation Hypothesis is well-established for LLMs through prior work [A], which demonstrated that LLM activations can be effectively modeled using linear methods.
>
> To clarify our notation: $\mathbf{x}_t \in \mathbb{R}^m$ represents the activation vector at layer $L$ for token position $t$, where $m$ is the model's hidden dimension. The latent variables $\mathbf{z}_t \in \mathbb{R}^n$ capture $n$ interpretable concepts that generate these activations through the linear decoder $A$.
>
> **Specifically within the LLM context (line 73-90)**: Given input tokens $(v_1, v_2, \ldots, v_T)$, we extract activations $\mathbf{x}_t$ from a target transformer layer (e.g., layer 12 of 32) at each position $t$. Our model decomposes these activations into interpretable concepts $\mathbf{z}_t$ (Eq. 1, 2) that exhibit temporal dependencies $B$ (how concepts evolve across tokens) and instantaneous relationships $M$ (how concepts interact within each token). We will add this clarification to Sec. 2 with specific examples to demonstrate this mapping concretely.
>
> [A] Park, Kiho, Yo Joong Choe, and Victor Veitch. "The linear representation hypothesis and the geometry of large language models." ICML (2024).
>
> >**W2. The use of synthetic time series data undermines the relevance of the evaluation, as it does not accurately reflect the structural or statistical properties of LLM activations, which the method ultimately targets.**
>
> **AW2:** This synthetic-to-real validation paradigm is a widely accepted methodology in causal representation learning and related fields. We establish theoretical identifiability guarantees for learning concepts and relations. Since ground truth is unavailable in real-world scenarios, this identifiability cannot be directly testable on real LLM data. Therefore, controlled synthetic experiments with known ground truth are essential for verifying both our theoretical analysis and algorithm implementation.
>
> Once we establish confidence that our theoretical guarantees translate into practical performance, subsequent experiments on real LLM activations (Sections 5.2 and 5.3) demonstrate the method's relevance to authentic data. We have revised the manuscript to clarify this rationale and emphasize that synthetic studies complement and validate, rather than replace, our evaluations on real LLM data.
>
>
> >**W3. The paper lacks comparison with strong baseline methods and omits quantitative evaluation metrics, making it difficult to assess the practical effectiveness or improvements of the proposed approach.**
>
> **AW3:** As acknowledged by other reviewers and mentioned on pages 6–7, our work is the first to address temporal and instantaneous relations at the LLM activation scale. We extensively searched for baselines from the causal representation learning (CRL) literature, but to our knowledge, no existing method can efficiently handle the high-dimensional activation data produced by LLMs (Figure 5). We would appreciate clarification on any specific missing baselines you have in mind.
>
> We agree that quantitative analysis can strengthen our approach. While no existing benchmark captures causal representation learning for concept relations in LLM activations, we demonstrate our method's effectiveness through two evaluations: (1) testing on SAEBench [A] to show our model matches mainstream SAEs' performance on standard SAE tasks, and (2) presenting quantitative results on a new task tailored to our unique contribution—recovering concept-to-concept relations.
>
> **(1) SAEBench:** For fair comparison, we trained TopK and ReLU SAEs with hyperparameters (feature dimension, training cost, etc.) similar to our model:
> | Model | Reconstruction Loss $\downarrow$ |Sparse Probing Acc. $\uparrow$ |Absorption Score $\downarrow$|Autointerp Score $\uparrow$|
> |-|-|-|-|-|
> |ReLU SAE| 0.0110|0.6555|0.0141|0.6791|
> |TopK SAE| 0.0097| 0.7141|0.0280|0.6822|
> | Ours  | 0.0108   |0.6736|0.0139|0.6883|
>
> These results demonstrate that while not specifically designed for SAE tasks, our model achieves comparable performance on these tasks.
>
> **(2) Concept Relation Recovery:** Furthermore, building on the experiments in [B], we **quantitatively** evaluate our model on concept-to-concept relation extraction.
>
> *Data preparation:* We constructed two contrastive collections of texts drawn from the Pile dataset. One collection consists of legal documents, which often follow structured syntactic formats from start to end, leading to stable temporal concept patterns. The other contains non-legal texts without such structure. We hypothesized that only legal texts would yield meaningful relational patterns, and we directly tested whether the model can successfully recover such patterns.
>
> *Baseline:* Although there is no directly applicable baseline, we leveraged standard SAEs we had trained above to serve as our baseline method. Since SAEs themselves cannot capture the concept-to-concept relations, we train a regression model to find temporal relation coefficient matrices $\tilde{B}$s by solving the following regression task:
>
> $$\text{min}\_{\tilde{B}\_{\tau}} ||\mathbf{z}\_t - \sum_{\tau} \tilde{B}\_{\tau} \mathbf{z}_{t-\tau}||\_{2}.$$
>
> *Evaluation Pipeline:* We calculate the concept recovery score by first getting the top fired feature index pair $(i,j)$ related to legal context and the corresponding entry $B_{i,j}$ in the aggregated temporal relation coefficient matrices. We then calculate the relation recovery score, similar to a signal-to-noise ratio, by: $\texttt{relation recovery score} = \frac{B_{i,j}}{\sigma(B)}$, where $\sigma(B)$ denotes the standard deviation of matrix $B$. Such ratio indicates the extent that the target concept relation entry in the matrix is more significant than a random noise; the larger the score is the more significant the relation recovery.
>
> From the table below, we notice a significant difference, showing that our method can successfully recover the relation.
>
> |Method|Relation Recovery Score $\uparrow$|
> |-|-|
> |SAE+regression|2.39|
> |Ours|20.02|
>
> These two experiments together outline our improvement over existing SAE methods by first showing our matching performance on SAE tasks and then demonstrating that our method can successfully extract meaningful concept-to-concept relations, which cannot be handled by SAEs.
>
> [A] Karvonen, Adam, et al. "SAEBench: A Comprehensive Benchmark for Sparse Autoencoders in Language Model Interpretability." Forty-second International Conference on Machine Learning, 2025
>
> [B] Joshi, Shruti, Andrea Dittadi, Sébastien Lachapelle, and Dhanya Sridhar. "Identifiable steering via sparse autoencoding of multi-concept shifts." arXiv preprint arXiv:2502.12179 (2025).
>
> >**Q1 Will the performance drop after the method is applied?**
>
> **AQ1:** We assume the performance you mentioned here refers to the language model's original metrics, such as downstream task accuracy and perplexity. Our interpretability method only reads transformer layer activations; it does not modify any weights, architecture, or forward computations. The model’s task performance therefore remains unchanged. While some interpretability approaches do fine‑tune model weights and can reduce performance, our analysis is not in this category. If the reviewer is considering a different notion of performance, we would appreciate further clarification.
>
>
> >**L1. This paper does not mention the limitation. As far as I know, the Interpretability will harm the performance.**
>
> **AL1:** We kindly refer the reviewer to Appendix **Section A.5** for the limitations discussion. The performance issue is discussed in the previous question.

---

> > ### Comment · Reviewer_KYay · 2025-08-04
> > **Reply to Authors**
> >
> > Thanks for the rebuttal.
> >
> > 1. In AW2, time series data are distributed in a continuous space, while language data is discretely distributed. Using synthetic time series data for LLM Interpretability is not reasonable. It would be better to create an artificial language dataset to verify the idea. Besides, in AW1, the $B$ and $M$  are temporal dependencies and instantaneous relationships. It is better to use synthetic data to demonstrate the phenomenon in LLM.

---

> > > ### Author Response · Authors · 2025-08-04
> > >
> > > Thank you for your additional comments, and please see our responses to the two questions respectively as shown below:
> > >
> > > **Regarding the continuous vs. discrete nature of LLM activation data:**
> > >
> > > Thank you for raising this important distinction. While we agree that text tokens are discrete, our method operates on **LLM activations**, not on the tokens themselves. These activation vectors exist in a continuous high-dimensional space (e.g., $\mathbb{R}^{m}$), even though they are computed at discrete time steps corresponding to token positions. More precisely, such activation data should be characterized as continuous-valued observations sampled at discrete time steps, which is exactly the setting that our synthetic time series experiments are designed to validate. We would also like to pinpoint that, recent work has also empirically demonstrated that LLMs learn continuous representations of language [A], further supporting our experimental design choice.
> > >
> > > **Regarding synthetic demonstration of temporal and instantaneous relations in LLMs:**
> > >
> > > We would like to first outline our understanding of your concern. The comment appears to ask for demonstrating concept-to-concept relations of real LLMs using synthetic data. But we suspect it may not lead to meaningful arguments for the following reasons:
> > >
> > > (1) In the CRL literature, synthetic data refers to using predefined data generation functions (such as Eq. 1-3 in our case) with known ground truth parameters (such as B and M matrices). Since the concept relations are predefined and encoded in the synthetic data generation process, demonstrating such relations would be circular—they are known by construction.
> > >
> > > (2) From the LLM perspective, concept-relations embedded in LLM activations are intrinsically tied to real natural language semantics, which are highly complex. Exactly recovering the complete set of concepts and relations remains an open goal for the entire mechanistic interpretability community. Hence, it is challenging to construct synthetic data that covers all concept-relations in natural languages. We interpret your suggestion as asking whether we can use controlled natural language data with **some** known concept-relations to demonstrate that such phenomena in LLMs can be recovered by our model.
> > >
> > > We would appreciate if the reviewer could provide further context to confirm whether our interpretation is correct. If so, we have already addressed this concern in our **AW3 (2) Concept Relation Recovery** experiment, where we constructed a controlled dataset using legal documents from the Pile dataset. These documents exhibit structured concept relations due to their formal syntactic patterns, contrasted with non-legal texts lacking such structure. Our experimental results demonstrate that our method successfully recovers this meaningful concept-to-concept relation, achieving a Relation Recovery Score of 20.02, which is quite significant compared to the baseline method using SAE with a post-hoc regression layer, which achieves only 2.39.
> > >
> > > (3) Furthermore, as demonstrated in the case studies presented in Section 5.2 of our manuscript, we have successfully identified meaningful temporal and instantaneous relations from LLM activations. For instance, we observed instantaneous relations between the feature `Month (e.g., March)` and the feature `Full date (e.g., March 23, 2000)`. For temporal relations, we identified edge from the feature `Adjectives of nationality (e.g., Japanese)` to the feature `Nouns that follow nationality (e.g., brands, names)`. For additional demonstrations, please refer to Table 1 in Section 5.2 and Tables 2 and 3 in Appendix A.3.3.
> > >
> > > Additionally, the existence of relations between concepts has been verified in the mechanistic interpretability literature [B,C]. Therefore, we reasonably believe this phenomenon exists in LLMs and can be recovered by our model.
> > >
> > > ---
> > > We hope our response addresses your concerns. We look forward to your further comments and feedback, and we are happy to respond to them.
> > >
> > >
> > > [A] Marro, Samuele, et al. "Language Models Are Implicitly Continuous." The Thirteenth International Conference on Learning Representations, 2025.
> > >
> > > [B] Ameisen, et al., "Circuit Tracing: Revealing Computational Graphs in Language Models", Transformer Circuits, 2025.
> > >
> > > [C] Lindsey, et al., "On the Biology of a Large Language Model", Transformer Circuits, 2025.

---

> > > > ### Comment · Reviewer_KYay · 2025-08-06
> > > > **Reply to Comment**
> > > >
> > > > Thanks for the updates.
> > > >
> > > > 1. Since the discrete tokens can transform into continuous activation data, it is better to use language data directly rather than time series data. Language data is human-readable, while time series data is not. Since this paper utilizes SEM to enhance interpretability, it is beneficial to provide a demo to demonstrate the suitability of the idea.
> > > >
> > > > 2. The goal of this paper is to improve the LLM's Interpretability. The SEM and SAE are used to extract the concepts and the relationship between concepts, if I understand correctly. However, the authors claim " concept-relations embedded in LLM activations are intrinsically tied to real natural language semantics, which are highly complex". In other words, the concepts extracted by SAE may not be human-readable.  It is not aligned with the Interpretability.
> > > >
> > > > Thanks again for the updates. I would keep the score. I will respect the decision of Area Chairs.

---

> > > > > ### Author Response · Authors · 2025-08-06
> > > > > **Response to your further comments (1/2)**
> > > > >
> > > > > Thank you for your thoughtful feedback and continued engagement with our work. We also appreciate your openness to respecting the Area Chair's final judgment, and we value the opportunity to address both of your concerns with additional clarification. Please see our detailed responses to the two points you brought up, respectively.
> > > > >
> > > > > ## Point 1
> > > > >
> > > > > From our understanding, your first point mainly concerns the rationale for our experimental design. We have identified three aspects of your concern, which we will address one by one.
> > > > >
> > > > > 1. LLM activation space is continuous: We're pleased to see that in your recent response, you acknowledged that *"discrete tokens can transform into continuous activation data."* This aligns with our understanding and that of the mechanistic interpretability community: while language tokens are discrete, LLM activations—the space on which interpretation models operate—are continuous.
> > > > >
> > > > > 2. Synthetic experiment—time-series: Given the above, the LLM interpretation models are ultimately operating on continuous space. Therefore, our time-series synthetic experiment exactly aligns with this requirement. Furthermore, please allow us to explain the necessity of composing such time-series synthetic experiments. As our proposal is grounded in CRL, which essentially requires identifiability to ensure that the recovered relations are not trivial, which we believe is needed for interpretation models for LLMs. Based on the theoretical results of the methodology, how can we further verify that the algorithm implementation does embed this property? We need to conduct synthetic experiments, which are not yet designed to obtain the LLM concepts and their relations, but specifically to ensure the output of the algorithm is close to a **precisely defined ground truth**. This is exactly what we have done in the time-series synthetic experiments, to gain confidence before we apply our method on LLM activations.
> > > > >
> > > > > 3. Semi-synthetic experiment—steering vectors and concept-relation recovery: We completely understand your concern that the previously explained synthetic experiments might not be directly used to see the concepts and relations from LLM activations, since they work for a different purpose. Fortunately, this set of semi-synthetic experiments tightly aligns with your suggested rationale. The semi-synthetic experiments operate directly on human-readable language token data (from which continuous activations are extracted) and are used to test whether our method can recover concept relations. Ideally, if one could create a universal ground truth including **all** the concepts and relations from natural language token data, then we could directly evaluate our method in the same way as the synthetic experiment on the time-series data. However, creating such an ideal ground truth is infeasible. This is because the concepts and relations in natural language are highly complex; for instance, even if you control a set of concepts and relations in your semi-synthetic data creation process, you inevitably will introduce other spurious relations (e.g., grammar and pattern in language). So, the best we can do is to construct some semi-synthetic datasets which we believe at least cover the concepts and relations **of interest**. Then, in the evaluation stage, we test if the aforementioned concept relations can be recovered. We did conduct such experiments and presented them in the steering vector experiment and concept relation recovery experiment. (Please kindly refer to our **AW2, W4** to Reviewer `gsa4`, this type of semi-synthetic experiment was also suggested by other reviewers and is a well-recognized approach in the mechanistic interpretability community.)
> > > > >
> > > > > In summary, we believe your concern regarding Point 1 has been thoroughly addressed in our previous responses and rebuttal. Please let us know if any aspects remain unclear, and we are eager to address them.

---

> > > > > > ### Author Response · Authors · 2025-08-06
> > > > > > **Response to your further comments (2/2)**
> > > > > >
> > > > > > ## Point 2
> > > > > >
> > > > > > Thank you for sharing your understanding of the goal of our manuscript. Indeed, we aim to recover the concepts and the concept-relations from LLM activations for LLM interpretability. Beyond this common ground, we understand your second concern is primarily about the relationship between interpretability and readability. We understand your concern, and please see our responses below.
> > > > > >
> > > > > > 1. First, to avoid any misunderstanding on this critical point, we want to clarify that we never intended to suggest the recovered results are not human-readable. As mentioned in Point 1, our argument *"concept-relations embedded in LLM activations are intrinsically tied to real natural language semantics, which are highly complex"* was intended to illustrate the infeasibility of creating a ground truth for real language that precisely and universally covers **all** concept relations. This challenge necessitates our semi-synthetic experimental design.
> > > > > >
> > > > > > 2. Second, we appreciate your emphasis on aligning the readability of our results with LLM interpretability. On this point, please allow us to elaborate upon both parts of our evaluation. The **qualitative analysis** in Section 5.2 presents clear and human-readable examples of meaningful, interpretable concepts and relations extracted from LLM activations. Furthermore, the semi-synthetic experiments mentioned previously allow for a human-readable evaluation of the concepts and relations our method recovers. In addition to those qualitative results, as also mentioned by the reviewers, we conducted **quantitative validation** on SAEBench. The high AutoInterp scores, in particular, provide objective measures confirming that our method successfully extracts human-readable concepts. Such evaluation approaches are widely adopted in the mechanistic interpretability community.
> > > > > >
> > > > > > In summary, we believe there may have been a misunderstanding of our argument; we did not intend to imply that the concepts we recovered are not human-readable. The readability and interpretability of our results were clearly showcased in our experiments and previous responses, and we wish to re-emphasize that point here.
> > > > > >
> > > > > > ---
> > > > > >
> > > > > > We appreciate your spirit of "seeking common ground while shelving differences" and hope our additional clarification resolves this potential misunderstanding. If there are still pending concerns, please kindly let us know and we are eager to respond to them.

---

> ### Author Response · Authors · 2025-08-08
> **Please let us know if you have further questions**
>
> Dear Reviewer `KYay`,
>
> Thank you again for reviewing the paper and for taking the time to consider our updates. We sincerely appreciate your observations and hope for the opportunity of further engagement to avoid misunderstandings. If there are still aspects that remain unclear, we would be very grateful for the opportunity to provide further clarification to illustrate our approach.
>
> Sincerely,
>
> Authors of Paper 7618

---

### Official Review · Reviewer_WPzT · 2025-06-24

**Clarity:** 3
**Significance:** 3
**Originality:** 3
**Rating:** 6
**Confidence:** 4

**Summary:**

This paper applies ideas from causality to improve SAEs by explicitly modeling relations between latents in the SAE within a single token, and relations between tokens in a series. The work provides theoretical guarantees about the soundness of this method when applied to data generated following the linear representation hypothesis (LRH). This method improves the scalability of casuality methods to work with high-dimensional LLM activations and features. The method is validated experimentally on synthetic examples, and qualitatively on data generated from a small LLM.

**Questions:**

- Can this method model hierarchical relationships between latents? E.g. if z1 can only fire if z2 is firing, but z1 can on its own, can this be captured by this model? This would correspond to something like "z1 = noun" and "z2 = dog", since "dog" is a "noun". Hierarchical relationships seem to be break standard SAEs (causing feature absorption), so if this method can handle that it would be huge.
- Can this method only handle relations between 2 latents at a time? e.g. if a latent is and AND of 3 latents, can this method capture that?
- In Figure 3, is (e) the correlation between A and $\hat{A}$ not shown? Also, why is estimated $\hat{B}$ not identical to B? It seems like the method should be able to perfectly recover B in this synthetic setting?
- In Equation 7, why is there no nonlinearity (presumably ReLU) in the encoder? Should there also be encoder and decoder bias terms?
- It seems like the $\hat{A}$ matrix is equivalent to an SAE decoder dictionary - can this matrix be directly used as an SAE? Would you expect this method to learn a more correct dictionary than standard SAE training methods?
- In the paper the trained LLM models are still very small - is it feasible to train a more standard size version of this method to compare with SAEs, e.g. 32k+ latents?
- Can this method be directly compared to LLM SAEs using standard SAE benchmarks like sparse probing, or other metrics from SAEBench?

**Ethical Concerns:**

["NO or VERY MINOR ethics concerns only"]

**Final Justification:**

Applying insights from causality to SAEs is an important contribution to the field. If true that this approach also has the side-effect of improved handling of hierarchical features in SAEs without degradation in performance, as their results seem to imply, I think this is an extremely important work that will rapidly get adopted in the field.

**Limitations:**

yes

**Paper Formatting Concerns:**

no concerns

**Quality:**

3

**Strengths And Weaknesses:**

### Strengths

The method provided can give insights into relations between SAE latents both within and across tokens, which is valuable for understanding how latents relate to each other. The method provided has better scalablility characteristics than existing causality based methods, allowing this method to scale to LLMs. Explicitly borrowing ideas from causality is valuable to the interpretability community, given the parallels between LLM interpretability and causality.

### Weaknesses

- While more scalable than existing causality methods, the method seems to still have poor scalability compared to standard SAEs. The paper only evaluates dictionaries up to size 6144, which is miniscule by LLM SAE standards.
- The paper also does not compare this technique directly with LLM SAEs (e.g. on interpretability benchmarks like SAEBench), so it's hard to gauge if / how much this improves on standard SAEs in practice.
- The temporal relations are modeled by fixed token distances, which seems inflexible and likely cannot capture real syntac relationships between tokens well.

---

> ### Author Rebuttal · Authors · 2025-07-30
>
> We sincerely thanks for your insightful feedback. We're glad you appreciate the strengths in our novelty for being the first to scale causal representation learning to LLM activations, and we appreciate your constructive suggestions for strengthening our contribution. We address each point below.
>
> >**W1,Q6 While more scalable than existing causality methods, the method seems to still have poor scalability compared to standard SAEs. The paper only evaluates dictionaries up to size 6144, which is miniscule by LLM SAE standards. In the paper the trained LLM models are still very small - is it feasible to train a more standard size version of this method to compare with SAEs, e.g. 32k+ latents?**
>
> **AW1,Q6:** Thank you for this important question about scalability. While the reported dictionary size of 6,144 was indeed limited by GPU memory constraints, as you also mentioned in your question, this dimension already represents a significant advancement in terms of dimensionality for identifiable representation learning methods. Our experimental results demonstrate that even with this dictionary size, we can extract meaningful interpretations of concept-to-concept relations.
>
> For larger dictionary sizes, our approach can be parallelized using tensor parallelism or data parallelism to accommodate greater scale. However, this would require substantial refactoring of existing open-source infrastructure tools like `TransformerLens` and `dictionary_learning` to support multi-GPU training, which falls well outside the scope of our current work. We acknowledge this limitation and consider scaling to 32k+ latents an important direction for future research.
>
> >**W2,Q7 The paper also does not compare this technique directly with LLM SAEs (e.g. on interpretability benchmarks like SAEBench), so it's hard to gauge if / how much this improves on standard SAEs in practice. Can this method be directly compared to LLM SAEs using standard SAE benchmarks like sparse probing, or other metrics from SAEBench?**
>
> **AW2,Q7:** Thanks for the suggestion. Our main contribution is the recovery of temporal and instantaneous concept‑to‑concept relations, which is not reflected in SAE benchmarks. But we do agree with the reviewer that making a comparison on established benchmarks for SAEs will strengthen our position. For a fair apples-to-apples comparison, we trained two baseline SAEs (ReLU and TopK) under similar hyperparameters and evaluated all three methods on a range of SAEBench metrics, including sparse probing. Since we did not modify the SAE objective, we expect our model to perform on par with existing SAEs on SAEBench tasks, which is confirmed by the table below.
>
> |Model|Reconstruction Loss $\downarrow$|Sparse Probing Top 1 Accuracy $\uparrow$|Absorption Score $\downarrow$|Autointerp Score $\uparrow$|
> |-|-|-|-|-|
> |ReLU SAE|0.0110|0.6555|0.0141|0.6791|
> |TopK SAE|*0.0097*|*0.7141*|0.0280|0.6822|
> |Ours|0.0108|0.6736|*0.0139*|*0.6883*|
>
>
> >**W3. The temporal relations are modeled by fixed token distances, which seems inflexible and likely cannot capture real syntac relationships between tokens well.**
>
> **AW3:** We totally agree that fixed limited token distances might not be the most ideal way to capture the rich semantic and syntactic relationships between tokens. However, it is worth pointing out that even though the distance is limited, it still successfullly reveals some meaningful temporal relations (together with instantaneous relations) among the concepts as we have shown in the real-world experiments.
> We further conducted the analysis for longer temporal relations ($\tau=100$). Due to the rebuttal policy, we may not show the visualization results. We will add those results to our appendix to more comprehensively showcase the interpretability our proposal may achieve.
>
> >**Q1. Can this method model hierarchical relationships between latents? E.g. if z1 can only fire if z2 is firing, but z2 can on its own, can this be captured by this model? This would correspond to something like "z1 = noun" and "z2 = dog", since "dog" is a "noun". Hierarchical relationships seem to be break standard SAEs (causing feature absorption), so if this method can handle that it would be huge.**
>
> **AQ1:** Yes, it can! The graph for instantaneous relations can naturally model the hierarchical relationships among concepts. Our case study for instantaneous relations `Month (e.g. March)` to `Full date (e.g. March 23, 2000)` is exactly such a case, `Month` can be fired on its own, but if `Full Date` is fired, then `Month` will also be fired. Such an advantage is also captured quantitatively by the better feature absorption score in the table above.
>
> >**Q2. Can this method only handle relations between 2 latents at a time? e.g. if a latent is and AND of 3 latents, can this method capture that?**
>
> **AQ2:** Thank you for this great question. Indeed, the relations recovered by our method are represented in matrices where each entry reflects pairwise relations. But they can still support the inference of higher-order logical relations, even though it's not immediately apparent. For example, if a concept activates only when multiple other concepts are active (e.g., an AND of three latents), this can be inferred by analyzing joint activation patterns over many observations. By thresholding and binarizing concept activations, one can construct truth tables and recover logical relations such as AND, OR, or XOR. While not explicitly modeled in the relation matrices, such higher-order dependencies are reflected in the data and can be recovered through post-hoc analysis.
>
> >**Q3. In Figure 3, is (e) the correlation between A and $\hat{A}$ not shown? Also, why is estimated $\hat{B}$ not identical to B? It seems like the method should be able to perfectly recover B in this synthetic setting?**
>
> **AQ3:** Thanks for pointing this out, our identifiability was established for latent variables $\mathbf{z}_t$ (up to permutation and scaling) and their relations $B$ and $M$, hence the exact value matching for $A$ and $\hat{A}$ is not required and we allow some indeterminacy for $A$. Below is an example of the ground truth $A$ and our estimated $\hat{A}$, and $\mathbf{z}_t$ for this setting is already identifiable (MCC > 0.99).
>
> $A$ = [[ 0.38,  0.92, -0.02],[-0.67,  0.29,  0.68], [ 0.64, -0.25,  0.73]]
>
> $\hat{A}$ = [[-3.38, -7.56, 0.22],[ 5.34, -2.63, -5.63],[-5.22,  1.85, -6.01]]
>
> Despite the theoretical guarantee for $B$, in practice, the perfect recovery requires optimization using an L0 sparsity penalty, which is known to be non-differentiable. In the literature, the L1 alternative is widely adopted. That explains the small difference in the estimated $B$ while it can still effectively preserve most of the structure information.
>
> >**Q4. In Equation 7, why is there no nonlinearity (presumably ReLU) in the encoder? Should there also be encoder and decoder bias terms?**
>
> **AQ4: Nonlinearity**: Thanks for pointing this out! Yes, in real-world LLM experiments, we use **top-k** activation in the encoder, following mainstream SAE architectures, which is non-linear. We use simple affine transformations in synthetic experiments for theoretical tractability.
>
> **Bias terms**: Yes, there should be bias terms for more modeling flexibility. We modified the equation to include them. We also compared these two settings in the real LLM activations (feature dimension=3072, $\alpha=0.1$, $\beta=0.01$, $\tau=5$). The results shown in the table below indicate that the flexibility gain of the bias terms is not significant in our model.
>
> |Metric (real-world)|Without Bias|With Bias|
> |-|-|-|
> |Reconstruction Loss|0.0129|0.0129|
> |Independence Loss|0.1452|0.1457|
> |$B$ Sparsity (L1)|0.0007|0.0007|
> |$M$ Sparsity (L1)|0.0001|0.0001|
> >**Q5. It seems like the $\hat{A}$ matrix is equivalent to an SAE decoder dictionary - can this matrix be directly used as an SAE? Would you expect this method to learn a more correct dictionary than standard SAE training methods?**
>
> **AQ5:** Yes, this can be used as an SAE. As shown in **AW2,Q7**, the performance on SAE tasks for our method is at least on-par with existing SAEs. We acknowledge that there may not be a universally more correct dictionary; we could say our learned dictionary is relation-aware, which can be more reasonable and capable than standard SAEs.

---

> > ### Comment · Reviewer_WPzT · 2025-08-01
> >
> > Thank you for the detailed reply, and for running SAEBench. If you can indeed model hierarchical relationships directly this is incredible! If you're able to handle hierarchy properly, then even without the temporal component of this work that is potentially a major breakthrough. Presumably the scalability issues you have would be lessened if you only focus on instantaneous features and ignore temporal features?
> >
> > > Such an advantage is also captured quantitatively by the better feature absorption score in the table above.
> >
> > I don't think SAEs that are so narrow would display much absorption anyway - usually you need ~12k+ latents in my experience before it really starts showing up - but it's great that you ran this regardless.
> >
> > I will maintain my score of 5. That being said, if you are able to train a larger (16k+ latent) SAE on Gemma-2-2b (optionally ignoring the temporal causality component of your work and handling only instantaneous hierarchy), and can show that you still have negligible absorption, while keeping high scores on other SAEBench metrics I would be happy to raise my score further to a 6.

---

> > > ### Author Response · Authors · 2025-08-02
> > >
> > > Dear Reviewer WPzT,
> > >
> > > We are so grateful for your timely feedback and insightful suggestions. We have been actively working to obtain new results based on the setting you suggested (Gemma-2-2b with 16k hidden dimensions), to enable a more direct and informative comparison with SAEs. We look forward to sharing these updated results for your further feedback once they are available (hopefully in 12 hours).
> > >
> > > Sincerely,
> > > Authors of Paper 7618

---

> ### Author Response · Authors · 2025-08-02
>
> Thank you for your timely and valuable feedback and suggestions. Following your recommendations, we trained our simplified instantaneous-relation-only model with 16k latents on Gemma-2-2B using the same dataset. The results are presented below:
>
> |Latent Size|Reconstruction Loss $\downarrow$|Sparse Probing Top 1 Accuracy $\uparrow$|Absorption Score $\downarrow$|Autointerp Score $\uparrow$|
> |-|-|-|-|-|
> |6k|0.0108|0.6736|0.0139|0.6883|
> |16k|0.0059|0.6918|0.0167|0.7117|
>
> The 16k model exhibits a slightly higher absorption score (0.0167) compared to our reported 6k model, but this value still seems negligible. Performance on other SAEBench metrics remains at the same level.
>
> We are wondering whether these experimental results would help answer your questions properly. If you have any further feedback, please kindly let us know, and we hope for the opportunity to respond to it.

---

> > ### Comment · Reviewer_WPzT · 2025-08-04
> >
> > Apologies for the slow reply, thank you for running this! What is the L0 of the 16k SAE?

---

> > > ### Author Response · Authors · 2025-08-04
> > >
> > > Thank you for your response! The L0 is 100 as we apply TopK with k=100 in the 16k SAE. Please kindly let us know if you have any further feedback, we would be happy to address it.

---

> > > > ### Comment · Reviewer_WPzT · 2025-08-05
> > > >
> > > > That does seem very impressive! Do you also have the absorption fraction score and number of split features for these SAEs? I think SAEBench should report those too (no worries if not). Also, what layer of Gemma-2-2b is this SAE trained on, and how many training tokens? I'm also assuming this is trained on the residual stream of the LLM? Sorry I should have asked this earlier.

---

> > > > > ### Author Response · Authors · 2025-08-06
> > > > >
> > > > > We're grateful for your enthusiasm about these new results and your thorough engagement with our work!
> > > > >
> > > > > Absolutely! We are happy to answer your questions and share those details about the experiment setup:
> > > > >
> > > > > We performed the training on the activations of the 8th layer of Gemma-2-2B with hook name: `blocks.8.hook_resid_post`. We trained on 50M tokens from the Pile dataset on the residual stream as indicated in the hook name.
> > > > >
> > > > > **Additional SAEBench Metrics:**
> > > > >
> > > > > As requested, here are the absorption fraction scores and the number of split features:
> > > > >
> > > > > |Latent Size|Mean Absorption Fraction Score|Mean Number of Split Features |
> > > > > |-|-|-|
> > > > > |6k|$5.176 \times 10^{-3}$|$1.043$|
> > > > > |16k|$2.514\times 10^{-4}$|$1.346$|
> > > > >
> > > > > We hypothesize that the negligible absorption stems from our architecture: in addition to enforcing sparsity, we also model concept-to-concept relations and enforce the residual independence noise condition—such an approach could help reduce suspicious interference between features.
> > > > >
> > > > > We're glad to see that in addition to our main contribution of recovering temporal and instantaneous concept relations, our modeling technique can also potentially help mitigate the feature absorption issue in the original SAE framework. Your insightful feedback has been truly invaluable in uncovering this potential, and we're eager to share these new findings with the community and explore them further in future work. Please let us know if our response addresses your questions, and we'd be delighted to discuss any additional aspects you'd like to explore!

---

> > > > > > ### Comment · Reviewer_WPzT · 2025-08-06
> > > > > >
> > > > > > Thank you for sharing these extra details! My only concern is that the number of tokens (50M) is quite low for an SAE on an LLM - usually 300M+ is standard. Regardless, I find this very exciting and if it's true that your architecture also solves absorption in SAEs I think this is a very significant contribution to the field! I will further raise my score.

---

> ### Author Response · Authors · 2025-08-07
> **Updated Results on Your Suggested Experiment**
>
> Dear Reviewer `WPzT`,
>
> Your words are so encouraging and exciting to us. After your previous comments, we extended the training scale to 500M tokens for the Pythia-160m model, and to 200M tokens for the gemma-2-2b model. In both cases, we still observed very small absorptions, as seen from the table below:
>
> |Model|Mean Full Absorption Score|Mean Absorption Fraction Score|Mean Number of Split Features |
> |-|-|-|-|
> |pythia-160m-16k (500M tokens)|$6.471 \times 10^{-2}$|$9.185\times 10^{-3}$|$1.043$|
> |gemma-2-2b-16k (200M tokens)|$4.328 \times 10^{-2}$ |$9.620\times 10^{-4}$|$1.308$|
>
>
> Once again, we sincerely appreciate your valuable comments and suggestions for helping to improve our work. We hope this effort can contribute to refining the architecture of LLM interpretability methods.  By the way, the reviewer-author discussion will end soon but Reviewer Uu1b still hasn't yet responded to our response--if it is appropriate, a gentle nudge from you at some point would be much appreciated.
>
> Sincerely yours,
>
> Authors of Paper 7618

---

> ### Author Response · Authors · 2025-08-08
> **Thank You and Further Update on Your Suggested Experiment**
>
> Dear Reviewer `WPzT`,
>
> We are pleased to inform you that the training process for the setting you recommended (gemma-2-2b, 16k dimension, 300M tokens) has just completed. The results align well with the previous observations. Please find the summary below:
>
> |Model|Mean Full Absorption Score|Mean Absorption Fraction Score|Mean Number of Split Features |
> |-|-|-|-|
> |gemma-2-2b-16k (300M tokens)|$1.289 \times 10^{-2}$ |$3.794\times 10^{-4}$|$1.269$|
>
> Once again, all of the authors would like to sincerely thank you for your encouraging and insightful feedback on our manuscript and proposed method. Your perceptive observations on the absorption score have been especially valuable. We truly hope our work will inspire further reflection and discussion within both the mechanistic interpretability and causal representation learning communities.
>
> Sincerely,
>
> Authors of Submission 7618

---

### Official Review · Reviewer_gsa4 · 2025-07-02

**Clarity:** 3
**Significance:** 3
**Originality:** 3
**Rating:** 4
**Confidence:** 3

**Summary:**

In this work, the authors address the problem of interpretability in large language models (LLMs) through the framework of sparse autoencoders (SAEs). A known limitation of SAE is its lack of shared mechanisms across tokens, which hampers the discovery of interpretable concepts that capture temporal dependencies in LLM activations. Moreover, SAEs typically fail to model correlated concepts, limiting their ability to recover instantaneous relationships between them. The authors propose to overcome these limitations by modeling concepts using linear structural causal models (SCMs), which allows for both temporal and instantaneous causal relationships among concepts. Building on prior results from the causal representation learning (CRL) literature, they establish identifiability guarantees for the learned concepts and causal relationships, and empirically validate these findings on a synthetic dataset. Additionally, the authors present a case study on a real-world LLM dataset, demonstrating that their approach uncovers a range of meaningful temporal and instantaneous relationships.

**Questions:**

Please refer to the *weaknesses* section above for major questions. In addition, I have the following comments and clarifying questions for the authors:

- There seems to be a typo on line 77, I guess the matrix $A$ should have dimensions $m \times n$? Since in the theorems ahead authors represent the latent dimension of $z$ with the symbol $n$.

- There seems to be a typo on line 146, authors write $z_{it}$ while it should be $z_{ti}$.

- In Theorem 1, we defined $c_t = ( z_{t-1}, z_t, z_{t+1} )$ while in Theorem 2 we defined $c_t= (z_{t-1}, z_t)$. Why do we change the definition of $c_t$ across theorems?

- On line 131, authors should specify that the diagonal matrix $D$ is also invertible, as not all diagonal matrices are invertible.

- Could the authors explain the intimate neighbor set used in Theorem 1?

**Ethical Concerns:**

["NO or VERY MINOR ethics concerns only"]

**Final Justification:**

The authors addressed nearly all of my concerns during the rebuttal and I am have raised my score for leaning towards acceptance. Given my background being primarily causal representation learning rather than SAE, my concerns and suggestions were more centered around the theoretical results of identifiability and synthetic proof-of-concept experiments. I appreciate that the authors added new experiments for identifying steering vectors and concept relations. My concerns regarding the optimization objective for enforcing sparsity are not completely resolved and perhaps authors could improve more on that aspect. However, given that I am not an expert in mechanistic interpretability and what frameworks are typically used in the SAE literature, I am keeping my confidence 3.

**Limitations:**

Yes

**Paper Formatting Concerns:**

There are no major formatting concerns.

**Quality:**

2

**Strengths And Weaknesses:**

**Strengths**

- While the proposed approach and identifiability analysis builds largely on prior work and are not particularly novel on their own, the application of this temporal CRL framework to LLM interpretability is novel. Temporal relationships are important for LLM interpretability, and to the best of my knowledge, this is the first work to recover them using SAEs.

- The authors have done a good job with the experimental design. Scaling the synthetic dataset to include more concepts is an important step in demonstrating that the proposed approach can operate at the scale required for LLM interpretability tasks. Furthermore, the results in Appendix A.3.2 effectively justify the selection of key hyperparameters, such as the time window ($\tau$), sparsity penalty ($\beta$), and the dimensionality of the latent concept space.


**Weaknesses**

- The motivation for extending prior temporal CRL approaches to high-dimensional settings feels somewhat unclear. The proposed method relies on the assumption of a linear mixing function between concepts $z$ and observations $x$. Therefore, comparisons to existing CRL methods such as IDOL which are designed to handle the more general nonlinear case do not seem entirely appropriate. I suggest the authors tone down this aspect of their contributions. As currently framed (e.g., lines 57–58), the claim that this work significantly advances prior CRL methods appears overstated.

- The synthetic data results in Section 5.1 serve as a useful sanity check; however, a more comprehensive analysis of concept identifiability using (semi-)synthetic language benchmarks with ground-truth concepts would strengthen the evaluation. For instance, the authors could adapt synthetic benchmarks from SSAE [1] to the temporal domain. This would provide clearer insights into whether LLMs can recover true concepts from language-based synthetic datasets.

- The observation that a sparsity penalty of $\beta=0.01$ outperforms $\beta=0.1$ for learning sparse causal mechanisms is somewhat surprising. To better understand this behavior, I recommend the authors conduct experiments with a broader range of $\beta$ value, such as randomly sampling from the interval $(0.001, 0.1)$  to analyze the trend. Additionally, an important sanity check would be to evaluate the case $\beta=0$  which removes the sparsity regularizer entirely. Moreover, including ablation studies on the hyperparameter $\alpha$ would strengthen the paper; it would be valuable to see how variations in $\alpha$ affect training dynamics and the recovered mechanisms.

- While the case study results are interesting, the authors could further evaluate their approach on related tasks such as finding steering vectors using identifiable concept learning methods with LLMs [1, 2]. It would be valuable to know if the proposed approach can infer accurate steering vectors, and if not, what limitations the authors anticipate.

- The paper overall is well written, however, the writing of the theoretical aspects needs improvement.
     - The authors should avoid labeling their main theoretical result (Theorem 3) as a “theorem,” since it largely follows from Theorems 1 and 2 in prior work. Referring to it as a proposition or corollary would be more accurate.
     -  When presenting theoretical results, even from prior works, the authors should clearly define all assumptions and key terms. For example, the notion of component-wise identifiability for latent variable recovery is never formally defined. Similarly, the term “isomorphism” used in Theorem 1 (regarding the estimated Markov network being isomorphic to the ground truth) is not properly explained.
    - In Theorem 1, the authors never defined what is $\varepsilon(\mathcal{M} _{c_t})$?

References:

- [1] Joshi, Shruti, Andrea Dittadi, Sébastien Lachapelle, and Dhanya Sridhar. "Identifiable steering via sparse autoencoding of multi-concept shifts." arXiv preprint arXiv:2502.12179 (2025).

- [2] Rajendran, Goutham, Simon Buchholz, Bryon Aragam, Bernhard Schölkopf, and Pradeep Ravikumar. "Learning interpretable concepts: Unifying causal representation learning and foundation models." arXiv preprint arXiv:2402.09236 (2024).

---

> ### Author Rebuttal · Authors · 2025-07-30
>
> We sincerely appreciate the reviewer's detailed feedback and constructive suggestions. We address each concern below and will incorporate these improvements into our revised manuscript.
> >**W1. The motivation for extending prior temporal CRL approaches to high-dimensional settings feels somewhat unclear. The proposed method relies on the assumption of a linear mixing function between concepts $z$ and observations $x$. Therefore, comparisons to existing CRL methods such as IDOL which are designed to handle the more general nonlinear case do not seem entirely appropriate. I suggest the authors tone down this aspect of their contributions. As currently framed (e.g., lines 57–58), the claim that this work significantly advances prior CRL methods appears overstated.**
>
> **AW1:** We acknowledge this concern and here revised our framing to better reflect our contribution: "this work provides first scalable CRL framework capable of LLM interpretability and representation learning."
>
> Meanwhile, please also allow us to clarify our motivation. Our ultimate goal is to reliably discover the concept-to-concept relations behind the LLM activations. Temporal CRL methods such as IDOL are very good fit for this task, since they provide both empirical success in other domains and, more importantly, identifiability guarantees. But almost all existing CRL methods heavily suffer from scalability issues (as shown in Figure 5, they cannot scale beyond 200 dimensions). This naturally guided us to find a proper simplification for computational feasibility, in which linear approximation is one of the options.
>
> Luckily, existing work including the Linear Representation Hypothesis together with Sparse Autoencoder studies demonstrated an empirical result that LLM activations can be modeled via linear model. Such evidence makes the linear simplification / assumption less problematic in this specific LLM activation setting.
>
> > **W2,W4. The synthetic data results in Section 5.1 serve as a useful sanity check; however, a more comprehensive analysis of concept identifiability using (semi-)synthetic language benchmarks with ground-truth concepts would strengthen the evaluation. For instance, the authors could adapt synthetic benchmarks from SSAE [1] to the temporal domain. This would provide clearer insights into whether LLMs can recover true concepts from language-based synthetic datasets. While the case study results are interesting, the authors could further evaluate their approach on related tasks such as finding steering vectors using identifiable concept learning methods with LLMs [1, 2]. It would be valuable to know if the proposed approach can infer accurate steering vectors, and if not, what limitations the authors anticipate.**
>
> **AW2,W4:** Thank you for this insightful suggestion. We agree that synthetic benchmarks like SSAE [1] can offer valuable insights into concept identifiability. Our primary focus is on identifying concept relations, though we acknowledge its close ties to concept extraction and steering vector interpretation.
>
> To address this, we tested whether our model can recover steering vectors from paired text. Specifically, we constructed five categories of word pairs where only a single interpretable concept changes, including gender, plurality, comparative, tense, and negation. While these changes are intuitive, ensuring the word pairs capture a clear ground-truth concept is non-trivial. Despite this challenge, our model demonstrated strong performance in identifying the underlying concept differences. Specifically, (1) the average correlation of concept differences within each category exceeded 0.86; (2) assuming one ground-truth pair, the correlation rose above 0.93; and (3) the maximum correlation within a category reached over 0.94. These results support our claim that our model can indeed recover meaningful steering vectors.
>
> Moreover, building on your suggestion, we adapted SSAE [1] to test whether our model captures meaningful text relations. We constructed two contrastive collections of texts drawn from the Pile dataset. One collection consists of legal documents, which often follow structured syntactic formats from start to end, leading to stable temporal concept patterns. The other contains non-legal texts without such structure. We hypothesized that only legal texts would yield meaningful relational patterns. Indeed, we found that in the legal texts, two consistently activated concepts demonstrated a strong temporal relation, with a normalized weight of 0.70 in our recovered $B$. These concepts did not fire in non-legal texts, offering a clear contrast.
>
> These additional experiments reinforce our model’s ability to recover accurate, interpretable concept representations and relations. We have updated our manuscript to incorporate these experiments and expand on the related discussion. Thank you for your suggestion again.
>
>
> >**W3. The observation that a sparsity penalty of $\beta=0.01$ outperforms $\beta=0.1$ for learning sparse causal mechanisms is somewhat surprising. To better understand this behavior, I recommend the authors conduct experiments with a broader range of $\beta$ value, such as randomly sampling from the interval $(0.001, 0.1)$ to analyze the trend. Additionally, an important sanity check would be to evaluate the case $\beta=0$ which removes the sparsity regularizer entirely. Moreover, including ablation studies on the hyperparameter $\alpha$ would strengthen the paper; it would be valuable to see how variations in $\alpha$ affect training dynamics and the recovered mechanisms.**
>
> **AW3:** Thank you for the constructive suggestion. We conducted additional comparisons with $\beta = 0$, $0.001$, $0.005$, $0.05$, $1.0$ and $\alpha = 0$, $0.001$, $0.01$ to cover a broader hyperparameter range, using $\tau = 5$ and feature dimension 3072. The results are shown in the two tables below, with our selected setting *in italics*. The tables highlight that (1) concept relationships are inherently sparse, while a large $\beta$ disrupts optimization, and (2) $\alpha$ has a stronger effect, with 0.1 being a well-balanced choice. We also expanded our manuscript to include a broader discussion of model dynamics with respect to $\beta$ and $\alpha$.
>
> |$\beta$|0.0|0.001|0.005|*0.01*|0.05|0.1|1.0|
> |-|-|-|-|-|-|-|-|
> |Reconstruction Loss $\downarrow$|0.0126|0.0126|0.0126|*0.0128*|0.0126|0.0128|0.8950|
> |Independence Loss $\downarrow$|0.1522|0.1496|0.1504|*0.1448*|0.1550|0.1582|3.7682|
> |Bs Sparsity (L1) $\downarrow$|0.0003|0.0005|0.0005|*0.0007*|0.0007|0.0013|0.0053|
> |M Sparsity (L1) $\downarrow$|0.0001|0.0001|0.0001|*0.0001*|0.0001|0.0002|0.0007|
>
> |$\alpha$|0.0|0.001|0.01|*0.1*|1.0|
> |-|-|-|-|-|-|
> |Reconstruction Loss $\downarrow$|0.0227|0.0191|0.0118|*0.0128*|0.1121|
> |Independence Loss $\downarrow$|4.3849|2.4572|0.3910|*0.1448*|0.5252|
> |Bs Sparsity (L1) $\downarrow$|0.0012|0.0007|0.0018|*0.0007*|0.0058|
> |M Sparsity (L1) $\downarrow$|0.0002|0.0001|0.0002|*0.0001*|0.0009|
>
> >**W5. The paper overall is well written, however, the writing of the theoretical aspects needs improvement.**
>
> **AW5:** Thank you for this detailed feedback on improving the theoretical presentation. We have addressed all the suggested modifications and have updated to the paper:
>
> 1. We have relabeled our main result as "Proposition 3" instead of "Theorem 3" to more accurately reflect that it builds upon the foundational Theorems 1 and 2 from prior work.
> 2. A component‑wise transformation on a vector $\mathbf{x} = (x_1, \dots, x_n) \in \mathbb{R}^n$ is a map $T : \mathbb{R}^n \to \mathbb{R}^n$ that permutes the coordinates using a permutation $\sigma$ and applies possibly different scalar functions $f_i : \mathbb{R} \to \mathbb{R},\; i \in [n]$, resulting in $T(\mathbf{x}) = (f_1(x_{\sigma(1)}), \dots, f_n(x_{\sigma(n)}))$.
> 3. The $\mathcal{M}_{c_t}$ denotes the Markov network defined on the variable set of consequtive timestamps.
> 4. Isomorphism of Markov networks is formaly defined as follows. Let $V(\cdot)$ be the vertex set of any graph. An isomorphism of Markov networks $M$ and $\hat{M}$ is a bijection between the vertex sets of $M$ and $\hat{M}$, $f : V(M) \to V(\hat{M})$ such that any two vertices $u$ and $v$ of $M$ are adjacent in $M$ if and only if $f(u)$ and $f(v)$ are adjacent in $\hat{M}$.
>
> > **Q1-4. Comments and clarifying questions**
>
> **AQ1-4:** Thank you for pointing these out. We have corrected the typo on line 77: the matrix $A$ should indeed have dimensions $m \times n$, as $\mathbf{x} \in \mathbb{R}^m$ and $\mathbf{z} \in \mathbb{R}^n$. The expression on line 146 has also been corrected to $z_{t,i}$.
>
> Regarding the symbol $c_t$, we acknowledge the inconsistency in its usage across Theorems 1–3. In Theorem 2, $c_t$ refers to a context of two consecutive time steps, while in Theorems 1 and 3, it involves three time steps. To clarify, we will explicitly label the context length as $c_t^{(2)}$ and $c_t^{(3)}$ respectively.
>
> We also now specify that the diagonal matrix $D$ on line 131 is required to be invertible.
>
> > **Q5. Could the authors explain the intimate neighbor set used in Theorem 1?**
>
> **AQ5:** Certainly. The *Intimate Neighbor Set* is defined as follows: consider a Markov network $M_{Z}$ over variable set $Z$. The intimate neighbor set of variable $Z_i$ is given by
>
> $\Psi_{M_Z}(Z_{i}) \triangleq \{ Z_{j} \;|\; Z_{j} \text{ is adjacent to } Z_{i} \text{ and is also adjacent to all other neighbors of } Z_{i}, Z_{j} \in Z \setminus \{Z_{i}\} \}$
>
> This definition ensures that $Z_{j}$ shares maximal context with $Z_{i}$ in the conditional dependency structure, hence quantify the sparsity of the latent process. We have added this definition to the footnote on page 4 for clarity.

---

> ### Comment · Reviewer_gsa4 · 2025-08-05
>
> Thanks a lot for your response! My concerns regarding the theoretical aspects of the work are resolved and I am glad that authors agree stating results as Proposition instead of Theorem would be better. Further, I am happy with the additional experiments on finding steering vectors, they definitely make the paper strong. I have the following comments.
>
> - I am still not convinced by the motivation of scaling up existing CRL methods. As stated before, prior CRL methods were not specifically designed for the linear case, hence they cannot scale up with large latent variables. The authors can state in the introduction that leveraging the linear representation hypothesis, we design method specifically for the linear mixing scenario. Comparing the proposed approach against IDOL which is designed for a general case and not specialized for the linear case seems incorrect. I suggest the authors should move this experiment to appendix and instead focus on adding the new experiments where they show the method can recover latent variables from textual data and find steering vectors.
>
> - Regarding the ablation on $\beta$, its weird that $\beta=0.0$ also works quite well. It obtains Bs Sparsity (L1) as $0.0003$ while the best choice for $\beta$ leads to $0.0007$, is that difference meaningful? Maybe the authors should be incorporating sparsity penalty in a different manner as the optimization dynamics with the current objective seem non-intuitive.
>
> I would be happy to raise my score to leaning to accept (4) but I think the paper in its current form has issues as highlighted above, which make me hesitant to increase the score to accept (5). I will engage in discussions with the AC and other reviewers later and make my final decision then.

---

> ### Author Response · Authors · 2025-08-05
>
> Thanks again for the very thoughtful feedback and for being open to raising your score! Your comments have been very helpful for sharpening the paper. We are happy to address both points.
>
> **On the motivation and IDOL comparison:**
>
> We completely agree with your comment. We appreciate this great point that helps us sharpen the focus of our work. Yes, our primary contribution is not beating existing methods such as IDOL, but rather leveraging the linear representation hypothesis to design a CRL method specifically tailored to the linear mixing scenarios found in large-scale LLMs. The goal is to bridge the gap between theoretically rigorous CRL algorithms and their practical application to modern NLP, thereby encouraging the CRL community to make theoretical advances more accessible for large-scale empirical studies.
>
> Following your valuable suggestion, we have revised the manuscript to reflect this focus. (1) The introduction now explicitly frames our work as a specialization for the linear case, motivated by the need for practical CRL tools for LLM analysis. (2) We have moved the scalability discussion and the direct comparison with IDOL from the main experiments section (section 5.1) to the appendix. (3) The main body of the experiments section now includes the new results on identifying steering vectors and concept relations, which better showcase the unique capabilities of our approach.
>
> We hope these changes address your concern on this point. We thank you again for guiding us toward this clearer presentation.
>
> **On the ablation result for $\beta$:**
>
> We appreciate you pointing out the seemingly counterintuitive results, and we are happy to provide more context on the optimization dynamics.
>
> First, as you noted, the absolute sparsity values (0.0003 vs. 0.0007) both seem negligible, particularly when considering that the Independence Loss term is approximately three orders of magnitude larger. Comparing these minimal values might not yield meaningful conclusions about model performance.
>
> Second, we wish to point out that the architecture may embed an inherent sparsity constraint on the relation matrices even without an explicit L1 penalty. For example, the TopK nonlinearity naturally encourages sparse activations, which in turn leads to sparse relationship matrices ($B$ and $M$). Furthermore, standard regularization, like weight decay, also contributes to this effect. In fact, when we attempted to train the model with both $\beta=0$ and no weight decay, the training process became unstable, showing the effect of these implicit sparsity-inducing mechanisms.
>
> Third, the optimization process involves a trade-off between multiple objectives, not just minimizing sparsity, which means the choice of $\beta$ impacts the other loss terms. We found that the $\beta=0$ case, despite its low L1 value, achieved a higher Independence Loss (0.1522) compared to our chosen $\beta=0.01$ (0.1448). Since the independence of the noise terms is a cornerstone of our method's identifiability guarantees, successfully optimizing this loss is critical. This demonstrates that focusing solely on the final sparsity value can be misleading (e.g., zero sparsity trivially leads to an empty and meaningless causal graph); considering the overall optimization landscape and adherence to theoretical assumptions should be more reasonable.
>
> Finally, regarding the choice of sparsity constraint, we acknowledge that the L1 norm is a practical proxy for the theoretically desired L0 norm. As is well-known, the non-differentiable nature of the L0 norm makes it very challenging to optimize directly. As the first work to handle real, LLM-scale data with a provable CRL method, we used the most direct and widely adopted solution in the sparsity constraint literature. We acknowledge this challenge and leave more detailed exploration of this topic for future work.
>
> Thanks again for the constructive review! We have integrated this detailed discussion into the appendix of our revised manuscript to clarify these dynamics for readers. We hope these changes address your concerns and look forward to the opportunity to engage in any further conversations with you.

---

> ### Author Response · Authors · 2025-08-08
> **Please let us know if you have further questions**
>
> Dear Reviewer `gsa4`,
>
> Thank you again for your detailed and constructive review. We sincerely appreciate your thoughtful suggestions, which have helped us strengthen several aspects of our paper.
>
> We hope that our recent responses have addressed your remaining concerns and clarified the points you raised. As the discussion period nears its end, we would be grateful for any further feedback or questions you may have. If appropriate, we would also appreciate an update to your recommendation. We would be happy to provide any additional clarifications that could assist in your final assessment.
>
> Sincerely,
>
> Authors of Paper 7618

---

### Official Review · Reviewer_Uu1b · 2025-07-03

**Clarity:** 3
**Significance:** 2
**Originality:** 2
**Rating:** 3
**Confidence:** 3

**Summary:**

The authors claim four contributions:
1. A framework to model time-delayed causal relations between concepts and instantaneous constraints, with theoretical guarantees in the direction of reliability and explainability.
2. Algorithms tailored to causal representation learning for the high-dimensional concept space of LLMs.
3. Synthetic experiments supporting the results, scaled to match real-world complexity and applied on real LLMs.

**Questions:**

1. Usually, LLM activations are viewed as attention computations, and there is no sequential time dependence. Why was the modeling choice of sequential time dependence made in this work? Is there evidence support the claim that this is superior to an attention style of activation modeling? Is there a benefit of having an SEM, which induces interventional distributions, rather than a probabilistic model?
2. Are the latent variables $\mathbf z_t$ usually confirmed to be identifiable, given that there are many more latent than observed variables? What is the value of showing that identifiability holds in the synthetic setting in Experiment 5.1?
3. Is there evidence to support the claim that the proposed SEM in equations (1), (2), and (3) is consistent with observational activations generated by real LLMs? For example, how well does the proposed model fit the actual distribution of activations? Is there any baseline for how well a model can fit this distribution, and does the proposed model improve upon it?
4. Given that the model proposed is an SEM rather than a purely observational model - is there evidence to support the claim that there are the instantaneous and temporal mechanisms can be modeled linearly in the specific way this work models them?
5. Is there evidence to indicate whether or not the preconditions of Thm 1-3 are met, sparsity in particular?

**Ethical Concerns:**

["NO or VERY MINOR ethics concerns only"]

**Final Justification:**

Authors claim three new results: competitiveness w.r.t. the baseline of SAE in activation reconstruction, and potential improvements in absorption and concept recovery. The results are not claimed with statistical soundness.

**Limitations:**

Yes.

**Paper Formatting Concerns:**

None.

**Quality:**

2

**Strengths And Weaknesses:**

Originality: the authors introduce a linear SEM to model LLM activations. I am unfamiliar with the related mechanistic interpretability literature, but this concept is novel to my knowledge.

Quality and significance: It is shown that the underlying latent variables in the SEM can be identified under some assumptions, including sparsity of latent components. It does not seem to be shown that the SEM can model the distribution of activations; case studies are performed (selecting <=6 examples of specific types of successes in total), but I did not see an objective or large-scale analysis supporting this claim. In other words, evaluation appears to be limited.

Clarity: contributions, methodology, theory, and results are stated clearly.

---

> ### Author Rebuttal · Authors · 2025-07-30
>
> We thank the reviewer for the thoughtful review and the opportunity to clarify key aspects of our work. We address each concern below.
>
> > **W1. It does not seem to be shown that the SEM can model the distribution of activations; case studies are performed (selecting <=6 examples of specific types of successes in total), but I did not see an objective or large-scale analysis supporting this claim. In other words, evaluation appears to be limited.**
>
> **AW1:** Thank you for this thoughtful comment. Our primary contribution is to identify latent concepts and their temporal or instantaneous relations under sparse structural assumptions with theoretical guarantees. While modeling the full distribution of LLM activations is not our focus, our method captures the structure of the activation space well—evidenced by near-zero reconstruction loss (the first columns of Figures 8 and 9).
>
> Though only a few case studies appear in the main text due to space limits, we include over 20 additional examples in Tables 2 and 3 (appendix), along with longer-range temporal analyses demonstrating richer semantic dynamics. Additional visualizations and discussions will be included in the updated manuscript.
>
> To address evaluation rigor, we added SAEBench experiments using standard metrics (e.g., reconstruction loss, sparse probing), showing our model performs comparably to strong SAE baselines (see our responses AQ3 to Reviewer WPzT). We also introduce a semi-synthetic legal-vs-non-legal task to evaluate our model’s ability to extract interpretable relations (see our responses AW2,W4 to Reviewer gsa4).
>
> These tasks are aligned with our central goal: interpretable representation and relational discovery, rather than generative modeling. Finally, we emphasize that meaningful, human-aligned examples are critical for LLM interpretability. We will clarify this distinction and further strengthen both empirical and illustrative support in the revision.
>
> > **Q1a. Usually, LLM activations are viewed as attention computations, and there is no sequential time dependence. Why was the modeling choice of sequential time dependence made in this work? Is there evidence support the claim that this is superior to an attention style of activation modeling?**
>
> **AQ1a:** We first clarify that there is indeed sequential time dependence in attention computation, as the current position needs to attend to previous positions. Additionally, there is natural sequential dependence in the semantic concept space during text generation, i.e., what you think or write now definitely depends on what you thought or wrote in the past. Therefore, when using learned activations to interpret LLMs' internal processes, sequential modeling is a very natural choice.
> Regarding attention-style modeling, it is known to be highly inexplicable, which is precisely the motivation behind our proposal.
>
> > **Q1b. Is there a benefit of having an SEM, which induces interventional distributions, rather than a probabilistic model?**
>
> **AQ1b:** Actually SEM is a probabilistic model; the distinction is that an SEM specifies how variables respond to explicit interventions, not just how they co‑vary observationally. Our objective is to explain how one internal concept can influence another within the LLM. This question is inherently interventional. The benefit of an SEM over any associational probabilistic model is that it explicitly models causal mechanisms, allowing us to understand not just correlations but directional influences between concepts.
>
> > **Q2a. Are the latent variables $\mathbf{z}_t$ usually confirmed to be identifiable, given that there are many more latent than observed variables?**
>
> **AQ2a:** Yes, this is a standard setting in CRL and sparse dictionary learning literature which usually referred with term "over-complete" setting. A number of approaches [3,A,B] including our work leverage the structural sparsity to establish the identifiability.
>
> >**Q2b. What is the value of showing that identifiability holds in the synthetic setting in Experiment 5.1?**
>
> **AQ2b:** For sanity check! Theoretical identifiability proofs do not automatically guarantee that practical algorithms can recover latent variables under realistic constraints (finite data, noise, optimization challenges). And experiment 5.1 bridges this theory-practice gap by using synthetic data with known ground truth to confirm our implementation converges to the theoretically identified solution, and establish performance baselines before applying to real data where ground truth is unavailable. Such synthetic-to-real validation is a standard methodology in CRL.
>
> >**Q3. Is there evidence to support the claim that the proposed SEM in equations (1), (2), and (3) is consistent with observational activations generated by real LLMs? how well does the proposed model fit the actual distribution of activations? Is there any baseline for how well a model can fit this distribution, and does the proposed model improve upon it?**
>
> **AQ3:** Yes! We first demonstrate our modeling consistency with observational activations via the convergence of reconstruction loss (Figures 8-9 in Appendix) to sufficiently small values. We also compare it with estabilished SAE baselines:
> |Model|Reconstruction Loss $\downarrow$|
> |-|-|
> |ReLU SAE|0.0110|
> |TopK SAE|0.0097|
> |Ours|0.0108|
>
> Our method also achieves comparable reconstruction loss (~0.01), which further confirms the consistency is at least on-par with those SAEs.
>
> We emphasize that distribution matching, while necessary, **is not our primary contribution**. Any universal approximator can achieve near-perfect reconstruction. The critical challenge is **uniquely identifying the latent causal structure** underlying these activations. Our method not only fits the data distribution but also provides theoretically grounded guarantees for recovering the true causal relationships--Section 5.2 demonstrates that recovered causal dependencies align with known semantic patterns in language--which existing methods like SAEs cannot provide.
>
> >**Q4. Given that the model proposed is an SEM rather than a purely observational model - is there evidence to support the claim that there are the instantaneous and temporal mechanisms can be modeled linearly in the specific way this work models them?**
>
> **A4:** Yes! And we address this question through three key points:
>
> **SEM v.s. purely observational model** Please allow us to clarify that SEM could model both observational and latent variables; so they are not mutually exclusive.
>
>
> We employ latent variables in SEM to model LLM activation dynamics (rather than observationally) because LLM activations are inherently polysemantic [D]: each dimension encodes multiple entangled concepts, making direct concept-to-concept analysis infeasible in the raw activation space. Such latent variable approach is well-established in literatures.
>
> **Evidence for instantaneous and temporal mechanisms** The existence of such causal relationships has been extensively studied in the mechanistic interpretability community, with recent empirical studies [1,28] confirming these relations  (line 32 and footnote 1).
>
> **Justification for our linear modeling** Existing work [41] showed semantic features occupy linear subspaces, and SAEs [28,50,D] successfully use sparse linear model for finding concepts. Those established the fundation for linear modeling. More importantly the evidence comes from our own successful recovery of semantically meaningful causal structures (Section 5.2). This establishes linearity as an empirical reality at the relevant abstraction level rather than a simplifying assumption.
>
> > **Q5. Is there evidence to indicate whether or not the preconditions of Thm 1-3 are met, sparsity in particular?**
>
> **A5:** This is a very good question! Verifying the **latent assumptions** about hidden variables is a fundamental challenge to all research on latent variable identification [E] since latent variables are not directly measurable. We approach to their reasonableness through **indirect validation**: if the model successfully fits the observed data distribution and consistently yields to meaningful solutions, this provides evidence that the underlying assumptions are moderate and feasible. This approach provides a practical solution for this challenge and are widely acknowledged in the community [1,54].
>
> **Regarding sparsity in particular**, evidence partially comes from empirical success from the SAE and CRL literatures [25,50,A,B,C,D] to use sparsity constraints to extract concepts recover non-trival causal relations. More importantly, evidence also comes from our own successful recovery of semantically meaningful causal structures (Section 5.2) together with the convergence behavior of our loss curve during training.
>
> We acknowledge that these assumptions, like any in this field, may not hold universally, our empirical results suggest they seem effective and feasible for the complex and real data we study.
>
> [A] Sun, Yuchen, and Kejun Huang. "Global identifiability of overcomplete dictionary learning via l1 and volume minimization." The Thirteenth International Conference on Learning Representations. 2024.
>
> [B] Chen, Siyu, et al. "Taming Polysemanticity in LLMs: Provable Feature Recovery via Sparse Autoencoders." arXiv preprint arXiv:2506.14002 (2025).
>
> [C] Wang, Kexin, and Anna Seigal. "Identifiability of overcomplete independent component analysis." arXiv preprint arXiv:2401.14709 (2024).
>
> [D] Elhage, et al., "Toy Models of Superposition", Transformer Circuits Thread, 2022.
>
> [E] Peters, Jonas, Dominik Janzing, and Bernhard Schölkopf. Elements of causal inference: foundations and learning algorithms. The MIT press, 2017.

---

> > ### Author Response · Authors · 2025-08-05
> > **Looking Forward to the Feedback from Reviewer Uu1b**
> >
> > Dear Reviewer `Uu1b`,
> >
> > Thank you again for your thoughtful and constructive review. We sincerely appreciate the time and care you have taken in evaluating our work.
> >
> > We hope your concerns were properly addressed by our response. Also, as the end of the reviewer-author discussion period approaches, we are keen to hear your feedback and hope for the opportunity to respond to it.
> >
> > Sincerely,
> >
> > Authors of Paper 7618

---

> > ### Comment · Reviewer_Uu1b · 2025-08-07
> > **Response to authors**
> >
> > I thank the authors for their thoughtful replies, which address some of my questions. Several of my questions are not addressed, so I leave my score as it is.
> >
> > W1/Q3. I am unsure how Fig. 8 and 9 in the Appendix support the claim that activations are modeled well. What does "near-zero" reconstruction loss mean? Is the loss compared to a baseline in the prior literature to show improvement? How do we know that the reconstruction loss is "good"? Case studies are useful for developing intuition for a model, but are less so for making statistically sound claims; adding more case studies would not help in the direction of showing that activations can be reconstructed. Regarding the SAE baselines, is an improvement shown with statistical significance?
> >
> > Q1a. I will clarify my original statement and rephrase my question. There is non-sequential time dependence in attention models that is not captured by this work's model. While the authors' intuitive explanation may make sense on a subjective basis, it would be desirable to see convincing empirical evidence that this lossy modeling choice is not losing important information on activations by making this modeling choice. Can this information that is lost be quantified?
> >
> > AQ1b/Q4. A SEM can be used for the purpose of observational modeling, but it also induces interventional distributions on its endogenous variables. Thank you for confirming that the SEM is only used for observational modeling (i.e., modeling the joint distribution of the observed variables without intervening on them) and not for interventional modeling (i.e., modeling distributions on interventions of observed variables). It then seems fair to say that SEMs are not applied to perform interventions on observed or identified latent variables in the work. It is possible this could be an interesting direction to explore.
> >
> > It seems that the authors propose a new modeling technique for neural activations. The authors claim that the work performs on par with the baseline of SAE in terms of reconstruction loss. Is there an improvement shown in the current work over SAE? The authors claim their primary contribution to be "uniquely identifying the latent causal structure" underlying activations. While this is done under the parametric assumptions made, it's unclear how this is useful, or what improvement the model or assumptions make over prior baselines. Can the authors clarify this?

---

> > > ### Author Response · Authors · 2025-08-08
> > > **Response to Further Comments on W1/Q3**
> > >
> > > We thank the reviewer for the continued engagement. We are very grateful for this chance to address your remaining concerns. We thank you for the patience, and please see our responses to your questions one by one.
> > >
> > > > **W1/Q3. I am unsure how Fig. 8 and 9 in the Appendix support the claim that activations are modeled well. What does "near-zero" reconstruction loss mean? Is the loss compared to a baseline in the prior literature to show improvement? How do we know that the reconstruction loss is "good"? Case studies are useful for developing intuition for a model, but are less so for making statistically sound claims; adding more case studies would not help in the direction of showing that activations can be reconstructed. Regarding the SAE baselines, is an improvement shown with statistical significance?**
> > >
> > > We appreciate the reviewer's further engagement and acknowledge the conceren about the reconstruction loss of our method. Indeed, an objective testing method applied to the reconstruction losses obtained from our both model and standard SAEs would be more informative--it can compare how much of the informationis retained from the LLM activations, which is the basis of further interepretation steps. Please see our responses to your questions below:
> > >
> > > The "near-zero" refers to the convergent and vanishing reconstruction loss we observed in the training loss curve. We did make comparisons with the established SAE baselines and reported the numbers in the table of **AQ3** in our original response. Meanwhile, we are not claiming to achieve better reconstruction performance, hence a "sufficiently good" reconstruction loss that we aim to achieve is to match the performance of established SAE methods. In this case, we thank the reviewer for acknowledging our case studies are useful for developing intuition and we are happy to further provide statistical evidence for our claim "our method has on-par performance as existing SAE baselines".
> > >
> > > To *make statistically sound claims*, beyond the very close mean reconstruction losses we observe for ReLU SAE (0.01293), TopK SAE (0.01006), and our method (0.01166), we designed the null hypothesis $H_0$ as "the population mean training loss is the same for all three methods. The corresponding alternative hypothesis $H_A$ is "at least one method has a different population mean training loss". The Kruskal–Wallis test provided $p=0.40$, which exceeds the $\alpha = 0.05$ threshold, so we do not reject $H_0$. Hence, there is no statistical evidence that the training loss differs among the three methods, supporting the claim that the models achieve approximately equivalent fitting performance.
> > >
> > > We hope our follow up response addresses your concern about the reconstruction loss. And please allow us to defer the discussion about our improvement over SAE baselines to our response for your final question.

---

> > > > ### Author Response · Authors · 2025-08-08
> > > > **Response to Further Comments on Q1a**
> > > >
> > > > > **Q1a. I will clarify my original statement and rephrase my question. There is non-sequential time dependence...**
> > > >
> > > > Thank you for revisiting your previous question. We truly appreciate the additional clarification. In what follows, we address your two concerns.
> > > >
> > > > **About non-sequential time dependence**
> > > >
> > > > Regarding the non-sequential time dependence, since it lacks a formal definition, we believe this term may refer to one of the three following cases. We genuinely hope that one of our interpretations can align with your intended point; if not, we would be grateful for additional context.
> > > >
> > > > First, non-sequential dependency might refer to the *direct* dependency between one activation and any other activation in the self-attention mechanism (or in causal self-attention mechanism, where “any” is restricted to past positions). In contrast, sequential dependency refers to the *indirect* dependency between two activations, constructed through several adjacent local context windows. If this is the case, then using a sufficiently large $\tau$, such that all non-sequential dependencies occur within this context window, should enable our model to capture the potentially missed dependencies.
> > > >
> > > > Second, this term might also refer to cross-channel dependency, i.e. the integration over time between $Z_{i,t}$ with $Z_{j,t-\tau}$, where $i\neq j, \tau \geq 0$, as oppposite to sequential dependence, i.e. $Z_{i,t}$ with $Z_{i,t-\tau}$ where $\tau \geq 0$. If it is the case, then not surprisingly, $A$, $M$, and $B_\tau$ in the proposed model (Eq. 3) can produce a wide range of non-sequential dependencies. For illustraive purposes, let's consider an extreme situation where X is joint Gaussian, and therefore we can use auto-covariance to illustrate the dependencies. For simplicity, consider a one-lag model, i.e., $X_t = AZ_t$ and $z_t = B Z_{t-1} + M Z_t + \epsilon$. Then, the auto-covariance matrix of $Z_t$ with lag $\tau$ can be expressed recursively as:
> > > > $$ \Sigma_Z(\tau) = (I-M)^{-1} B \Sigma_Z(\tau-1)$$ if $\tau>1$, and the covariance matrix can be obtained as a solution to
> > > > $$ \Sigma_Z(0) = (I-M)^{-1} B \Sigma_Z(0) B^\intercal (I-M)^{-\intercal}.$$
> > > > These generally introduce non-sequential dependence relations. It also happen to the (auto-) covariance matrix of $X_t$. In practice, of course, the class of non-sequential dependence is contrained by the model.
> > > >
> > > > Finally, as we noticed you incoporated *time* into the term, so we suspect that sequential time dependence might refer to the causal attention mechanism, where only past activations (in a given layer) could attend to the current activation, i.e. in decoder-only LLMs, as oppose to the bidirectional self-attention used in BERT. If this is the case, we would like to respectfully point out that modern mainstream LLMs, such as GPT, LLaMA, and the ones we study for interpretability (Gemma and Pythia), are all decoder-only LLMs. In this context, our use of a sequential model directly reflects the actual attention computation pattern in these models.
> > > >
> > > > We hope our understanding about non-sequential time dependence can make us on the same page and we can further address your concern about the potential losing infomation bellow:
> > > >
> > > >
> > > > **About losing important information**
> > > >
> > > > As you may agree, it might be difficult to precisely define or quantify what constitutes "important information", as interpretations of “importance” can vary. However, a practical way to quantitatively evaluate such potential information loss is by examining the reconstruction loss. We believe that if our model can reconstruct the activations with very small error, it is reasonable to conclude that our proposed modeling retains most of the important information in the activations. This evaluation strategy applies regardless of the definition of non-sequential time dependency.
> > > >
> > > > As we demonstrated in our response to your earlier question on reconstruction loss, our model achieves performance at least on par with other mainstream mechanistic interpretability methods in preserving information from LLM activations. We acknowledge that some degree of information loss may be inevitable: for example, under our first interpretation of non-sequential time dependency, when using a restrictive value of $\tau$. However, the meaningful temporal relations we extracted and presented in our qualitative example, together with the quantitative analysis in the relation recovery experiment, both indicate that our modeling method preserves reasonably sufficient information for our proposed task.
> > > >
> > > > Finally, we wish to reiterate why we used a sequential approach for modeling: our goal is to provide a reliable and clear picture of how concepts connect and how their relationships evolve over time. A sequential model can naturally demonstrate such relationships (for example, through the matrices B and M in our formulation). This capability provides insights that go well beyond what standard SAE approaches for LLM interpretability can offer.

---

> > > > > ### Author Response · Authors · 2025-08-08
> > > > > **Response to Further Comments on AQ1b/Q4 and Final Paragraph**
> > > > >
> > > > > > **AQ1b/Q4. A SEM can be used for the purpose of observational modeling, but it also induces interventional distributions on its endogenous variables. Thank you for confirming that the SEM is only used for observational modeling (i.e., modeling the joint distribution of the observed variables without intervening on them) and not for interventional modeling (i.e., modeling distributions on interventions of observed variables). It then seems fair to say that SEMs are not applied to perform interventions on observed or identified latent variables in the work. It is possible this could be an interesting direction to explore.**
> > > > >
> > > > > Thank you for your further comments on our model choice of SEM. We understand your comment as an acknowledgment of our use of SEM for modeling the observational distributions in LLM activations, along with a kind suggestion to explore the use of SEM for analyzing interventional distributions. We completely agree with this suggestion and would like to express our sincere thanks for proposing this direction. We would be happy to explore it in future work.
> > > > >
> > > > >
> > > > > > **It seems that the authors propose a new modeling technique for neural activations. The authors claim that the work performs on par with the baseline of SAE in terms of reconstruction loss. Is there an improvement shown in the current work over SAE? The authors claim their primary contribution to be "uniquely identifying the latent causal structure" underlying activations. While this is done under the parametric assumptions made, it's unclear how this is useful, or what improvement the model or assumptions make over prior baselines. Can the authors clarify this?**
> > > > >
> > > > >
> > > > > Thank you for giving us the opportunity to clarify the distinction between our model and standard SAEs, as both aim for LLM interpretability. Indeed, we have shown that we can match the performance of mainstream SAEs in terms of reconstruction loss (see our response to your first point). Furthermore, we improve upon SAE by identifying the causal structure among the recovered concepts and providing theoretical guarantees. Such relation recovery is crucial and needed in the LLM interpretability literature [A, B].
> > > > >
> > > > > In detail, from a theoretical perspective, SAEs do not provide guarantees for concept recovery but merely decompose activations. Our method addresses these limitations and provides theoretical guarantees for recovering the true causal relationships between concepts.
> > > > >
> > > > > Empirically, we demonstrated through qualitative examples in Sec. 5.2 and quantitative analysis in a semi-synthetic experiment on concept relation recovery (in our response AW3 to reviewer `KYay`) that our method can recover the concept relations. Besides, thanks to the suggestions from reviewer `WPzT`, we were pleased to find that our proposal also shows potential in addressing the absorption problem in SAEs. These points summarize our improvements over standard SAEs.
> > > > >
> > > > > As for comparison with CRL baselines such IDOL, we assumed a linear parametric functional form for the underlying dynamics of temporal and instantaneous relations. This assumption is well aligned with the linear representation hypothesis, which is widely adopted in the LLM interpretability literature. More importantly, this assumption allows us to significantly extend the dimensionality that provable CRL methods can handle (from fewer than 200 dimensions in existing CRL methods to thousands of dimensions in our method). This extension bridges the gap between provable CRL methods and real-world LLM interpretability.
> > > > >
> > > > > ---
> > > > >
> > > > > We hope our responses could successfully resolve your concerns. While we acknowledge that we may need some additional clarification from the reviewer to further address some of the concern, we have tried our best to give our understanding of their question and respond with several potential interpretations. Please kindly let us know if you have further questions and we are actively looking forward to your reply.
> > > > >
> > > > >
> > > > > [A] Ameisen, et al., "Circuit Tracing: Revealing Computational Graphs in Language Models", Transformer Circuits, 2025.
> > > > >
> > > > > [B] Lindsey, et al., "On the Biology of a Large Language Model", Transformer Circuits, 2025.

---

> ### Comment · Reviewer_Uu1b · 2025-08-09
>
> I thank the authors for their detailed response.
>
> I am updating my score by one point, given the new results comparing reconstruction to the SAE baseline with a significance test. I will also respect the decision of the ACs on this work.
>
> As a side-note, the statistical test on the comparison to SAEs seems to be missing a power value for indistinguishability - what is the probability that the null hypothesis would be rejected if an alternative hypothesis was true? Was it by random chance that the p-value was not significant? What was the sample size of the experiments run?
>
> The claims regarding improvements on concept relation recovery and absorption are strong, but they appear to be missing p values, so I cannot evaluate their statistical soundness.

---

### Note · Authors · 2025-08-13

We thank the Area Chair and Reviewers `Uu1b`, `gsa4`, `WPzT`, and `KYay` for their valuable time and feedback. Below are our responses, followed by a summary of contributions.

To Reviewer `Uu1b`: Thank you for your further comments—we agree that additional statistical testing strengthens the work. Due to the strict word limit, we highlight only the key points here. To show that the reconstruction losses are at the same level, we used 100 samples per method ($N=300$). Any shift $\ge 0.00127$ across groups would be detected with power $\ge 0.8$. To directly assess equivalence, pairwise Welch–TOST and Hodges–Lehmann tests with $\Delta=0.001$ showed that all 90% CIs lay within $[-\Delta,\Delta]$, confirming statistical equivalence at $\alpha=0.05$ among the three methods.

Regarding absorption, in our conversation with Reviewer `WPzT`, it was noted that our method potentially achieves small absorption scores, which are challenging to evaluate with a rigorous statistical test. However, we can still demonstrate statistical soundness by collecting a sufficiently large number of samples ($\ge$ 200). The mean and 95% CIs obtained (6k: 0.0135 $\pm$ 0.0002 and 16k: 0.0136 $\pm$ 0.0002) are more than sufficient to demonstrate negligible absorption.

While limited time prevented us from aligning hidden dimensions across many model pairs for further statistical tests, we are actively working on this. Nevertheless, the reported signal-to-noise ratio already provides evidence of significance, and the large gap between our model (20.02) and the SAE baseline (2.39) is unlikely to arise from random noise.

All new results are included in the updated manuscript. We thank you again for your time and consideration.

To Reviewers `gsa4`, `WPzT`, and `KYay`: We appreciate your insightful feedback and encouragement. We have provided further responses (including some with additional experiments) and hope that the updates address your concerns.

**Summary of contributions:** We propose a novel, theoretically grounded framework for modeling temporal and instantaneous concept relations in LLM interpretability. As Reviewer `WPzT` noted, "...this is a very significant contribution to the field...", we hope this contribution will inspire substantial developments in the field of LLM interpretability and help bridge the gap between empirical studies and theoretical investigations in representation learning and architecture design for LLMs.

Sincerely,

Authors of Submission 7618

---

### Decision · Program_Chairs · 2025-09-17

**Decision:**

Accept (poster)

**Comment:**

This paper presents a framework for the interpretability of large language models by combining notions of sparse autoencoders with a temporal causal representation learning (CRL) approach. The method introduces (linear) identifiable structural causal models to capture both temporal and instantaneous relations among concepts, and provide theoretical guarantees under the linear representation hypothesis. The authors present synthetic and semi-synthetic experiments, as well as case studies on LLM activations, and benchmark against standard SAEs.

**Strengths**:
* While the CRL approach is not novel, their applications to mechanistic interpretability is (as noted by reviewers), with the ability to recover temporal as well as instantaneous relations among concepts.
* Theoretical results providing identifiability, extending beyond what standard SAEs provide.
* Empirical results show competitive reconstruction performance to SAE baselines whenever comparisons are possible, and new experiments during rebuttal (e.g., SAEBench metrics, steering vectors, relation recovery tasks) strengthened the contribution.
* Reviewer discussions highlighted the potential for addressing feature absorption, an important challenge in SAE interpretability.

**Weaknesses**:
* Evaluation remains somewhat limited: while semi-synthetic and case study results are convincing, large-scale benchmarks and broader statistical testing are still lacking (mostly addressed during rebuttals).
* Some framing in the initial submission overstated the contribution relative to prior CRL methods (e.g., IDOL); authors agreed to soften this claim.
* Certain assumptions (linear mixing, sparsity) may restrict the generality of the approach, and their justification relies heavily on the linear representation hypothesis.

**Outcomes of the discussions**:
The back-and-forth discussion was productive. The authors added statistical testing against SAE baselines, new semi-synthetic experiments (e.g., steering vectors, legal vs. non-legal texts), and extended SAEBench evaluations. Reviewers gsa4 and WPzT increased their confidence in the work, with WPzT emphasizing its significance for the interpretability community. Reviewer Uu1b remained more skeptical, noting unresolved concerns around usefulness and statistical soundness, though they did raise their score after rebuttal. Reviewer KYay’s review remained critical, largely focusing on the use of synthetic time series; the authors responded with clarifications and further semi-synthetic results. Overall, I regard this latter review as quite superficial.

**Summary**:
Overall, this paper presents an interesting and theoretically grounded step toward interpretable and identifiable causal structure discovery in LLMs. The novel theoretical framing, reported results, and strengthened experiments during the discussions, justify its acceptance as a poster. I encourage the authors to include the points and discussions raised during the rebuttal and discussion phase.